



**Carbon dioxide and methane fluxes from different surface types**
**in a created urban wetland**
Xuefei Li[1], Outi Wahlroos[2], Sami Haapanala[3], Jukka Pumpanen[4], Harri Vasander[5], Anne Ojala[1, 5, 6]
Timo Vesala[1, 5] and Ivan Mammarella[1]
[1]Institute for Atmospheric and Earth System Research (INAR)/Physics, Faculty of Science, University of Helsinki, P.O.
Box 68, 00014 University of Helsinki, Finland
[2]University of Turku, Turku, Finland
[3]Suvilumi, Ohrahuhdantie 2 B, 00680 Helsinki, Finland
[4]Department of Environmental and Biological Sciences, University of Eastern Finland, P.O. Box 1627, 70211 Kuopio,
Finland
[5]Institute for Atmospheric and Earth System Research (INAR)/Forest Sciences, Faculty of Agriculture and Forestry,
University of Helsinki, P.O. Box 27, 00014 University of Helsinki, Finland
[6]Ecosystems and Environment Research Programme, Faculty of Biological and Environmental Sciences, University of
Helsinki, P.O. Box 65, 00014 University of Helsinki, Finland
*Correspondence to*: Xuefei Li (xuefei.z.li@helsinki.fi)





**Abstract.** Many wetlands have been drained due to urbanization, agriculture, forestry or other purposes, which has resulted in losing their ecosystem services. To protect receiving waters and to achieve services such as flood control and stormwater quality mitigation, new wetlands are created in urbanized areas. However, our knowledge of greenhouse gas exchange in newly created wetlands in urban areas is currently limited. In this paper we present measurements carried out at a created urban wetland in boreal climate.

We conducted measurements of ecosystem $CO_2$ flux (NEE) and $CH_4$ flux ($F_{CH4}$) at the constructed stormwater wetland Gateway in Nummela, Vihti, Southern Finland using eddy covariance (EC) technique. The measurements were commenced the fourth year after construction and lasted for one full year and two subsequent growing seasons. Besides ecosystem scale fluxes measured by EC tower, the diffusive $CO_2$ and $CH_4$ fluxes from the open-water area ($F_w\_CO_2$ and $F_w\_CH_4$, respectively) were modelled based on measurements of $CO_2$ and $CH_4$ concentration in the water. Fluxes from vegetated area were estimated by applying a simple mixing model using above-mentioned fluxes and footprint-weighted fractional area. The half-hourly footprint-weighted contribution of diffusive fluxes from open water ranged from 0 to 25.5 % in year 2013.

The annual NEE of the studied wetland was 8.0 g $C-CO_2$ $m^{-2}$ $yr^{-1}$ with the 95 % confidence interval between -18.9 and 34.9 g $C-CO_2$ $m^{-2}$ $yr^{-1}$ and $F_{CH4}$ was 3.9 g $C-CH_4$ $m^{-2}$ $yr^{-1}$ with the 95 % confidence interval between 3.75 and 4.07 g $C-CH_4$ $m^{-2}$ $yr^{-1}$. The ecosystem sequestered $CO_2$ during summer months (June-August), while the rest of the year it was a $CO_2$ source. $CH_4$ displayed strong seasonal dynamics, higher in summer and lower in winter, with a sporadic emission episode in the end of May 2013. Both $CH_4$ and $CO_2$ fluxes, especially those obtained from vegetated area, exhibited strong diurnal cycle during summer with synchronized peaks around noon. The annual $F_w\_CO_2$ was 297.5 g $C-CO_2$ $m^{-2}$ $yr^{-1}$ and $F_w\_CH_4$ was 1.73 g $C-CH_4$ $m^{-2}$ $yr^{-1}$. The peak diffusive $CH_4$ flux was 137.6 nmol $C-CH_4$ $m^{-2}$ $s^{-1}$, which was synchronized with the $F_{CH4}$.

Overall, during the monitored time period, the established stormwater wetland had a climate warming effect with 0.263 kg $CO_2$-eq $m^{-2}$ $yr^{-1}$ of which 89 % was contributed by $CH_4$. The radiative forcing of the open-water exceeded the vegetation area (1.194 kg $CO_2$-eq $m^{-2}$ $yr^{-1}$ and 0.111 kg $CO_2$-eq $m^{-2}$ $yr^{-1}$, respectively), which implies that, when considering solely the climate impact of a created wetland over a 100-year horizon, it would be more beneficial to design and establish wetlands with large patches of emergent vegetation, and to limit the areas of open-water to the minimum necessitated by other desired ecosystem services.

# 1 Introduction

Wetlands provide many beneficial ecosystem services such as flood control and water quality mitigation, natural habitat for flora and fauna and recreational opportunities (Mitsch and Gosselink, 2015). Many wetlands have been drained globally for agriculture, forestry and other purposes including urbanization at the cost of losing wetland ecosystem services (Vasander et al., 2003). Migration from rural area to cities will increase in even greater number in the near future, and UN 2016 report has predicted that 75 % of the world population will be living in cities by 2030. There is an urgent need for more sustainable urbanism and one effective measure is to create functional and connected wetland networks in cities (Lucas et al., 2015;Mungasavalli and Viraraghavan, 2006).

Wetlands can take up carbon dioxide ($CO_2$) through emergent and submerged vegetation but they are also important sources of methane ($CH_4$), a greenhouse gas more potent than $CO_2$ when considered over a 100-year horizon (Stocker et al., 2014). The exchange of greenhouse gases (GHG) such as $CO_2$ and $CH_4$ between atmosphere and ecosystem have



direct influence on the atmospheric concentration of these gases, thus besides the ecosystem services that wetland provide,
the GHG budget of constructed wetlands should be accounted for according to international agreements such as the Kyoto
protocol.
Reports on boreal wetlands, such as peatlands, have shown that large carbon storage remains in the soil due to anaerobic
conditions limiting microbial decomposition, and thus offering a global cooling effect (Frolking et al., 2006). However,
in newly constructed urban wetlands on mineral soil the gas exchange may be very different from natural wetlands: 1)
the cooling effect of a wetland may be reduced or it becomes a source of carbon due to the early successional stage of the
wetland, 2) wetlands in close proximity to urban centers receive significant amount of nutrients and dissolved organic
carbon from runoff and 3) urban wetlands exhibit high spatial heterogeneity and hydrology where different processes of
the production and transportation of GHG are involved. At the areas with emergent vegetation, $CO_2$ is absorbed by
photosynthetic activity during daytime and growing season and is released through respirational processes. At open-water
surfaces, the net production of $CO_2$ is a result of photosynthesis by algae, cyanobacteria as well as submerged aquatic
plants, respiration of organic carbon and oxidation of $CH_4$ produced in the water. When the $CO_2$ concentration in the
water exceeds atmospheric equilibrium, the surface becomes a source of $CO_2$. $CH_4$ can be produced through anaerobic
metabolism in wetland soil and can be transported to the atmosphere by plant-mediated pathway through aerenchyma,
sediment ebullition and diffusive fluxes at water-atmosphere interface. In open water, the transport is dominated by
diffusion whereas in vegetated area the plant-mediated transport is most prominent.
Urban wetlands have received extensive attention globally and their societal and economical importance have been
evaluated (Salminen et al., 2013), whereas their climate impact is still largely overlooked except for only a few studies
(e.g. Morin et al., 2014a;Morin et al., 2014b). The only review of GHG emission in constructed wetlands for wastewater
treatment reported that the average $CO_2$ emission was 92.3 mg $CO_2$-C m$^{-2}$ h$^{-1}$ and that the $CH_4$ emission ranged from 1.6
to 27 mg $CH_4$-C m$^{-2}$ h$^{-1}$ from free water surface (Mander et al., 2014). All of the studies were based on static chamber
measurements during a short period so that the annual carbon balance of the ecosystem could not be assessed. In contrast
to static chamber measurements, eddy covariance (EC) method provides continuous measurements of GHG exchange at
ecosystem scale, presenting the net result of fluxes as exchange in different source area contributing simultaneously within
the footprint extent (Baldocchi, 2003). It is worth noticing that one of the assumptions of the EC method is surface
homogeneity, yet in many study sites the situation are far from ideal. The change of source area due to changes in wind
provides difficulties in estimating GHG emissions in spatially heterogeneous sites especially in short-term flux
measurements (Baldocchi et al., 2012). Therefore, for heterogeneous sites such as urban wetlands, accurate footprint
modelling and surface area map at high spatial resolution are important in identifying the source area, and a land-surface
specific analysis is vital to reveal the diel pattern, sink/source strength of the wetland.
The objective of this study is to investigate how $CO_2$ and $CH_4$ surface-atmosphere exchange vary with seasonality and
spatial heterogeneity and what the annual radiative forcing of these gases are in a constructed urban wetland near town
Nummela, Municipality of Vihti, Southern Finland. The studied Gateway wetland was designed and implemented to serve
the purposes of stormwater quality treatment, creating an urban park, as well as supporting biodiversity. Besides taking
advantage of ecosystem-scale EC measurements, we also parse the variability of gas exchange induced by surface
heterogeneity (open water and vegetated area) using diffusional flux modeling and footprint modelling overlapped on a
high-resolution surface map. To illustrate how the urban wetland functions as a source or a sink of GHG equivalents, we



calculate separately the sustained global warming potential (SGWP) of $CO_2$ and $CH_4$ over a hundred-year horizon in each
surface type.

## 2 Materials and Methods

### 2.1 Site description

Our study site is a created stormwater wetland Gateway, located by an eutrophicated Lake Enäjärvi in the District of
Nummela, Municipality of Vihti, Southern Finland (60.3272°N, 24.3369°E). Southern Finland experiences a climate with
a 30-year mean air temperature of 4.6 °C and an annual precipitation rate of 627 mm in the period of year 1981-2010
(Pirinen et al., 2012).
The wetland was constructed in 2010 at the mouth of a 550 hectare largely urbanized (35 % impervious) watershed of
Stream Kilsoi. It was excavated over six weeks in early winter 2010 on an abandoned agricultural field growing meadow
vegetation. All of the old drainage ditches were blocked as amphibian habitats, which also ensured only one inlet route
receiving water from Stream Kilsoi and one outlet route discharging water to the nearby Lake Enäjärvi. Lake Enäjärvi is
a eutrophicated lake. The internal phosphorus load from human activities and the run-off from its catchments have resulted
in regular cyanobacterial blooms and fish kills in the lake.
The wetland park has a total area of 7 hectares within which - during mean water flow conditions - a 0.5 hectare inundated
wetland is located. This stormwater treatment wetland consists of an inlet stilling pond, a meandering shallow water area
with three habitat islands, and an outlet pond. The average water depth in the ponds is 1.5 m; within emergent vegetation
patches water depth ranges between 0.3 and 0.5 m. There are also submerged macrophytes in the open water as the water
is shallow, thus in the paper we refer the "vegetated area" to the area with emergent vegetation and "open water" to the
area covered by water in the absence of emergent macrophytes. The outlet bottom dam sets low water level (WL) to 50.04
m above the Baltic Sea level (N60+ coordinate system). Herbaceous vegetation has been allowed to fully self-establish
after the construction of the wetland. Annual monitoring of vegetation carried out in summers 2010, 2011 and 2012
indicated rapid self-establishment of vegetation which was rich in taxa and dominated by native species (Wahlroos et al.,
2015). At frequently-inundated area (elevation levels of 50-50.35 m), vegetation was arranged in dense patches with
different dominating wetland plant species: *Typha latifolia* L., *Iris pseudacorus* L., *Carex* spp. or *Juncus effuses* L. At the
major less-frequently inundated area (elevation levels of 50.35-50.45m), the wet meadow species *Filipendula ulmaria* L.
(Maxim.), *Lysimachia vulgaris* L., and *Lythrum salicaria* L. with the three species co-existing at 1:1:1 ratio formed the
plant community. Drier areas (elevation levels of 50.45-50.60 m) were mostly colonized by dry meadow species such as
*Poa* spp. and *Calamagrostis* spp., including patches dominated by *Cirsium* species (Fig. S1).

### 2.2 Water and micrometeorological measurements

Water monitoring stations were set up at the inlet (60.3283° N, 24.3356° E) and at the outlet (60.3281° N, 24.3377° E) of
the wetland. During the 2012-2013 and 2013-2014 monitoring periods, water temperature as well as water turbidity,
oxygen concentration, conductivity and pH were measured at the inlet and outlet monitoring station with the YSI-6000
series multiparameter sonde (YSI Inc., Yellow Springs, OH, USA). Measurements were conducted continuously with 10-
minute interval. Water level at the outlet was measured continuously with a pressure gauge (STS sensor, Sensor Technik
Sirnach AG, Switzerland). At the outlet monitoring station, the concentration of dissolved carbon dioxide ($[CO_2]$) and
dissolved methane ($[CH_4]$) were measured with Contros HydroC™ $CO_2$ and HydroC™ $CH_4$ sensors (CONTROS Systems



& Solutions GmbH, Germany). In 2014, the same sensors were also installed at the inlet monitoring station to measure
[$CO_2$] and [$CH_4$]. Dissolved $CO_2$ and $CH_4$ molecules diffuse from water column into the detection chamber through a
thin-film composite membrane where the concentration of $CO_2$ and $CH_4$ is determined by means of IR absorption
spectrometry and Tunable Diode Laser Absorption Spectroscopy, respectively.
Local weather conditions were recorded with a Vaisala WXT weather transmitter (WXT520, Vaisala Oyj, Finland) at the
inlet monitoring station. Rainfall, wind speed and direction, temperature and relative humidity were recorded
continuously at 10-minute interval. Photosynthetic photon flux density (PPFD) was measured with a PQS1 PAR quantum
sensor (Kipp & Zonen, the Netherland). Due to instrument failure we obtained PPFD data only from 26 Jan to 7 April
and from 22 July to 29 Dec 2013. The gaps were filled with PPFD data from another meteorological station nearby (60°38'
N, 23°58' E) in Lettosuo, Finland. The prevailing wind directions were southwest and northeast, and the average of half-
hourly average wind speed was 1.13 m s$^{-1}$ from January to December 2013 with higher wind speed in winter than in
summer. The average daily air temperature was 5.9 ℃ with the minimum and maximum daily temperatures of -24.4 ℃
and 23.3 ℃ in year 2013. During the winter 2012-2013, there was ice coverage from the beginning of December 2012 to
the end of March 2013. In contrast, winter was mild and warm in 2014 and there was practically no snow cover during a
winter period (December 2013-March 2014).
**2.3 Greenhouse gas measurements by EC tower and gap-filling**
To understand the whole-ecosystem exchange of $CO_2$ and $CH_4$ in the wetland, a 2.9 m eddy covariance tower was
established in the autumn of 2012 on the southern side of the wetland. The operational period of the EC tower was the
entire calendar year of 2013 (from 1 January to 31 December 2013) and the peak growing season in 2014 (from 1 June to
31 August 2014). The EC set-up included a 3D-sonic anemometer (uSonic-3, Metek, Elmshorn, Germany) to measure
the three wind speed components and sonic temperature, a gas analyser (LI-7200, Li-Cor Inc., Lincoln, Nebraska, USA)
which measures $CO_2$ and $H_2O$ mixing ratio and a TDL gas analyser (TGA100A, Campbell Scientific Inc., USA ) to
measure $CH_4$ mixing ratio. Data from the analyzers were collected on a computer at the frequency of 10 Hz. The post-
processing of the EC flux data has been done with EddyUH post-processing software (Mammarella et al., 2016). The
fluxes were calculated as 30-min covariances between the vertical wind velocity and the gas mixing ratio using block
averaging. The raw data was despiked according to standard methods (Vickers and Mahrt, 1997). Coordinate rotations
were conducted by performing a two-step rotation to make the x-axis along the mean wind direction and the mean vertical
wind velocity zero within each 30-min block. The time lag between the anemometer and gas analyzer signals, resulting
from the transport through the inlet tube, were determined for each 30-min interval by maximizing the cross-correlation
function between vertical wind speed and the scalar ($CO_2$ and $CH_4$). The fluxes were corrected for high-frequency loss
due to the limited frequency response of the EC system and low-frequency loss due to the limited averaging time period
used for calculating the fluxes. Theoretically and experimentally determined co-spectral transfer functions at low and
high frequency were used in the correction (Mammarella et al., 2009).
After calculating the fluxes, data collected from periods when sonic anemometer showed sign of freezing (mean
temperature < 0.5 °C and standard deviation of temperature > 1.5 °C) were discarded. The data collected during weak
turbulence with friction velocity below 0.1 m s$^{-1}$ have been removed. The measurement points with flux stationarity
greater than 1 were omitted to ensure the quality of the co-variances. Fluxes were further filtered according to the wind
direction. Since the patchy forest to the southeast of the EC tower (from 100° to 200°) and the highway to the west (from
200° to 280°) could potentially lead to flow distortion and additional source of $CO_2$ and $CH_4$, only fluxes from 280° to



100° were accepted for further analysis. The percentage of 30-min fluxes excluded from this analysis was 72 % for $CO_2$
and 73 % for $CH_4$ in 2013, whereas in 2014 the percentage for data exclusion was 54 % for $CO_2$ and 68 % for $CH_4$.
We used an artificial neural network (ANN) technique to gap-fill half-hourly flux data using meteorological variables
(Moffat et al., 2007;Papale et al., 2006). Those variables included radiation, air temperature, water temperature, water
level, wind speed, relative humidity, time of the day, season, and dissolved $CO_2$ and $CH_4$ concentration in the water. We
tested the model performance with different ANN architectures, starting from the architecture with the most complexity,
then reduced the variables to find the simplest ANN architecture with good performance (more than 5 % loss in model
accuracy with additional variable reduction). For $CO_2$, water level and wind speed were found to have trivial contribution
to the ANN model thus they were removed from the model input, while for $CH_4$, only wind speed was removed for the
same reason. We found that dissolved gas concentration greatly improved the model prediction as they captured the
variation of diffusive fluxes from the water (Fig. S2). Ancillary meteorological variables in general had good data
coverage and short gaps (up to several hours) were gap-filled by linear interpolation. The only exception was dissolved
gas concentration, which had long measurement breakage in year 2013 (day of year 214-254). Fluxes were therefore gap-
filled with two separate ANNs, one with dissolved gas concentration and one without. During the above mentioned period
with long gaps, the ANN modeled without dissolved gas concentration were used to gap-fill.
Levenberg-Marquardt algorithm was used in the learning process of ANN. The optimized number of neurons in the hidden
layer were determined by training the network 100 times with varying number of neurons (from 3 to 15), and 10 neurons
was considered to be sufficient after evaluating the performance of the network using root-mean-square-error (RMSE)
(data not shown). The entire dataset was divided into three parts, 2/3 of the data was used to train the networks, 1/6 for
testing the networks and the remaining 1/6 was used for validating the networks. Since the training of the networks can
be biased towards periods with greater data coverage (e.g. daytime conditions), the environmental variables were first
divided into five natural clusters using a k-mean clustering algorithm in Matlab (MATLAB 2015a, The MathWorks, Inc.,
Natick, Massachusetts, United States), and then the data used for training, testing and validation was proportionally
extracted from each cluster. After each data extraction, the network was reinitialized for 10 times to avoid local minima
and the initialization with the lowest RMSE was selected and resulting network was saved. We repeated the whole process
of data extraction and initialization for 20 times, and we used the median of these 20 predictions to gap-fill the missing
flux values. The uncertainty of the ANN gap-filling procedure was presented using a 95 % confidence interval of the 20
ANN predictions.
The gap-filled net ecosystem exchange (NEE) can be further partitioned into two components gross ecosystem production
(GEP) and ecosystem respiration ($R_{eco}$) according to the following equation:
$$NEE = GEP + R_{eco,} \qquad (1)$$
where positive $R_{eco}$ represents a net carbon flux from the ecosystem to the atmosphere and negative GEP represents a net
carbon input from the atmosphere to the ecosystem. Thus the negative NEE indicates that the ecosystem is a carbon sink
and the positive NEE means the ecosystem is a carbon source. $R_{eco}$ was estimated using a model describing the temperature
dependence of $R_{eco}$
$$R_{eco} = R_0\, e^{\left[E\left(\frac{1}{T_0} - \frac{1}{T_{air}+T_1}\right)\right]} \qquad (2)$$
where $E = 346.37$ K is an activation-energy-related physiological parameter, $T_{air}$ is the air temperature, $T_0 = 56.02$ K and
$T_1 = 227.13$ K (Lloyd and Taylor, 1994;Aurela et al., 2009). $R_0$ is the rate of ecosystem respiration at 10 °C. We first fitted





the model with nighttime NEE (which represents the nighttime ecosystem respiration since photosynthesis is assumed to
be zero at night) and determined $E$. We then calculated $R_0$ for each of the bi-weekly periods (Aurela et al., 2009). This
model was then extrapolated to daytime periods so that $R_{eco}$ in the daytime was obtained. GEP was estimated as the
difference between NEE and $R_{eco}$.

**2.4 Diffusive gas exchange**

We calculated diffusive gas exchange $F$ from open water according to the boundary layer model
$$F = k\left(c_{aq} - c_{eq}\right),\tag{3}$$
where $k$ is the gas transfer velocity (cm h$^{-1}$), $c_{aq}$ is the gas concentration in surface water (mol m$^{-3}$) and $c_{eq}$ is the gas
concentration that surface water would have when it reaches equilibrium with the air (mol m$^{-3}$). $c_{aq}$ and $c_{eq}$ can be obtained
according to the solubility of the gas
$$c_{aq} = 10^{-3}\,k_H\,p\chi_{water}\tag{4}$$
$$c_{eq} = 10^{-3}\,k_H p\chi_{air}\tag{5}$$
where $k_H$ is Henry's law constant for the respective gas (mol L$^{-1}$ atm$^{-1}$), $p$ is air pressure (atm), $\chi_{water}$ is the gas mixing ratio
in surface water (ppm) and $\chi_{air}$ is the gas mixing ratio in the air (ppm). In this study, $\chi_{water}$ was obtained from outlet
monitoring station as it was located most time in the flux footprint area and it had longer data coverage than from inlet
monitoring station. The gas transfer velocity $k$ can be calculated as the formula below (Cole and Caraco, 1998):
$$k = \left(2.07 + 0.215\,U_{10}^{1.7}\right)\left(\frac{S_c}{600}\right)^{-0.5},\tag{6}$$
where $U_{10}$ is the horizontal wind speed extrapolated to 10 m using the theoretical log wind profile equation (m s$^{-1}$,
approximately $U_{10} = 1.15\,U$ where $U$ is the measured wind speed at 2.9 m height in the study site) and $S_c$ is the
temperature-dependent Schmidt number of the respective gas. When gas concentration measurement was not available,
linear interpolation was applied to obtain monthly and annual diffusive GHG fluxes from the open water.
Although the above-mentioned Cole-Caraco (CC) method is the most simple and most often used model for gas transfer
velocity, the limitation of CC method is that it considers wind as the sole factor to cause the water turbulence and to drive
the gas exchange. More complicated models were suggested to include the effect of buoyancy flux driven turbulence
(Heiskanen et al., 2014;Tedford et al., 2014). It is important to note that we should apply with caution the model
parameterization concluded from other sites with different meteorological and environmental condition. In the present
study, the open water is connected shallow open-water pools with a maximum depth of 2 m while other studies are for
deeper waters. Meanwhile, recent study showed good agreement between the diffusive fluxes calculated using CC
methond and measurements based on floating chamber (Cole et al., 2010).

**2.5 Estimating zone fluxes and radiative forcing**

By combining EC tower and diffusive flux from the open-water, the following model can be derived
$$F_{EC} = F_{water} \times f_{water} + F_{water} \times f_{water}\tag{7}$$



where $F_{EC}$ is the flux measured by EC tower, $F_{water}$ and $F_{veg}$ stands for the flux from open-water and vegetated area,
respectively, $f_{water}$ and $f_{veg}$ are the footprint-weighted spatial fraction of open-water and vegetated area. In this study,
ebullition was neither measured nor calculated, so the flux from water was only represented by the diffusive flux.
Specifically, we first modelled the half-hourly flux footprint with a parameterization of a three-dimensional backward
Lagrangian footprint model (Kljun et al., 2015) in Matlab (MATLAB 2015a, The MathWorks, Inc., Natick, Massachusetts,
United States). Periods in which the wind came from the patchy forest to the southeast of the EC tower (between 100°
and 200°) and the highway to the west (between 200° and 280°) were eliminated in the footprint analysis. Secondly, a
land cover classification map of vegetated and open-water zones was delineated manually using a high-resolution aerial
image acquired from National Land Survey of Finland during the growing season of 2013 (open source:
https://www.maanmittauslaitos.fi/) with an image manipulation software (Gimp 2.10.6). Thirdly, the flux footprints were
aligned and combined with the land cover classification map to calculate half-hourly $f_{veg}$ and $f_{water}$ within 90 % footprint
contour lines. Specifically, we assigned each footprint pixel within the 90 % footprint area to either open-water or
vegetated area on the land cover classification map while the footprint of the pixels outside 90 % footprint area were
regarded as zero. $f_{water}$ was calculated as the sum of footprint within open-water area to the total footprints while $f_{veg}$ was
calculated as the sum of footprint within vegetated area to the total footprints. Noting that none of the 90 % footprint
contour lines exceeded the map area, the sum of $f_{veg}$ and $f_{water}$ equaled to 1. In order to obtain the long-term aggregated
footprint of carbon fluxes, we calculated also the monthly and annual aggregated footprint climatology during the study
period.
To better understand the influence of greenhouse gas fluxes in this urban wetland, we calculated the sustained global
warming potential (SGWP) for $CO_2$ and $CH_4$ over a hundred-year horizon in each surface type. The difference between
SGWP and global warming potential (GWP) is that SGWP accounts for the effect of GHG remains in the atmosphere
during the period. Since $CH_4$ is a more potent greenhouse gas, we multiply the emission of $CH_4$ by a factor of 45 to
convert it to kg $CO_2$-eq m$^{-2}$ yr$^{-1}$ (Neubauer and Megonigal, 2015).
**2.6 Statistical analysis**
The Pearson correlations ($r$) were determined between fluxes and environmental variables. Differences in the fluxes and
environmental variables between the two peak growing seasons (summer 2013 and 2014) were evaluated using the $t$-test.
Cumulative annual GHG fluxes measured by EC tower are reported as the median of the 20 ANN predictions and
uncertainty are presented as 95 % confidence interval of the 20 ANN predictions. As diffusive GHG fluxes were
calculated from gas concetration meteorological parameters, no standard error is reported for the cumulative annual fluxes
from the open water. All statistical analysis were performed in Matlab (MATLAB 2015a, The MathWorks, Inc., Natick,
Massachusetts, United States).
**3 Results**
**3.1 Ecosystem seasonality and environmental variables**
Daily average PPFD ranged from 0.9 to 691.5 µmol m$^{-2}$ s$^{-1}$ in year 2013 with the highest value appeared in July. June had
the highest monthly average PPFD with 486.1 µmol m$^{-2}$ s$^{-1}$ followed by July and August with 470.2 and 430.6 µmol m$^{-2}$
s$^{-1}$, respectively. The PPFD during the peak growing season in year 2014 was on average 361.8 µmol m$^{-2}$ s$^{-1}$, lower than
that during the same period in 2013 (Fig. 1a).



Mean daily water temperature ($T_{water}$) ranged from 0 °C in March to 23.7 °C in June with an annual average of 7.9 °C in
2013 and from 0 °C in February to 21.4°C in July in 2014. Mean daily air temperature ($T_{air}$) had more fluctuation and
ranged from -15.6 °C in January to 23.3 °C in June 2013 and from -19.0 °C in January to 23.4 °C in July 2014 (Fig. 1b).
The open-water area experienced an ice-covered period between 1 January and 31 March 2013, while the winter 2013-
2014 was so mild and warm that there was practically no snow cover during December 2013 – March 2014. Comparing
the temperature between the two peak growing seasons, both $T_{water}$ and $T_{air}$ were higher in June 2013 while $T_{air}$ was lower
in July 2013 than in 2014. In August, there was no significant temperature difference between the two years. Four seasons
were classified for the ecosystem based on the trend in $T_{air}$ and $T_{water}$. In spring (April and May), the daily temperature
started to increase, the vegetation showed a sign of early growing season and the warm temperature unfroze the lake. In
summer, the peak growing season (June - August), vegetation exhibited the maximum growth which was reflected in the
large negative GEP value, and the temperatures reached the annual maxima. In autumn (September and October), daily
temperatures began to drop and the vegetation showed signs of early senescence. In winter (January to March and
November, December), temperatures reached the annual minima and vegetation was inactive in carbon sequestration.
Precipitation was higher in August 2014 than in the preceding August, almost twice as high as that of 2013.
WL was higher in the winter and lower in the summer in 2013. The daily average of WL varied between 50.06 m in July
2013 and 50.4 m in April 2013. There was a spring peak in 2013 when the highest WL was observed due to snow melt
while in 2014 no such event appeared due to the mild winter 2013-2014 without ice-covered period (Fig. 1c). The average
daily WL from January to August was similar (50.13 cm and 50.15 cm for 2013 and 2014, respectively). However, during
peak growing season, it was on average 5.7 cm higher in 2014 than in 2013.
The annual rainfall in 2013 (snowfall not included) was 363.6 mm which happened mostly during summer and autumn
(Fig. 1d). The maximum daily-averaged rainfall was in August (26.7 mm day$^{-1}$) while monthly-averaged rainfall was
highest in November with 73.8 mm month$^{-1}$ followed by August with 68.3 mm month$^{-1}$. In 2014, exceptionally high
amount of rainfall was observed in August (125.7 mm month$^{-1}$), while the amount of rainfall in the other months were
similar to 2013.
The daily-averaged $CO_2$ concentration in the water ($[CO_2]$) in 2013 had large variation with the maximum (9324 ppm)
and the minimum (353 ppm) both happening in October (Fig. 1e). $[CO_2]$ was higher in summer months (5457 ppm) and
lower in winter months (3345 ppm). $[CO_2]$ was higher in 2014 with an average of 4924 ppm from January to August than
in 2013 with an average of 3781 ppm. It also exhibited seasonal variation with high concentration in summer (8084 ppm)
and low concentration in winter (3513 ppm). The $[CO_2]$ measured in the inflow was generally lower than that in the
outflow and they were well correlated ($r$=0.84). $[CH_4]$ in the outflow was on average five times higher in 2014 than in
2013. The average annual concentration was 0.81 μmol L$^{-1}$ in year 2013 and 2.25 μmol L$^{-1}$ in 2014. There were peak
$[CH_4]$ episodes in the outflow in May 2013 with a maximum of 5.43 μmol L$^{-1}$. During the summer months in 2014 there
were even higher outflow $[CH_4]$ peaks with a maximum of 16.83 μmol L$^{-1}$. The $[CH_4]$ had a mean of 0.42 μmol L$^{-1}$ in the
inflow which was lower than that in the outflow, and there was no prominent $[CH_4]$ peaks observed in the inflow. $[CH_4]$
in the inflow and outflow were weakly correlated ($r = 0.2$) (Fig. 1f).
**3.2 Flux footprint mapping**
A footprint distribution was modeled for each half hour when an eddy flux measurement was collected at the EC tower.
The open-water area accounted for 10 % to 16 % of the total wetland area within the footprint while the rest was comprised





of wetland vegetation. When weighted with footprint distribution, $f_{water}$ ranged from 0 to 25.5 % and $f_{veg}$ from 74.5 % to
100 %. The 1$^{st}$ quantile, median and 3$^{rd}$ quantile of $f_{water}$ and $f_{veg}$ were 0.09 %, 14.1 %, 17.9 % and 82.1 %, 85.9 %, 91.3
%, respectively.
The monthly cumulative footprint was slightly different for $CO_2$ and for $CH_4$ due to the different missing flux values.
However, the difference on average was so small (7 %) and the footprint of $CO_2$ was used in further analysis. The flux
footprints were shown to be northeast to the EC mast due to the wind direction filtering meaning only half-hourly data
with wind directions from the wetland area were considered in the analysis (Fig. S3). The monthly-average of the 90 %
footprint area covered a minimum of 0.69 ha to a maximum of 2.28 ha with a mean of 1.3 ha. The mean extent of the 90
% flux footprints was 128 m. After applying flux footprint function, the monthly-average of the footprint-weighted spatial
fraction of open water showed lower value in summer and higher value in winter ranging from 11.3 % to 21.4 % with a
mean of 13.3 % in 2013. In 2014 during the peak growing season, on average 13.8 % of the wetland area was comprised
of open water and the mean $f_{water}$ was 10 %.

**3.3 $CO_2$ and $CH_4$ fluxes**

**3.3.1 Ecosystem $CO_2$ and $CH_4$ fluxes**

Ecosystem $CO_2$ and $CH_4$ fluxes measured by EC tower showed the ecosystem was nearly $CO_2$ neutral and it was a small
$CH_4$ source in year 2013.
Daily average of NEE was near zero during winter time (January to March, on average 0.37 µmol C-$CO_2$ m$^{-2}$ s$^{-1}$), slightly
positive in spring and it became negative from the end of May till the end of August indicating the ecosystem was a $CO_2$
sink during this period, with a maximum negative value of -5.14 µmol C-$CO_2$ m$^{-2}$ s$^{-1}$ in June. Daily-average NEE was
highest in September with a maximum of 3.29 µmol C-$CO_2$ m$^{-2}$ s$^{-1}$, possibly due to the suppressed GEP and high $R_{eco}$. In
October, November and December, NEE remained low but still positive (on average 0.77 g µmol C-$CO_2$ m$^{-2}$ s$^{-1}$),
demonstrating the milder winter between 2013 and 2014 (Fig. 2 and Fig. S4). NEE, GEP and $R_{eco}$ exhibited strong
seasonality in 2013 NEE was negative during June, July and August meaning the ecosystem was a $CO_2$ sink while the
rest of year it was a $CO_2$ source. NEE was lowest in June and highest in September. Both GEP and $R_{eco}$ achieved their
highest values in July (Fig. S4). The cumulative NEE in 2013 was 8 g C-$CO_2$ m$^{-2}$ yr$^{-1}$ with the 95% confidence interval
between -18.9 and 34.9 g C-$CO_2$ m$^{-2}$ yr$^{-1}$ (Fig. 2).
Daily-averaged $CH_4$ was low but not negligible from January to April (on average 5.1 nmol C-$CH_4$ m$^{-2}$ s$^{-1}$), with a sudden
rise in the end of May reaching a maximum of 48.9 nmol C-$CH_4$ m$^{-2}$ s$^{-1}$. During summer months the ecosystem exhibited
relatively high $CH_4$ emission (on average 15.4 nmol C-$CH_4$ m$^{-2}$ s$^{-1}$), not comparable with the emission episode in May
but higher than winter months. In autume (September and October) the daily-average CH4 was 8.8 nmol C-$CH_4$ m$^{-2}$ s$^{-1}$
and after that it gradually decreased throughout the rest of the year with an average of 5.5 nmol C-$CH_4$ m$^{-2}$ s$^{-1}$. The
cumulative $CH_4$ for 2013 was 3.9 g C-$CH_4$ m$^{-2}$ yr$^{-1}$ with the 95% confidence interval between 3.75 and 4.07 g C-$CH_4$ m$^{-2}$
yr$^{-1}$ (Fig. 3).
Comparing the peak growing season between 2013 and 2014, the 30 mins NEE ranged from -20.0 µmol C-$CO_2$ m$^{-2}$ s$^{-1}$ in
June to 18.5 µmol C-$CO_2$ m$^{-2}$ s$^{-1}$ in September 2013. GEP reached maximum negative value in July 2013 with -30.5 µmol
C-$CO_2$ m$^{-2}$ s$^{-1}$ and $R_{eco}$ in June with 13.9 µmol C-$CO_2$ m$^{-2}$ s$^{-1}$. During the peak growing season 2014, NEE had lowest
value -22.6 µmol C-$CO_2$ m$^{-2}$ s$^{-1}$ in June, GEP -28.6 µmol C-$CO_2$ m$^{-2}$ s$^{-1}$ and $R_{eco}$ had its maximum in the beginning of
August 2014 with 11.3 µmol C-$CO_2$ m$^{-2}$ s$^{-1}$. The monthly NEEs of peak growing season were -84.1, -76,1 and -22.2 g C-



$CO_2$ m$^{-2}$ month$^{-1}$ in June, July and August 2013, and -97.6, -47,5 and -19.6 g C-$CO_2$ m$^{-2}$ month$^{-1}$ in 2014. In both years,
daily-averaged GEP had its maximum negative value in July (-13.4 and -12.8 g C-$CO_2$ m$^{-2}$ d$^{-1}$). Daily-averaged $R_{eco}$ was
highest in June 2013 with 12.1 g C-$CO_2$ m$^{-2}$ d$^{-1}$ while in 2014 $R_{eco}$ was low in June and the peak was in the end of July
with 10.5 g C-$CO_2$ m$^{-2}$ d$^{-1}$ (Fig. S5a). The average $CH_4$ emission in June, July and August were 24.4, 10.8 and 11 nmol
m$^{-2}$ s$^{-1}$ in 2013, and 15.5, 21.3 and 21.3 nmol m$^{-2}$ s$^{-1}$ in 2014, respectively (Fig. S5b).
**3.3.2 Diffusive $CO_2$ and $CH_4$ fluxes from open-water area**
Diffusive $CO_2$ and $CH_4$ fluxes from the open water were estimated based on wind speed, [$CO_2$] and [$CH_4$] (See Sect. 2.4).
The variation of diffusive fluxes demonstrated a pattern driven by both wind speed in short term and gas concentration
dynamics in the water in long term. Diffusive $CO_2$ fluxes ranged from -0.07 to 4.09 µmol $CO_2$ m$^{-2}$ s$^{-1}$ with a mean of 1.04
µmol $CO_2$ m$^{-2}$ s$^{-1}$ in 2013 indicating $CO_2$ oversaturation in the water. From June to September the averaged flux (1.27
µmol $CO_2$ m$^{-2}$ s$^{-1}$) was higher than that of the other months (Fig. 4a), corresponding to the higher [$CO_2$] in the water
during summer months (Fig. 1d). The monthly-averaged diffusive $CO_2$ flux during peak growing season in 2014 was
2.34, 2.71 and 1.99 µmol $CO_2$ m$^{-2}$ s$^{-1}$ for June, July and August, significantly higher than during the same period in 2013
due to the high [$CO_2$] in the open water (Fig. 1d).
The average diffusive $CH_4$ emissions in 2013 was 4.9 nmol C-$CH_4$ m$^{-2}$ s$^{-1}$, where a peak emission appeared in late May
with the highest flux of 137.6 nmol C-$CH_4$ m$^{-2}$ s$^{-1}$. Monthly-averaged $CH_4$ diffusive fluxes showed an increasing trend
towards the end of the year with large variation in May due to the peak concentration episode. This phenomenon was
mainly driven by the increasing dissolved $CH_4$ concentration in the outflow in 2013. The monthly-averaged diffusive $CH_4$
flux during peak growing season in 2014 was 20.9, 18.9 and 13.5 nmol $CH_4$ m$^{-2}$ s$^{-1}$ for June, July and August, respectively
and they were significantly higher than the same period in 2013 due to the high [$CH_4$] in the open water (Fig. 1e).
**3.3.3 Diel patterns in $CO_2$ and $CH_4$ fluxes**
Only non-gapfilled data were used for determination of diel patterns in both gas fluxes. $CO_2$ and $CH_4$ fluxes from vegetated
area ($F_{veg}$) was calculated for each 30-min interval according to formula (5). As expected, $CO_2$ flux showed strong diel
pattern in summer with $CO_2$ uptake during daytime and release in the night, which was controlled by photosynthetic
activity (Fig. 5a). The summer peak $CO_2$ uptake reached 11.5 µmol m$^{-2}$ s$^{-1}$ for the whole constructed wetland ecosystem
and 15.2 µmol m$^{-2}$ s$^{-1}$ for the vegetated area. The $CO_2$ flux from the vegetated area had higher maximum uptake than the
EC measurements carried out over the whole constructed wetland. In the winter, the $CO_2$ fluxes from both tower and
vegetation were similar, being on average 0.46 and 0.55 µmol m$^{-2}$ s$^{-1}$ respectively (Fig. 5b).
$CH_4$ flux also showed diel patterns in the summer with much larger variability than those from $CO_2$ flux. $CH_4$ emission in
general was higher in daytime than in nighttime. In the daytime in summer, $CH_4$ flux from the vegetated area was higher
than the flux measured from the tower while there was no difference during the nighttime (Fig. 5c, 5d). The summer peak
daytime flux from tower (18.9 nmol m$^{-2}$ s$^{-1}$) and vegetated area (24.7 nmol m$^{-2}$ s$^{-1}$) was 2.4 times and 3.3 times higher
than the nighttime flux (7.5 nmol m$^{-2}$ s$^{-1}$), respectively. This can be understood as daytime $CH_4$ flux is linked with
photosynthesis while nighttime $CH_4$ flux is controlled by other processes like diffusion, ebullition and convection between
the soil, water and atmosphere. In winter there was small (on average 4.6 nmol m$^{-2}$ s$^{-1}$) but constantly positive $CH_4$ flux
without obvious diel pattern.
**3.4 Environmental variables with fluxes**





Only non-gapfilled flux data were used in the Pearson correlation analysis between environmental variables and flux
pairs. Radiation, $T_{air}$ and $T_{water}$ all had high negative correlation coefficient ($r$) with NEE and high positive $r$ with $CH_4$
flux in 2013, corresponding to the results of ANN model parameter selection. Radiation was best correlated with NEE
and $T_{water}$ was best correlated with $CH_4$ (Table 1). The correlations were rather weak (small $r$ or even the opposite sign of
$r$) during 2014 due to the short measuring period and narrow ranges of the variables. Water level was positively correlated
with NEE and negatively correlated with $CH_4$, which was counter intuition, possibly because it was masked by
temperature variation as the water level was in general higher in winter and lower in summer. [$CO_2$] and [$CH_4$] were not
correlated with either NEE or $CH_4$ although they were shown to be important parameters in ANN model selection.

**3.5 Estimating radiative forcing from different zones**

To obtain the climate forcings from each land surface type, we calculated the half-hourly and annual gas fluxes from the
vegetated area based on eq. (7) using footprint-weighted spatial fraction, ecosystem fluxes and diffusive fluxes from the
open water (See Sect. 2.5). The annual median value of footprint-weighted spatial extent was used to calculate the annual
fluxes, which showed open-water area was a $CO_2$ source (297.5 g $C\text{-}CO_2$ m$^{-2}$ yr$^{-1}$) and vegetated area was a $CO_2$ sink (-
39.5 $C\text{-}CO_2$ m$^{-2}$ yr$^{-1}$). Both open-water and vegetated area were $CH_4$ sources but the $CH_4$ emission from vegetated area
was higher than open-water area, being 4.26 and 1.73 g $C\text{-}CO_2$ m$^{-2}$ yr$^{-1}$, respectively (Table 2).
Open water has contributed large amount of $CO_2$ emission into the atmosphere through diffusion (1.09 kg $CO_2$-eq m$^{-2}$ yr$^{-1}$
) whereas the $CH_4$ emission was relatively small (0.104 kg $CO_2$-eq m$^{-2}$ yr$^{-1}$). Vegetated area was a small sink of $CO_2$ but
the cooling effect of vegetation by $CO_2$ uptake was relatively small (-0.145 kg $CO_2$-eq m$^{-2}$ yr$^{-1}$) compared to its $CH_4$
emission (0.256 kg $CO_2$-eq m$^{-2}$ yr$^{-1}$). Overall, the ecosystem had a small warming effect with 0. 263 kg $CO_2$-eq m$^{-2}$ yr$^{-1}$
of which 89% was contributed by $CH_4$ (Table 2).

**4 Discussion**

**4.1 The GHG fluxes from an urban stormwater wetland ecosystem**

The studied urban wetland ecosystem was a small carbon source over the full-year studied period in year 2013. Due to
the scarcity of studies on urban wetlands using the EC method, we compare our results to restored wetlands which can be
considered to be proxy ecosystems to urban wetlands with both including rewetting practice in an ecosystem which has
been drained previously. The annual $CO_2$ balance of 8 g $C\text{-}CO_2$ m$^{-2}$ yr$^{-1}$ from the ecosystem, or -39.5 g $C\text{-}CO_2$ m$^{-2}$ yr$^{-1}$
from the vegetated area (Table 2), were small compared to a restored wetland in western Denmark where the annual $CO_2$
balance ranged from -286 to -53 g $C\text{-}CO_2$ m$^{-2}$ yr$^{-1}$ (Herbst et al., 2013), and the annual $CH_4$ balance of 3.9 g $C\text{-}CH_4$ m$^{-2}$
yr$^{-1}$ was less than half of the annual $CH_4$ emission (between 9 and 13 g $C\text{-}CH_4$ m$^{-2}$ yr$^{-1}$) in that study. Over a network of
restored freshwater wetlands in the California, the $CO_2$ sequestration can be up to nearly 700 g C m$^{-2}$ yr$^{-1}$ and $CH_4$ emission
up to 63 g C m$^{-2}$ yr$^{-1}$ (Hemes et al., 2018). It is not surprising that the studied ecosystem appeared to $CO_2$ neutral as it was
recently constructed. The herbaceous vegetation has been allowed to fully self-establish without human intervention and
at the early successional stage, plant diversity and biomass were still increasing each year (Wahlroos, 2019). With the
vegetation being more developed, a greater $CO_2$ uptake from the vegetated area can be expected in the following years.
The low $CH_4$ emission observed in this study may be due to the depletion of organic matters in the bottom soil from
agricultural uses thus it provided little substrate for anaerobic microbial activity to produce $CH_4$. With the accumulation
of organic matters in the anoxic wetland sediment, $CH_4$ production may increase in the future. Certain chemical





compounds like Fe in mineral soils can also inhibit $CH_4$ production leading to much lower ecosystem-scale $CH_4$ flux
(Chamberlain et al., 2018). In the meanwhile, methane-oxidizing bacteria (methanotroph) regulates $CH_4$ consumption at
the soil-water interface. With the ecosystem being used previously as cropland, the physical disturbance of soil may have
greatly reduced the methanotroph communities so that the $CH_4$ oxidation may also be low in the soil (Smith et al.,
2000;Saggar et al., 2008). Furthermore, after the initial establishing phase, the ecosystem productivity can also be reduced
due to the standing litter that inhibits the generation of new vegetation growth. It was shown that in a restored freshwater
wetland the ecosystem was a net $CO_2$ sink (-804 ± 131 g C-$CO_2$ $m^{-2}$ $yr^{-1}$) in 2002-2003, six years after the restoration but
near $CO_2$ neutral in 2010-2011 due to the reduced photosynthetic plants (Anderson et al., 2016). Thus, given the urban
wetland is sustained for a sufficiently long period, it is still unclear whether the $CO_2$ uptake from vegetated zone would
compensate its $CH_4$ emission, not considering the large GHG emission from the open-water zone. Thus, similar studies
as the present one should be conducted at a later stage after the construction of the wetland to fully reveal the GHG
balance of the ecosystem along time.
Overall, the ecosystem $CO_2$ and $CH_4$ fluxes measured by EC tower ranged from -5.33 to 3.4 g C-$CO_2$ $m^{-2}$ $day^{-1}$ and from
1.0 to 55.2 mg C-$CH_4$ $m^{-2}$ $day^{-1}$, respectively. They are consistent with the flux ranges provided by other studies on GHG
fluxes in restored wetlands (Anderson et al., 2016;Knox et al., 2015;Matthes et al., 2014;Morin et al., 2014b;Herbst et al.,
2013), although for both gases they tend to be on the lower end. NEE, GEP and $R_{eco}$ exhibited seasonal variation so that
the ecosystem was a $CO_2$ sink between June and August. The highest NEE appeared in September possibly because GEP
has greatly reduced due to plant senescent while $R_{eco}$ remained relatively high because of the warm temperature (Fig. S4).
Previous studies have found good agreement between $CH_4$ emission and GEP as plants provide substrates for
methanogenesis (Rinne et al., 2018), which was not observed in the daily-average of gas fluxes in this study (Figs 2 and
3) as the peak $CH_4$ flux appeared in May and peak GEP appeared in July. Nonetheless, both $CH_4$ and $CO_2$ fluxes, especially
those obtained from vegetated area, exhibited strong diurnal cycle during summer with synchronized peaks around noon
(Fig. 5a, 5c). This finding reflects that short-term $CH_4$ emission from vegetation is linked with photosynthesis by
providing labile carbon from root exudate and by gas transport through aerenchyma and open stomata while long-term
$CH_4$ emission may be determined by complex processes related to environmental variables e.g. temperature and redox
potential (Linden et al., 2014).

**4.2 Parsing GHG fluxes from heterogeneous land surfaces**

We found the open-water area was constantly a source of $CO_2$ and $CH_4$ to the atmosphere during the studied period as the
[$CO_2$] and [$CH_4$] in the water generally exceeded the atmosphere equilibrium except the ice-covered period (Fig. 4). The
annual average of [$CO_2$] in the surface water in 2013 was 0.3% in our study, comparable to 0.4% in another temperate
restored wetland (McNicol et al., 2017), while the seasonal pattern (higher in summer and fall) was the opposite as they
have found. We also found that both [$CO_2$] and [$CH_4$] were higher in 2014 than 2013 (Fig. 1d, 1e). The $O_2$ concentration
([$O_2$]) and $O_2$ balance ([$O_2$]$_{outlet}$ - [$O_2$]$_{inlet}$) measured by another study on the same wetland (Wahlroos, 2019) could partially
explain the observed phenomenon. The relatively high water temperature and oxic conditions in the water in fall 2013
have allowed high decomposition of detritus leading to high [$CO_2$] (Wahlroos, 2019). The long period of hypoxia during
summer 2014 could explain the three-fold increase in [$CH_4$] as the condition was more favorable for $CH_4$ production. The
negative $O_2$ balance in summer 2014 indicated strong $O_2$ consumption by microbial decomposition producing $CO_2$ in the
water. As the long-term diffusive fluxes (daily and monthly) was mainly driven by gas concentration in the water, it was
straight forward to understand high diffusive $CO_2$ and $CH_4$ fluxes in 2014 comparing to 2013. Interestingly, the ecosystem



CH$_4$ emission in 2013 was well synchronized with the diffusive CH$_4$ flux by capturing sporadic emission episodes from
the water (Fig. S6a, S6c) while they were not synchronized in summer 2014 although several stronger diffusive peaks
happened (Fig. S6b, S6d). When footprint-weighted contribution was accounted for, it clearly revealed that the
synchronization of CH$_4$ emission from ecosystem and water was closely related to the flux footprint distribution. When
there was high flux contribution from the open water (20-25 %), high diffusive CH$_4$ was also reflected in ecosystem flux
measured by EC. This has further proved the application of footprint analysis is essential in explaining gas exchange from
heterogeneous surfaces using EC data.
It is worth noticing that in our study we only classified the surface landscapes into "open water" and "vegetation" but
neglected the difference in sink/source strength from different plant types within the vegetation zone (Fig. S1). We did
not account for the dissimilarity between vegetation types because the characteristics in gas exchange are much more
distinct between open water and vegetation, which was the focus of this study. For the same reason, ebullition was not
considered in this study neither, as ebullition was shown to have only minor significance in a restored wetland accounting
for less than 0.1% of ecosystem CO$_2$ flux and 4.1% of ecosystem CH$_4$ flux (McNicol et al., 2017). However, for a proper
downscaling analysis of EC data, the subareas of different plant types and ebullition should also be taken into account.
**4.3 Climate impact of urban wetland and implications for management**
In the present study, the urban boreal wetland had an overall SGWP of 0.263 kg CO$_2$-eq m$^{-2}$ yr$^{-1}$ which was comparable
or higher than other restored wetlands in boreal region (Herbst et al., 2013), and within the range of inter-annual variation
or lower than restored wetlands in temperate zone (McNicol et al., 2017;Anderson et al., 2016). Different from other
studies, the urban wetland was CO$_2$ neutral and a CH$_4$ source. It is worth noting that the paramount contribution of CH$_4$
in ecosystem SGWP was mainly driven by the large footprint-weighted spatial area of vegetation (See Sect. 3.2). In fact,
The SGWP of GHG emission from open water (1.194 kg CO$_2$-eq m$^{-2}$ yr$^{-1}$) was 10 times as large as that from vegetation
(0.111 kg CO$_2$-eq m$^{-2}$ yr$^{-1}$) (Table 2). The implication of this result is that during wetland restoration, it would be more
beneficial to have large patches of emergent vegetation area at least from the GHG emission point of view. Similar results
have been obtained by other studies as well that open water has more climate-warming impact than emergent vegetation
due to the large diffusive fluxes from open water (Stefanik and Mitsch, 2014;McNicol et al., 2017). The climate impact
of natural wetland depends on the net balance between the cooling effect of CO$_2$ uptake by vegetation and the warming
effect of other GHG emissions, mainly CH$_4$ (Bridgham et al., 2013). In wetlands constructed in urban area, the large
fraction of open water which is a significant emitter of CO$_2$, should also be taken into consideration when evaluating the
role of urban wetland in global climate change.
**5 Conclusions**
Urban wetlands have received global attention as a nature-based urban runoff management solution for sustainable cities,
as they provide cost efficient flood control and water quality mitigation as well as many ecological and cultural services.
In the meantime, the climate impact of urban wetlands should also be considered. Wetting a landscape may enhance the
CO$_2$ sequestration in the ecosystem, whereas CH$_4$ can be emitted due to the anaerobic conditions in the soil after wetting.
Furthermore, heterogeneity induced in newly created urban wetland may contribute differently to the overall climate
impact.



In the present study, for the first time a full annual carbon balance of an urban stormwater wetland in the boreal region
was evaluated and the radiative forcing from heterogeneous landscapes were presented. We found that, during the
monitored period at the study wetland, both the open water area and the vegetated area within the created wetland were
carbon sources, and thus the urban wetland had a net climate warming effect, the monitored fourth year after the wetland
establishment. The radiative forcing effect of the open-water area exceeded the vegetated area, which indicated that
limiting open-water surfaces and setting a design preference for areas of emergent vegetation in the establishment of
urban wetlands can be a beneficial practice when considering only the climate impact of a created urban wetland. In the
meanwhile, we also emphasize that the value of urban wetlands should not be determined solely by GHG radiative forcing.
The values of urban wetlands in other areas e.g. flood control, pollutant removal, biodiversity, recreation and education
are as well of paramount importance to human society.

**Data availability**
Eddy covariance, gas concentration and meteorological data are available from the DRYAD database at
https://datadryad.org/stash/share/WrtTNnpIt6FgLoMSZ_Wlr0lK22IcxqjGZAStuuKdHLs

**Author contribution**
IM, OW, HV, AO and TV designed the field study. SH, IM and JP carried out eddy covariance measurements, automatic
gas concentration measurements in the open water and manual field measurements. XL and IM participated in eddy
covariance data processing and analysis. XL analysed the results and prepared the manuscript with contributions from all
co-authors.

**Competing interests**
The authors declare that they have no conflict of interest.

**Acknowledgments**

We thank Mikko Yli-Rosti and Kiril Aspila for assistance for the maintenance of the field measurements. This research
was supported by the EU Life+11 ENV/FI/911 Urban Oases project grant, Academy of Finland, Academy Professor
projects (312571 and 282842), ICOS-Finland (281255), the Maa- ja vesitekniikan tuki ry, the Ministry of the Environment
of Finland and the Municipality of Vihti. In memoriam: The greenhouse gas exchange measurements at the Gateway
Wetland were made possible due to the creative and caring support by late Professor Eero Nikinmaa to the Urban Oases
project.

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





**Tables**

Table 1. Pearson correlation coefficient ($r$) between the daily averages of environmental variables and fluxes in year 2013 and 2014. NEE – net ecosystem exchange; $T_{air}$ – air temperature; $T_{water}$ – water temperature; PPFD – photosynthetic photon flux density; WL – water level; [$CO_2$] and [$CH_4$] – $CO_2$ and $CH_4$ concentration measured in the outlet; * indicates only peak growing season (June, July and August) are included in the analysis.

| Flux | Year | $T_{air}$ | $T_{water}$ | PPFD | WL | [$CO_2$] | [$CH_4$] |
|------|------|-----------|-------------|------|------|---------|---------|
| $CO_2$ | 2013 | -0.45 | -0.61 | -0.62 | 0.46 | -0.34 | 0.18 |
| | 2014 | 0.43 | 0.54 | -0.12 | 0.12 | -0.12 | -0.05 |
| $CH_4$ | 2013 | 0.61 | 0.65 | 0.56 | -0.3 | 0.17 | -0.09 |
| | 2014* | 0.37 | 0.26 | 0.27 | -0.24 | 0.28 | 0.25 |



Table 2. Annual $CO_2$ and $CH_4$ exchange from different surface zones and their sustained global warming potential
(SGWP). Ecosystem, water and vegetation represent flux and SGWP measured or calculated from the ecosystem by EC
tower, from open water and from vegetated area. The numbers in the square bracket represent the 95% confidence interval
of the average. No error bounds are reported for flux and SGWP from water as they are modelled using gas concentration
in the water and meteorological measurements.

| | | Ecosystem | Water | Vegetation |
|---|---|---|---|---|
| Flux (g C m$^{-2}$) | $CO_2$ | 8 [-18.9, 34.9] | 297.5 | -39.5 [-70.8, -8.1] |
| | $CH_4$ | 3.9 [3.8, 4.1] | 1.7 | 4.3 [4.1, 4.5] |
| SGWP (kg $CO_2$-eq m$^{-2}$) | $CO_2$ | 0.029 [-0.069, 0.128] | 1.090 | -0.145 [-0.260, -0.030] |
| | $CH_4$ | 0.234 [0.225, 0.244] | 0.104 | 0.256 [0.246, 0.268] |


**Biogeosciences** Open Access
Discussions
EGU

**Figures**

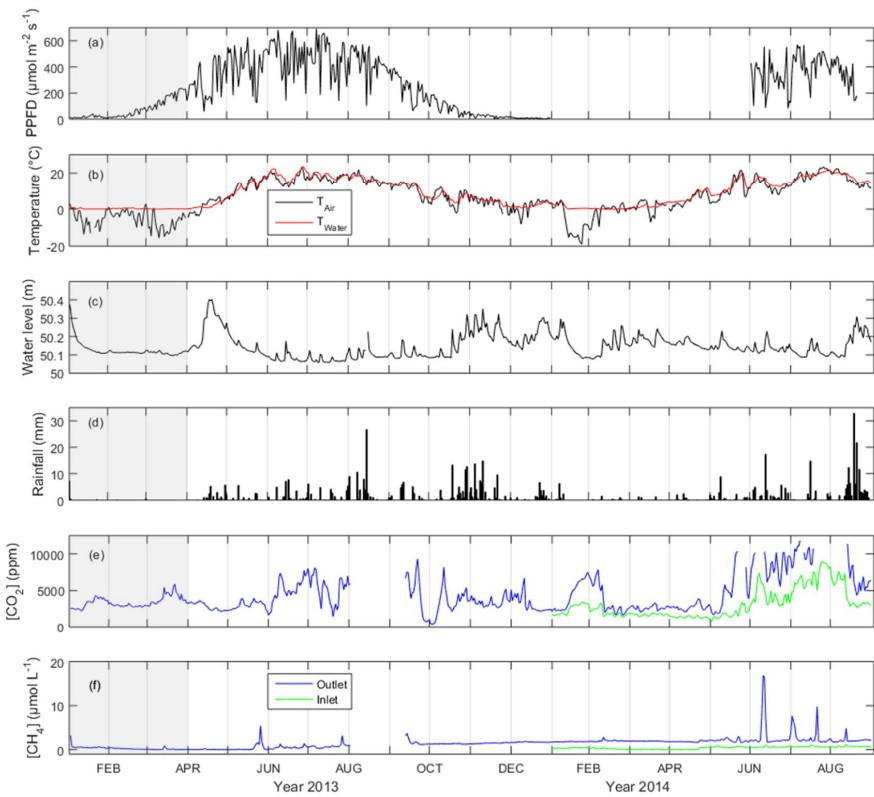


Figure 1: The daily-average of (a) photosynthetic photon flux density (PPFD), (b) air and water temperature, (c) water
level, (d) rainfall, (e) $CO_2$ concentration and (f) $CH_4$ concentration from inlet and outlet of Nummela wetland from
January 2013 to August 2014. The grey zone indicates the ice-covered period.



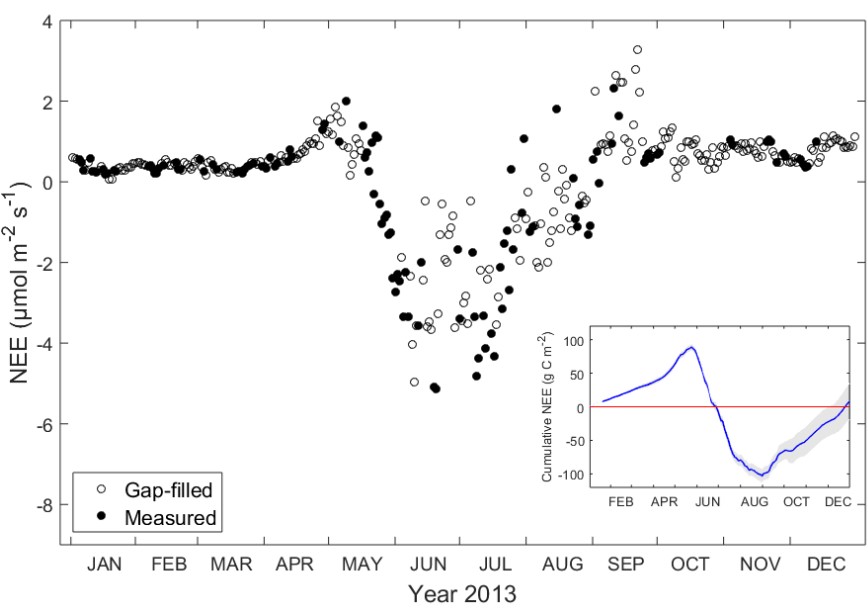


Figure 2: Daily average of net ecosystem exchange of $CO_2$ (NEE, μmol m$^{-2}$ s$^{-1}$) in year 2013. Filled dots indicate measurement (wen available half-hourly measurement data $\geq$ 10) and circles indicate gap-filled data (when available half-hourly measurement data < 10). The insert shows cumulative NEE (g C m$^{-2}$) in the ecosystem and the red line indicates the zero reference line. Error bounds (marked in grey) on cumulative NEE reflect the 95 % confidence interval for the gap-filling procedure.








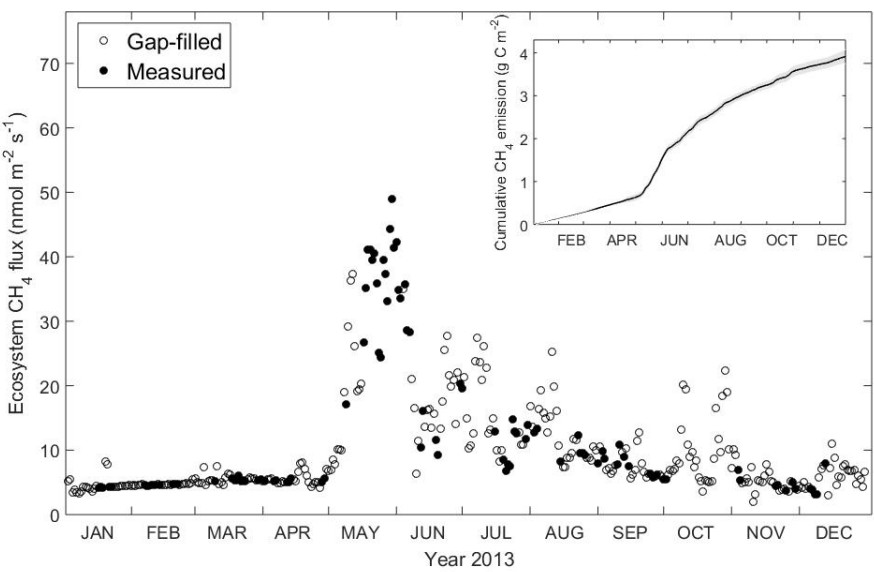


Figure 3: Daily average of ecosystem CH$_4$ flux measured by EC tower and cumulative CH$_4$ emission in year 2013. Filled

dots indicate measurement (when available half-hourly measurement data ≥ 10) and circles indicate gap-filled data (when

available half-hourly measurement data < 10). The insert shows cumulative CH$_4$ emission with the error bounds in grey

reflecting the 95 % confidence interval for the gap-filling procedure.



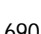

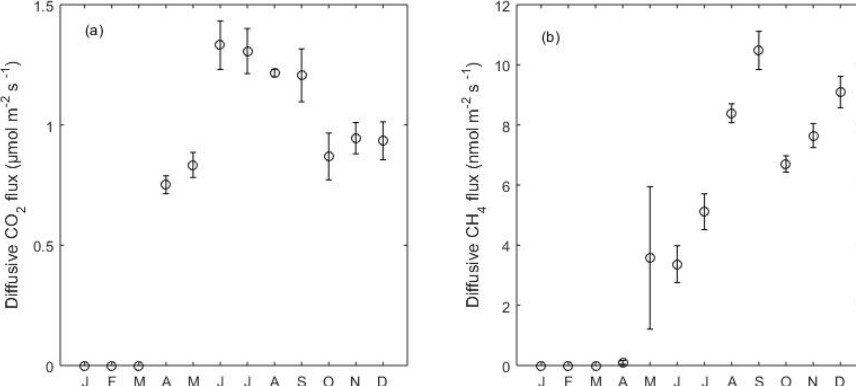


Figure 4: Monthly-average of (a) diffusive CO$_2$ and (b) CH$_4$ flux from the open-water in year 2013. Error bar indicates

the standard error of the mean. From January to March there was ice-covered period.


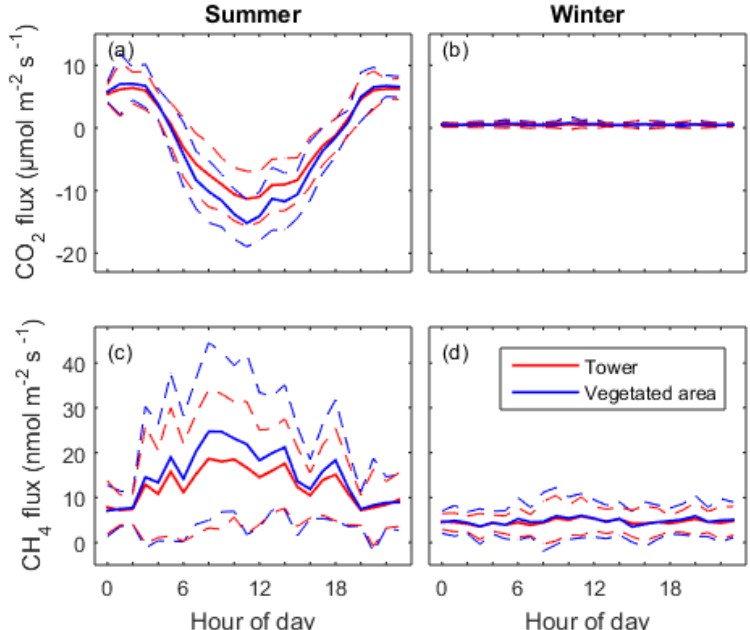


Figure 5: Mean diel pattern of the half-hourly net $CO_2$ and $CH_4$ fluxes in summer ((a) and (c)) and in winter ((b) and

(d)). The dashed lines represent the standard deviation. Red lines indicate measurement from EC tower and the blue

lines show the fluxes modelled for vegetated area.
