# Peer review of "Carbon dioxide and methane fluxes from different surface types in a created urban wetland"

_Biogeosciences, 2019_

## Referee Comment (RC1) · Dennis Baldocchi (Referee) · 29 Oct 2019

While there is a growing number of CO2 and CH4 studies from natural ecosystems, relatively few studies come from urban wetlands. Hence, this paper caught my attention as being a potentially important, new and novel contribution.

What does the term urban wetlands mean and why may greenhouse gas exchange to and from it differ from other wetlands? To my mind, I would expect urban wetlands to be recycling water from urban uses and be subject to runoff from urban landscapes, which may have elevated levels of N applications, herbicides, oil runoff from roads etc. So. these factors may affect the redox ladder and alter methane fluxes compared to those from more remote wetlands. Let's see what the authors find.

I suspect the definition of an urban wetland is overly broad and more specification may be needed. In this case the authors are studying a constructed, stormwater wetland. I suspect there are many other types of urban wetlands, just look at the urban LTER in Baltimore, MD and Phoenix, AZ as a comparison. So building a database on how they may differ or be similar should be a long term goal, initated by a project like this. It would be nice to frame this urban wetland in Finland in context to those in wetter/drier and warmer worlds.

A limitation of this study is the time scale..

'The measurements were commenced the fourth year after construction and lasted for one full year and two subsequent growing seasons'.

This study is missing many of the important pulses after construction to truly understand the dynamics of this system. This aspect is one of the greatest weaknesses of this work. But given so little data on this topic, I decided it is not a fatal flaw, in this instance. But I would not view future studies of this type that miss the dynamic of the restoration pulse viable.

The authors report:

The annual NEE of the studied wetland was 8.0 g C-CO2 m-2 yr-1 with the 95 % confidence interval between -18.9 and 34.9 g C-CO2m-2 yr-1 and FCH4 was 3.9 g C-CH4 m-2 yr-1 with the 95 % confidence interval between 3.75 and 4.07 g CH4 m-2 yr-1.

I must admit I am surprised how tiny the fluxes are, given it is a wetland, even if in Finland. I would expect a stronger sink, but granted this would be conditional of what is in the flux footprint. So careful correspondence between fluxes and footprints are key to interpret these data.

As I read on I take home the key point that it is a weak sink for only 2 months and a slow C source rest of the year. Guess in hindsight it all makes sense.

As I read the introduction, I am finding necessary conditional information. For example, open water is just not always open water. With N inputs there can be other life forms. Here the authors note

'At open-water surfaces, the net production of CO2 is a result of photosynthesis by algae, cyanobacteria as well as submerged aquatic plants, respiration of organic carbon and oxidation of CH4 produced in the water.'

This conditions meets with some of our experiences where we see azola and other aquatic plants in the open water sections. It has changed my perspective and open to this observation. The authors will need to be careful as they evaluate their 'open' water data and inform the reader if it is or not truly open water.

Glad to see citation to the work of the Estonian team of Mander et al, as they are among the few teams looking at this problem. I would also double check literature by Bill Mitsch. Their wetlands in Ohio may qualify as an urban wetland as it was close to the University in Columbus OH. Recent reports of methane fluxes come from Gil Bohrer's group, Morin et al and others.

Glad to see the authors are clued in about the key role of flux footprints. As we bend the rules of eddy covariance and ask contemporary questions and problems, we will need footprint models to partition the heterogeneity of the landscape.

Materials

The wetland is over 500 ha. This is a good size field for this work. Standard and well vetted eddy covariance is used by experts in the field who know how to carefully interpret the data. Closed path CO2, Licor, and TDL is used to measure methane fluctuations. Given the cold, wet environment I think closed path is best for this work. The authors have looked at cospectra to ensure filtering is limited or appropriated corrected for. Good micromet protocol.

Standard neural networks are used to gap fill. The methods are described in great

detail and proper attention to nodes, validation data, etc are made.

Overall I am confident about these measurements as this team has a long history of well vetted studies. The paper needs an assessment, map of the heterogeneous fetch and the flux footprint. I did not see this in the material. It is in the supplement, but it may be better placed in the paper. Starting to lose track of what is a paper vs supplement.

This paper is novel with water ch4 sensors to apply the diffusion model. First time I have seen these sensors. Bravo/brava/bravum.

The authors try to partition fluxes by the veg water fraction. I realize this is a legitmate quest and one with good intentions. We have tried this approach in the past and failed. We used multiple towers to close the system of equations with water/veg fractions. But my student, Jacyln Hatala Matthes found that the fractal dimension of the patches was key. So be careful in your partitioning.

Matthes, Jaclyn Hatala, et al. "Parsing the variability in CH4 flux at a spatially heterogeneous wetland: Integrating multiple eddy covariance towers with high‐resolution flux footprint analysis." Journal of Geophysical Research: Biogeosciences119.7 (2014): 1322-1339.

Use of the Kljun model is good. It has evolved as one of the better and most widely used.

With this the authors calculate the veg water fractions. But I must confess I don't have confidence in these numbers, especially from one tower. The reason we tried to use two towers was to get different fractions of water and vegetation with two equations and two unknowns.

I'd like to have the authors discuss the uncertainty more and critique the pros and cons of their method.

The reporting of flux reports is straight forward and standard. I have no critique or suggestions for this part.

What interests me is information on controls and processes. Here the paper has an advantage with measurements of the fluxes from the water section. But we have to be careful here. If the water is open then simple models will work. But with urban systems the N inputs can green up the water and the presence of green material will cause the diffusion models to be invalid. I need to hear more about this. So first confirm if the open water is open or is it clogged.

The controls need a bit more information on N load of the water. What is the nitrate or phosphorus levels. If there is runoff P and NO3- may affect the CH4 fluxes.

The control and process section is very simple and using correlations. I does not go into great enough detail and I am not sure if it makes a dent in our ignorance. I like methods using information theory at different time scales, I continue to worry about the roles of photosynthetic inputs to prime archaea. We learn that at different time scales temperature control may be dominant and photosynthesis may at others. Water table is important, but if it does not vary much it will not be a notable factor, yet we know mechanistically it is and if water table dropped below ground level one would see the effect.

Glad to see the authors using sustained warming potential method of Neubauer and Megonigal. I just reviewed another wetland restoration paper and they Did NOT use this method and it was a criticism of mine Methane emissions are not a single pulse, like used with the old method. It is key to use a sustained emission method.

Discussion

The authors do a nice job putting this work in context and reviewing the literature. I don't want to micromanage as there are many ways to go. I do like the discussion on O2 consumption. This is a nice angle and looks at mechanisms.

I do like seeing a bit of advice on how best to design these systems. What are the pros and cons of different water/veg fractions and what can one do to minimize methane

emissions or what are the effects of nutrient inputs on the greening of open water spaces.

In closing this paper has some novel aspects and I think it will merit publication. I do think it has some lingering issues that need to be resolved. Most seriously fraction of the water and vegetation and the modeling of fluxes from the water portion if the water is not pure. The other limitation is the time scale. It misses critical dynamic of the pulse and recovery after the wetland has been developed. This is a hole that cannot be filled.

---

## Referee Comment (RC2) · Anonymous Referee #2 · 23 Dec 2019

**Major comments**

Li et al. report a data-set of CO2 and CH4 fluxes measured by eddy-covariance (EC) in an artificial wetland in Southern Finland. The topic of the study is to quantify air-water and air-vegetation CO2 and CH4 fluxes in wetlands which is very interesting as well as extremely challenging, and rarely investigated. However, the analysis relies heavily on data gap filling, and data are reconstructed up to >70% for the first year and to >50% for the second year. I'm aware that there is commonly a very substantial data rejection for EC measurements, and that data filling is a common and accepted practice in studies of terrestrial ecosystem fluxes. However, in terrestrial ecosystem flux studies, data filling relies on relations that make sense such as primary production vs PAR and respiration vs temperature that are based on robust biological principles. Here, the authors used

correlations with the dissolved CO2 concentration to data fill the EC CO2 fluxes, which does not necessary make sense specially for the air-vegetation fluxes (because some of the CO2 signal must come from hydrological input and is independent from wetland metabolism). Furthermore, the authors use the CO2 concentration to compute the air-water CO2 fluxes that are then used in a more detailed analysis in conjunction with the EC CO2 fluxes to discuss the relative contribution of air-water and air-vegetation fluxes. So, the same variable (CO2 dissolved concentration) is used to compute two variables (air-water CO2 and EC CO2 fluxes) that are subsequently treated as independent, when they are obviously not. This, in my opinion, strongly weakens the analysis and conclusions of this study.

My other concern is that the air-water CO2 fluxes were computed from a gas transfer velocity parameterization, when it could have been relatively easy and inexpensive to measure it directly with floating domes. While it is not necessarily very constructive to point out what should have been measured, I have also some strong concerns on the choice of the parameterization. The gas transfer parameterization of Cole and Caraco (1998) was developed for large lakes, and is most probably inadequate for very small water bodies (such as the one in the present case) that usually have much lower gas transfer velocity values (Holgerson et al. 2017). The gas transfer velocity in small water bodies are even less constrained than in larger water bodies, and are bound to lead to a large source of uncertainty for computation of the fluxes that will propagate into the additional analysis based on these fluxes. Turbulence (hence gas transfer velocity) in small water bodies is mainly related to convection and less to wind speed (Holgerson et al. 2016), so wind speed based parameterizations are inadequate for small water bodies.

Minor comments

L 51: What "UN report" ? Please provide a reference.

L58: The Kyoto protocol is obsolete, we've moved on to the Paris Agreement.

[Figure]

L62-66: Are these hypothetical or based on prior studies?

L 66: Does this mean you assume "spatial heterogeneity" of artificial wetlands to be stronger than natural ones ? Why ? Natural wetlands also have "different processes of production and transportation of GHGs".

L68: dissolved $CO_2$ concentrations are usually orders of magnitude larger than $CH_4$ concentrations, so $CH_4$ oxidation plays a negligible role in the balance of production and uptake of $CO_2$.

L83: "the situation are"

L107: Might be useful to provide nutrient and chlorophyll levels to characterize the eutrophication of the lake.

L108: Please provide a reference.

L201: Part of the Reco signal is due to hydrological input of $CO_2$, and does not equate with ecosystem respiration.

L236: A nine year old paper is not a "recent study". There are numerous other studies that show a disagreement between floating chamber and other methods, for instance Vachon et al. (2010). Conversely, there are numerous studies that report gas transfer velocities in lakes that diverge from the parameterization of Cole and Caraco (1998) such as Jonsson et al. (2008) and MacIntyre et al. (2010). This is particularly the case in small water bodies where turbulence is largely unrelated to wind (Holgerson et al. 2016).

L240: The Fveg term also includes the $CH_4$ ebullition component, however the fveg term for $CH_4$ only corresponds to the vegetation, so when ebullition occurs (most of the time probably) the Fveg term is over-estimated.

L 262: This GWP value is much higher than the one proposed by the IPCC that is unanimously used. For consistency with the rest of the literature it could have been

wiser to use the IPCC values.

L 302: ppm unit in aquatic GHG literature relates to a partial pressure of CO2 and not the concentration of CO2 as stated.

References

Holgerson, M. A., E. R. Farr, and P. A. Raymond (2017), Gas transfer velocities in small forested ponds, J. Geophys. Res. Biogeosci., 122, 1011–1021, doi:10.1002/2016JG003734.

Holgerson, M. A., C. J. Zappa, and P. A. Raymond (2016), Substantial overnight reaeration by convective cooling discovered in pond ecosystems, Geophys. Res. Lett., 43, 8044–8051, doi:10.1002/2016GL070206.

Jonsson, A., J. AËŽ berg, A. Lindroth, and M. Jansson (2008), Gas transfer rate and CO2 flux between an unproductive lake and the atmosphere in northern Sweden, J. Geophys. Res., 113, G04006, doi:10.1029/2008JG000688.

MacIntyre et al. (2010) Buoyancy flux, turbulence, and the gas transfer coefficient in a stratified lake, https://doi.org/10.1029/2010GL044164 Citations: 94

Vachon D, YT Prairie and JJ. Cole (2010)The relationship between near-surface turbulence and gas transfer velocity in freshwater systems and its implications for floating chamber measurements of gas exchange, Limnol. Oceanogr., 55(4), 2010, 1723–1732

---

## Author Comment (AC1) · 24 Jan 2020

The reviewer's comments are shown in blue and our responses in black.

**(For technical reason, the figures and references cannot be displayed properly using the LaTeX option provided by the interface. Thus we uploaded the correct version in pdf in the supplement.)**

"While there is a growing number of CO2 and CH4 studies from natural ecosystems, relatively few studies come from urban wetlands. Hence, this paper caught my attention as being a potentially important, new and novel contribution.
What does the term urban wetlands mean and why may greenhouse gas exchange to and from it differ from other wetlands? To my mind, I would expect urban wetlands to be

recycling water from urban uses and be subject to runoff from urban landscapes, which may have elevated levels of N applications, herbicides, oil runoff from roads etc. So. these factors may affect the redox ladder and alter methane fluxes compared to those from more remote wetlands. Let's see what the authors find." I suspect the definition of an urban wetland is overly broad and more specification may be needed. In this case the authors are studying a constructed, stormwater wetland. I suspect there are many other types of urban wetlands, just look at the urban LTER in Baltimore, MD and Phoenix, AZ as a comparison. So building a database on how they may differ or be similar should be a long term goal, initated by a project like this. It would be nice to frame this urban wetland in Finland in context to those in wetter/drier and warmer worlds."

We thank the reviewer for the effort spent on our manuscript and the appreciation of the importance of our study. We agree with the reviewer that the definition of an urban wetland is very broad. We will rephrase it in the text as follows: "In this paper we present measurements carried out at a created urban wetland in Southern Finland in the boreal climate." (Line 22-23)

"A limitation of this study is the time scale..
'The measurements were commenced the fourth year after construction and lasted for one full year and two subsequent growing seasons'. This study is missing many of the important pulses after construction to truly under- stand the dynamics of this system. This aspect is one of the greatest weaknesses of this work. But given so little data on this topic, I decided it is not a fatal flaw, in this instance. But I would not view future studies of this type that miss the dynamic of the restoration pulse viable."

We are aware of the time scale of this study limited our capability to draw conclusions about the climate impact of the management (rewetting) when constructing an urban wetland. However, our study focused on the climate impact of the urban wetland after its establishment.

"The authors report:

The annual NEE of the studied wetland was 8.0 g $C-CO_2$ m-2 yr-1 with the 95% confidence interval between -18.9 and 34.9 g $C-CO_2$ m-2 yr-1 and $FCH_4$ was 3.9 g $C-CH_4$ m-2 yr-1 with the 95% confidence interval between 3.75 and 4.07 g $CH_4$ m-2 yr-1.

I must admit I am surprised how tiny the fluxes are, given it is a wetland, even if in Finland. I would expect a stronger sink, but granted this would be conditional of what is in the flux footprint. So careful correspondence between fluxes and footprints are key to interpret these data.

As I read on I take home the key point that it is a weak sink for only 2 months and a slow C source rest of the year. Guess in hindsight it all makes sense. As I read the introduction, I am finding necessary conditional information. For example, open water is just not always open water. With N inputs there can be other life forms. Here the authors note

'At open-water surfaces, the net production of $CO_2$ is a result of photosynthesis by algae, cyanobacteria as well as submerged aquatic plants, respiration of organic carbon and oxidation of $CH_4$ produced in the water.'

This conditions meets with some of our experiences where we see azola and other aquatic plants in the open water sections. It has changed my perspective and open to this observation. The authors will need to be careful as they evaluate their 'open' water data and inform the reader if it is or not truly open water."

In fact, we did not observe lots of metaphytic or filamentous algae during the study period. There was not large number of free floating small plants neither. There were some submerged aquatic plants which did not affect the openness of the water.

"Glad to see citation to the work of the Estonian team of Mander et al, as they are among the few teams looking at this problem. I would also double check literature by Bill Mitsch. Their wetlands in Ohio may qualify as an urban wetland as it was close

to the University in Columbus OH. Recent reports of methane fluxes come from Gil Bohrer's group, Morin et al and others.
Glad to see the authors are clued in about the key role of flux footprints. As we bend the rules of eddy covariance and ask contemporary questions and problems, we will need footprint models to partition the heterogeneity of the landscape."

The two papers from Morin et al. were cited in the manuscript (Line 76).

"Materials
The wetland is over 500 ha. This is a good size field for this work. Standard and well vetted eddy covariance is used by experts in the field who know how to carefully interpret the data. Closed path CO2, Licor, and TDL is used to measure methane fluctuations. Given the cold, wet environment I think closed path is best for this work. The authors have looked at cospectra to ensure filtering is limited or appropriated corrected for. Good micromet protocol.
Standard neural networks are used to gap fill. The methods are described in great detail and proper attention to nodes, validation data, etc are made.
Overall I am confident about these measurements as this team has a long history of well vetted studies. The paper needs an assessment, map of the heterogeneous fetch and the flux footprint. I did not see this in the material. It is in the supplement, but it may be better placed in the paper. Starting to lose track of what is a paper vs supplement."

The map of study site overlapped with climatological footprints over the study period will be moved to the main body of the manuscript.

"This paper is novel with water ch4 sensors to apply the diffusion model. First time I have seen these sensors. Bravo/brava/bravum.
The authors try to partition fluxes by the veg water fraction. I realize this is a legitmate quest and one with good intentions. We have tried this approach in the past and failed.

We used multiple towers to close the system of equations with water/veg fractions. But my student, Jacyln Hatala Matthes found that the fractal dimension of the patches was key. So be careful in your partitioning.

Matthes, Jaclyn Hatala, et al. "Parsing the variability in CH4 flux at a spatially het- ero-geneous wetland: Integrating multiple eddy covariance towers with high resolution flux footprint analysis." Journal of Geophysical Research: Biogeosciences,119.7 (2014): 1322-1339.

Use of the Kljun model is good. It has evolved as one of the better and most widely used.

With this the authors calculate the veg water fractions. But I must confess I don't have confidence in these numbers, especially from one tower. The reason we tried to use two towers was to get different fractions of water and vegetation with two equations and two unknowns.

I'd like to have the authors discuss the uncertainty more and critique the pros and cons of their method.

The reporting of flux reports is straight forward and standard. I have no critique or suggestions for this part."

We will critique our method as follows: "The uncertainty of the vegetation and water fraction come from two sources. Firstly, the delineation of the distinct surface types was conducted based on a land surface map of the growing season in 2013, which neglect the change in the spatial extent of the vegetation throughout time. Secondly, although Kljun model **?** is proved to be robust and general, there are uncertainties in the model prediction. To be more confident in the footprint estimation, it would be good to compare our results with large eddy simulation, however it is out of the scope of the current study. With only one EC tower we could not cross check the results as done in another study **?**.

However, we chose to follow a simple approach dividing the landscape into vegetation and open water because we did not observe significant vegetation expansion during

the growing season and the area of open water is relatively constant. Furthermore, the clear effect of the footprint-weighted fraction of open water on the synchronization between EC $CH_4$ measurements and diffusive $CH_4$ flux from water (Line 471-477, Fig.S6 in the supplement material) was nicely presented in our analysis, so that we think the simple method used is sufficient to capture the major pattern in vegetation and water fraction in our study. "

"What interests me is information on controls and processes. Here the paper has an advantage with measurements of the fluxes from the water section. But we have to be careful here. If the water is open then simple models will work. But with urban systems the N inputs can green up the water and the presence of green material will cause the diffusion models to be invalid. I need to hear more about this. So first confirm if the open water is open or is it clogged."

The open water is open (see the corresponding responses above). Submerged aquatic plants should not affect the validation of the diffusion model. Furthermore, the estimated diffusive fluxes of methane ($CH_4$) and carbon dioxide ($CO_2$) were well situated in the range of the diffusive gas fluxes over open lakes from other studies **??**, which supported our assumption that the water was not covered by floating plants.

"The controls need a bit more information on N load of the water. What is the nitrate or phosphorus levels. If there is runoff P and NO3- may affect the CH4 fluxes."

A figure showing the nutrient levels in the water will be added to the manuscript (Fig. 1) as well as the following text: "$NO_3$-N was measured with Scan sensors (Scan gmbh, Austria) and total phosphorus (TP) was calculated based on turbidity data which was measured at 10-min intervals **?**. The median TP concentration measured at the outflow monitoring station was 56 $\mu$g $L^{-1}$ and the median $NO_3$-N concentration was 0.69 mg $L^{-1}$. In annual perspective TP and $NO_3$-N concentration consisted of several runoff

peaks occurring after rain or snow melting events. This wetland serves as a nutrient removal measure as it improved water quality by retaining P and N from runoff before the release to the receiving lake, where the annual TP reduction was 13% and $NO_3$-N reduction was 14% in year 2014 ?."

"The control and process section is very simple and using correlations. I does not go into great enough detail and I am not sure if it makes a dent in our ignorance. I like methods using information theory at different time scales, I continue to worry about the roles of photosynthetic inputs to prime archaea. We learn that at different time scales temperature control may be dominant and photosynthesis may at others. Water table is important, but if it does not vary much it will not be a notable factor, yet we know mechanistically it is and if water table dropped below ground level one would see the effect."

We conducted wavelet coherence analysis to reveal the processes and environmental controls of the gases at different time scales. The magnitude of the wavelet coherence and the phase differences between ecosystem $CO_2$ and $CH_4$ fluxes and environmental variables are shown in Figs. 2 and 3. Here we show the results of net ecosystem productivity (NEP; NEP= -NEE) instead of NEE for a better interpretation of the phase arrow (higher positive value in NEP means higher $CO_2$ uptake).

We found strong positive correlations between NEP and temperature, radiation at 1-day scale due to the diel temperature and radiation cycles. On average, $T_{air}$ and $T_{water}$ are leading NEP by $\sim$3h and $\sim$8h, respectively, while radiation is almost in-phase with NEP. The variation of TP is leading the change in NEP at 1-day scale (more TP leads to more $CO_2$ uptake) where the time lag varies between 1 to 5 hours (Fig.4 (d)).

$CH_4$ flux has correlation with temperature at 1-day scale where $T_{air}$ and $T_{water}$ are leading $CH_4$ flux by $\sim$1h and $\sim$6h, respectively (Fig. 5). $CH_4$ flux has also correlation with temperatures at 16-32-day scale (Fig. 3). Radiation is in-phase with $CH_4$ flux at

1-day scale (Fig. 4(c)). TP has positive correlation with $CH_4$ flux (more TP leads to more $CH_4$ emission) at 1-day scale and TP is leading $CH_4$ flux by $\sim$2h. Surprisingly, water level did not show any consistent correlation with $CH_4$ flux at any time scale which may be due to the small variation in water level during the growing season.

GPP and $CH_4$ flux are correlated at multiple time scales (Fig. 6). At 1-day scale they are nearly in-phase, which indicates rapid link between photosynthesis and $CH_4$ emission (Fig. 7(a)). At 90-day scale, GPP is leading $CH_4$ flux by 17-20 days which can possibly be explained by the lags between environmental controls of GPP and that of $CH_4$ on a seasonal scale (Fig. 7(b)).

After all, it is worth noting that the correlations between the fluxes and environmental variables revealed by wavelet coherence analysis can be overstated, as much of the flux data has been gap-filled using these variables. Therefore, in the revised manuscript we will only add figures which show the results between fluxes ($CO_2$ and $CH_4$) and those independent environmental variables ($NO_3$-N and TP).

"Glad to see the authors using sustained warming potential method of Neubauer and Megonigal. I just reviewed another wetland restoration paper and they Did NOT use this method and it was a criticism of mine Methane emissions are not a single pulse, like used with the old method. It is key to use a sustained emission method."

To be consistent with other references using IPCC value as reviewer # 2 suggested, we will add also the results using the conventional global warming potential in the manuscript.

Discussion

"The authors do a nice job putting this work in context and reviewing the literature. I don't want to micromanage as there are many ways to go. I do like the discussion on O2 consumption. This is a nice angle and looks at mechanisms. I do like seeing a bit of advice on how best to design these systems. What are the pros and cons of different

water/veg fractions and what can one do to minimize methane emissions or what are the effects of nutrient inputs on the greening of open water spaces."

We will add the following paragraphs in the text on the advices of designing urban wetland ecosystem.

"Firstly, in our study we found that the radiative forcing effect of the open-water area exceeded the vegetation area in an urban wetland in Finland. Thus, if considering only the climate impact, it would be advisable to have lower water/vegetation fraction which means limiting open-water surfaces and setting a design preference for areas of emergent vegetation in the establishment of urban wetlands.
Secondly, our results showed that total phosphorus enhanced both $CO_2$ uptake and $CH_4$ emission which have contradictory climate impacts to the ecosystem. Although it is out of the scope of our study, it would be very interesting to understand the mechanisms, to quantify the magnitude and the duration of these enhancements induced by nutrient input. Previous studies have found that nutrient inputs can influence the identity of the key primary producer (submerged plants versus phytoplankton) in the water, which is crucial in shaping the $CH_4$ emission from shallow water **??**. Submerged plants may decrease $CH_4$ production in the lake by producing alleochemicals, transporting oxygen to the sediment and providing good habitat for $CH_4$ oxidizing bacteria **?**, while phytoplankton was shown to significantly increase $CH_4$ ebullition by changing the quality of the dissolved organic carbon which promotes methanogenesis **?** or/and by altering the sediment texture and redox conditions favoring the release of bubbles. As a result, we suggest to control the nutrient input to the water of the newly established wetland to limit the abundance of phytoplankton as well as to support the existence of submerged plants."

"In closing this paper has some novel aspects and I think it will merit publication. I do think it has some lingering issues that need to be resolved. Most seriously fraction of the water and vegetation and the modeling of fluxes from the water portion if the water

is not pure. The other limitation is the time scale. It misses critical dynamic of the pulse and recovery after the wetland has been developed. This is a hole that cannot be filled."

We thank the reviewer for his constructive suggestions. We are aware of the limitation in our study and they will be more clearly acknowledged in the revised manuscript. Future studies are ought to be planned in a manner which can "fill the hole".

**Fig. 1.** The daily average of (a) rainfall, (b) total phosphorus concentration and (c) $NO_3$-N concentration measured at the outlet monitoring station in year 2013. The lake was covered by ice from January to March and it was free of ice after the end of March.

figures/wc_50_CO2.png

**Fig. 2.** Wavelet coherence analysis and the phase difference between net ecosystem production (NEP; NEP=-NEE) and environmental controls from January to December 2013. The color represents the power of the coherence from 0 to 1. The phase difference is indicated by black arrows which only show up where the coherence is greater than or equal to 0.5. $\rightarrow$ indicates in-phase (two time series in synchrony) and arrows in other direction indicate out of phase (representing lags between time series), i.e. $\leftarrow$ indicates anti-phase, $\downarrow$ indicates the $1^{st}$ series (NEP) leads by quarter-cycle and $\uparrow$ indicates $2^{nd}$ series (environmental controls) leads by quarter-cycle. White dash contour lines indicate the cone of influence. PCO2, PCH4, NO3-N and TP indicate the concentrations of $CO_2$, $CH_4$, $NO_3$-N and total phosphorus in the water.

figures/wc_50_CH4.png

**Fig. 3.** Wavelet coherence analysis and the phase difference between ecosystem $CH_4$ flux (FCH4) and environmental controls from January to December 2013. The color represents the power of the coherence from 0 to 1. The phase difference is indicated by black arrows which only show up where the coherence is greater than or equal to 0.5. $\rightarrow$ indicates in-phase (two time series in synchrony) and arrows in other direction indicate out of phase (representing lags between time series), i.e. $\leftarrow$ indicates anti-phase, $\downarrow$ indicates the $1^{st}$ series (FCH4) leads by quarter-cycle and $\uparrow$ indicates $2^{nd}$ series (environmental controls) leads by quarter-cycle. White dash contour lines indicate the cone of influence. PCO2, PCH4, NO3-N and TP indicate the concentrations of $CO_2$, $CH_4$, $NO_3$-N and total phosphorus in the water.

(a) (b) (c) (d)

**Fig. 4.** Time lag between NEP and (a) air temperature($T_{air}$), (b) water temperature ($T_{water}$), (c) radiation and (d) total phosphorus (TP) at 1-day time scale. Positive time lags indicate the environmental variables are leading NEP and vice versa.

(a) (b) (c) (d)

**Fig. 5.** Time lag between $CH_4$ flux ($F_{CH4}$) and (a) air temperature ($T_{air}$), (b) water temperature ($T_{water}$) and (c) radiation and (d) total phosphorus (TP) at 1-day time scale. Positive time lag indicate the environmental variables are leading ($F_{CH4}$) and vice versa.

**Fig. 6.** Wavelet coherence analysis for the time series between GPP and FCH4 (caption same as Fig. 2).

(a) (b)

**Fig. 7.** Time lag between gross primary productivity (GPP) and CH$_4$ flux (FCH4) (a) at 1-day and (b) 90-day time scale. Negative time lag means that GPP is leading FCH4.

---

## Author Comment (AC2) · 24 Jan 2020

The reviewer's comments are shown in blue and our responses in black.

**(For technical reason, the figures and references cannot be displayed properly using the LaTeX option provided by the interface. Thus we uploaded the correct version in pdf in the supplement.)**

Major comments
"Li et al. report a data-set of $CO_2$ and $CH_4$ fluxes measured by eddy-covariance (EC) in an artificial wetland in Southern Finland. The topic of the study is to quantify air-water and air-vegetation $CO_2$ and $CH_4$ fluxes in wetlands which is very interesting as well as extremely challenging, and rarely investigated. However, the analysis relies

heavily on data gap filling, and data are reconstructed up to > 70% for the first year and up to > 50% for the second year. I'm aware that there is commonly a very substantial data rejection for EC measurements, and that data filling is a common and accepted practice in studies of terrestrial ecosystem fluxes. However, in terrestrial ecosystem flux studies, data filling relies on relations that make sense such as primary production vs PAR and respiration vs temperature that are based on robust biological principles. Here, the authors used correlations with the dissolved CO2 concentration to data fill the EC CO2 fluxes, which does not necessary make sense specially for the air-vegetation fluxes (because some of the CO2 signal must come from hydrological input and is independent from wetland metabolism)."

We thank the reviewer for the time and effort used to our manuscript.
The wetland ecosystem in our study is comprised of both open water and vegetation surface type, both of which contribute simultaneously to the EC measurement. As the dictinct processes involved in each surface type, the relationships between environmental variables and EC fluxes are very complicated, which makes the gapfilling using traditional process-based method difficult. Therefore, we gap-filled the EC data using an artificial neural network (ANN) model. ANN is essentially a empirical non-linear regression model **?**, which is a data-based model rather than process-based models such as Michaelis-Menten light response function for photosynthesis. ANN is known for its capability of modelling complex relationships **??**. The input parameters of the model are chosen to maximize the model accuracy in keeping with the principle of parsimony. The dissolved $CO_2$ and $CH_4$ concentrations are chosen in the model as they greatly increased the model precision (see Figure S2 from supplement material). This is also reasonable because a fraction of the flux measured by EC tower comes from the diffusive fluxes from the open water which is linked to gas concentration in the water.

"Furthermore, the authors use the CO2 concentration to compute the air- water CO2

fluxes that are then used in a more detailed analysis in conjunction with the EC CO2 fluxes to discuss the relative contribution of air-water and air-vegetation fluxes. So, the same variable (CO2 dissolved concentration) is used to compute two variables (air-water CO2 and EC CO2 fluxes) that are subsequently treated as independent, when they are obviously not. This, in my opinion, strongly weakens the analysis and conclusions of this study."

We do not fully understand the comment on the independency of the variables. We did not assume air-water $CO_2$ and EC $CO_2$ fluxes to be independent quantities. Rather, they are interlinked as EC flux is comprised of air-water flux and air-vegetation flux. We calculated the air-vegetation flux ($f_{veg}$) using eq. 7:

$F_{EC} = F_{water} \times f_{water} + F_{veg} \times f_{veg}$

We then simply added up the numbers to obtain annual balance of the flux. We did not apply any statistical model to the calculated variables where the independency of data is required.

"My other concern is that the air-water CO2 fluxes were computed from a gas transfer velocity parameterization, when it could have been relatively easy and inexpensive to measure it directly with floating domes. While it is not necessarily very constructive to point out what should have been measured, I have also some strong concerns on the choice of the parameterization. The gas transfer parameterization of Cole and Caraco (1998) was developed for large lakes, and is most probably inadequate for very small water bodies (such as the one in the present case) that usually have much lower gas transfer velocity values (Holgerson et al. 2017). The gas transfer velocity in small water bodies are even less constrained than in larger water bodies, and are bound to lead to a large source of uncertainty for computation of the fluxes that will propagate into the additional analysis based on these fluxes. Turbulence (hence gas transfer velocity) in small water bodies is mainly related to convection and less to wind speed (Holgerson et al. 2016), so wind speed based parameterizations are inadequate for small water bodies."

[Figure]

We are fully aware of the limitation of using Cole-Caraco parameterization to estimate air-water fluxes from small lake (discussed in Line 232- 239). To quantify the potential uncertainty, we have calculated the gas transfer velocity using another model which takes heat flux into account **?**. However, due to the shortage of the incoming shortwave and incoming longwave radiation data, applying Heiskanen model has created much larger gaps. Especially for $CH_4$, the spring peak in air-water $CH_4$ flux would then be completely missing. Additionally, unlike ponds surrounded by the forest **?**, the water body in this study is located in an open area where the contribution of wind shear to the turbulence in the surface mixed layer should be relatively high. Like in our study, the parameterization of Cole and Caraco has been similarly applied to connected small open-water pools in a restored wetland which found reasonable agreement between the model estimation and the measurements **?**. Furthermore, the estimated air-water fluxes of $CH_4$ and $CO_2$ based on the current model were well within the range of the diffusive gas fluxes over small lakes from other studies **??**. Therefore, we decide to continue using Cole-Caraco model to estimate diffusive fluxes from the water, bearing in mind that the calculated fluxes can be underestimated.

Minor comments

"L 51: What "UN report" ? Please provide a reference."

The reference is : United Nations, Department of Economic and Social Affairs, "Global Sustainable Development Report 2016", New York, July, 2016.

"L58: The Kyoto protocol is obsolete, we've moved on to the Paris Agreement."

We will change "Kyoto protocol" to "Paris Agreement".

"L62-66: Are these hypothetical or based on prior studies?"

1) is based on the knowledge of vegetation dynamics. We will spell it out in the text: "When a urban wetland is newly created by rewetting the landscape, it takes time for the vegetation to establish itself in the new environment. The low coverage of vegetation at the initial phase of wetland establishment can lead to low $CO_2$ sequestration on a ecosystem scale." For 2), the high nutrient level in the receiving water into the urban wetland was observed by multiple studies. We will add references to back up this statement **???**. And for 3), we agree that natural wetlands can also exhibit large spatial heterogeneity in vegetation and hydrology, thus we will remove this sentence.

"L 66: Does this mean you assume "spatial heterogeneity" of artificial wetlands to be stronger than natural ones ? Why ? Natural wetlands also have "different processes of production and transportation of GHGs""

We will remove this sentence as mentioned above.

"L68: dissolved CO2 concentrations are usually orders of magnitude larger than CH4 concentrations, so CH4 oxidation plays a negligible role in the balance of production and uptake of CO2."

We will remove the "oxidation of CH4 produced in the water"

"L83: 'the situation are' "

We will change it to "the situations are "

"L107: Might be useful to provide nutrient and chlorophyll levels to characterize the eutrophication of the lake."

The level of total phosphorus and $NO_3$-N are provided now in Fig. **??**. Chlorophyll level was not measured, unfortunately.

"L108: Please provide a reference."

The following references will be added to the manuscript:

Varis O, Sirvio H, Kettunen J. 1989. Multivariate analysis of lake phytoplankton and environmental factors. Arch Hydrobiol. 117:163-175.

Salonen V-P, Varjo E. 2000. Vihdin Enäjärven kunnostuksen vaikutus pohjasedimentin ominaisuuksiin [The effects of restoration actions at the Lake Enäjärvi in Vihti, Finland on bottom sediment characteristics]. Geologi. 52:159-163. Finnish.

"L201: Part of the Reco signal is due to hydrological input of CO2, and does not equate with ecosystem respiration."

We will remove the section of NEE partitioning (Line 201-214).

L 236: A nine year old paper is not a 'recent study'. There are numerous other studies that show a disagreement between floating chamber and other methods, for instance Vachon et al. (2010). Conversely, there are numerous studies that report gas transfer velocities in lakes that diverge from the parameterization of Cole and Caraco (1998) such as Jonsson et al. (2008) and MacIntyre et al. (2010). This is particularly the case in small water bodies where turbulence is largely unrelated to wind (Holgerson et al. 2016)."

See the corresponding responses above. While we acknowledge that both wind shear and convection have significant contributions to turbulence in the surface mixed layer above small water bodies, but we think the current method is sufficient to capture the basic patterns in the diffusive fluxes.

"L240: The Fveg term also includes the CH4 ebullition component, however the fveg term for CH4 only corresponds to the vegetation, so when ebullition occurs (most of the time probably) the Fveg term is over-estimated."

We acknowledge that Fveg term can be over-estimated as we did not have independent measures for ebullition. We have discussed about it in the text as one of the potential uncertainties in our study (Line 478-485). Furthermore, in a recent study where ebullition was measured with chambers in a restored wetland, ebullition from the open water was shown to have only minor significance accounting for 4.1% of ecosystem $CH_4$ flux **?**. We think that our ignorance of ebullition would not change much of the general conclusion of our study.

"L 262: This GWP value is much higher than the one proposed by the IPCC that is unanimously used. For consistency with the rest of the literature it could have been wiser to use the IPCC values."

We used sustained global warming potential with a 45 as the $CO_2$ equivalents of $CH_4$ fluxes **?** because greenhouse gas emissions are not single pulses, so it is reasonable to "use a sustained emission method" as mentioned by Reviewer #1. But for a easier comparison with other studies, we now also calculate $CH_4$ fluxes as $CO_2$ equivalents using a global warming potential (GWP) of 34 following the 5th Assessment Report of IPCC **?**. The GWP of $CH_4$ fluxes from ecosystem, water and vegetation are 0.177, 0.077 and 0.195 $kg\,CO_2$-eq $m^{-2}$, and they will be added to the result section.

"L 302: ppm unit in aquatic GHG literature relates to a partial pressure of CO2 and not the concentration of CO2 as stated."

ppm unit will be converted to µmol/L using Henry's law.

**Supplement:**

The reviewer's comments are shown in blue and our responses in black.

Major comments
"Li et al. report a data-set of CO2 and CH4 fluxes measured by eddy-covariance (EC) in an artificial wetland in Southern Finland. The topic of the study is to quantify air-water and air-vegetation CO2 and CH4 fluxes in wetlands which is very interesting as well as extremely challenging, and rarely investigated. However, the analysis relies heavily on data gap filling, and data are reconstructed up to $> 70\%$ for the first year and up to $> 50\%$ for the second year. I'm aware that there is commonly a very substantial data rejection for EC measurements, and that data filling is a common and accepted practice in studies of terrestrial ecosystem fluxes. However, in terrestrial ecosystem flux studies, data filling relies on relations that make sense such as primary production vs PAR and respiration vs temperature that are based on robust biological principles. Here, the authors used correlations with the dissolved CO2 concentration to data fill the EC CO2 fluxes, which does not necessary make sense specially for the air-vegetation fluxes (because some of the CO2 signal must come from hydrological input and is independent from wetland metabolism)."

We thank the reviewer for the time and effort used to our manuscript.
The wetland ecosystem in our study is comprised of both open water and vegetation surface type, both of which contribute simultaneously to the EC measurement. As the dictinct processes involved in each surface type, the relationships between environmental variables and EC fluxes are very complicated, which makes the gapfilling using traditional process-based method difficult. Therefore, we gap-filled the EC data using an artificial neural network (ANN) model. ANN is essentially a empirical non-linear regression model (Papale & Valentini, 2003), which is a data-based model rather than process-based models such as Michaelis-Menten light response function for photosynthesis. ANN is known for its capability of modelling complex relationships (Moffat et al., 2007; Richardson et al., 2008). The input parameters of the model are chosen to maximize the model accuracy in keeping with the principle of parsimony. The dissolved $CO_2$ and $CH_4$ concentrations are chosen in the model as they greatly increased the model precision (see Figure S2 from supplement material). This is also reasonable because a fraction of the flux measured by EC tower comes from the diffusive fluxes from the open water which is linked to gas concentration in the water.

"Furthermore, the authors use the CO2 concentration to compute the air- water CO2 fluxes that are then used in a more detailed analysis in conjunction with the EC CO2 fluxes to discuss the relative contribution of air-water and air-vegetation fluxes. So, the same variable (CO2 dissolved concentration) is used to compute two variables (air-water CO2 and EC CO2 fluxes) that are subsequently treated as independent, when they are obviously not. This, in my opinion, strongly weakens the analysis and conclusions of this study."
We do not fully understand the comment on the independency of the variables. We did not assume air-water $CO_2$ and EC $CO_2$ fluxes to be independent quantities. Rather, they are interlinked as EC flux is comprised of air-water flux and air-vegetation flux. We calculated the air-vegetation flux ($f_{veg}$) using eq. 7:
$F_{EC} = F_{water} \times f_{water} + F_{veg} \times f_{veg}$
We then simply added up the numbers to obtain annual balance of the flux. We did not apply any statistical model to the calculated variables where the independency of data is required.

"My other concern is that the air-water CO2 fluxes were computed from a gas transfer velocity parameterization, when it could have been relatively easy and inexpensive to measure it directly with floating domes. While it is not necessarily very constructive to point out what should have been measured, I have also some strong concerns on the choice of the parameterization. The gas transfer parameterization of Cole and Caraco (1998) was developed for large lakes, and is most probably inadequate for very small water bodies (such as the one in the present case) that usually have much lower gas transfer velocity values (Holgerson et al. 2017). The gas transfer velocity in small water bodies are even less constrained than in larger water bodies, and are bound to lead to a large source of uncertainty for computation of the fluxes that will propagate into the additional analysis based on these fluxes. Turbulence (hence gas transfer velocity) in small water bodies is mainly related to convection and less to wind speed (Holgerson et al. 2016), so wind speed based parameterizations are inadequate for small water bodies."

We are fully aware of the limitation of using Cole-Caraco parameterization to estimate air-water fluxes from small lake (discussed in Line 232- 239). To quantify the potential uncertainty, we have calculated the gas transfer velocity using another model which takes heat flux into account (Heiskanen et al., 2014). However, due to the shortage of the incoming shortwave and incoming longwave radiation data, applying Heiskanen model has created much larger gaps. Especially for $CH_4$, the spring peak in air-water $CH_4$ flux would then be completely missing. Additionally, unlike ponds surrounded by the forest (Holgerson, Farr, & Raymond, 2017), the water body in this study is located in an open area where the contribution of wind shear to the turbulence in the surface mixed layer should be relatively high. Like in our study, the parameterization of Cole and Caraco has been similarly applied to connected small open-water pools in a restored wetland which found reasonable agreement between the model estimation and the measurements (McNicol et al., 2017). Furthermore, the estimated air-water fluxes of $CH_4$ and $CO_2$ based on the current model were well within the range of the diffusive gas fluxes over small lakes from other studies (Erkkila et al., 2018; Mammarella et al., 2015). Therefore, we decide to continue using Cole-Caraco model to estimate diffusive fluxes from the water, bearing in mind that the calculated fluxes can be underestimated.

Minor comments

"L 51: What "UN report" ? Please provide a reference."

The reference is : United Nations, Department of Economic and Social Affairs, "Global Sustainable Development Report 2016", New York, July, 2016.

"L58: The Kyoto protocol is obsolete, we've moved on to the Paris Agreement."

We will change "Kyoto protocol" to "Paris Agreement".

"L62-66: Are these hypothetical or based on prior studies?"

1) is based on the knowledge of vegetation dynamics. We will spell it out in the text: "When a urban wetland is newly created by rewetting the landscape, it takes time for the vegetation to establish itself in the new environment. The low coverage of vegetation at the initial phase of wetland establishment can lead to low $CO_2$ sequestration on a ecosystem scale." For 2), the high nutrient level in the receiving water into the urban wetland was observed by multiple studies. We will add references to back up this statement (Lu et al., 2009; Vohla, Alas, Nurk, Baatz, & Mander, 2007; Valkama et al., 2017). And for 3), we agree that natural wetlands can also exhibit large spatial heterogeneity in vegetation and hydrology, thus we will remove this sentence.

"L 66: Does this mean you assume "spatial heterogeneity" of artificial wetlands to be stronger than natural ones ? Why ? Natural wetlands also have "different processes of production and transportation of GHGs""

We will remove this sentence as mentioned above.

"L68: dissolved CO2 concentrations are usually orders of magnitude larger than CH4 concentrations, so CH4 oxidation plays a negligible role in the balance of production and uptake of CO2."

We will remove the "oxidation of CH4 produced in the water"

"L83: 'the situation are' "

We will change it to "the situations are "

"L107: Might be useful to provide nutrient and chlorophyll levels to characterize the eutrophication of the lake."

The level of total phosphorus and $NO_3$-N are provided now in Fig. 1. Chlorophyll level was not measured, unfortunately.

"L108: Please provide a reference."

The following references will be added to the manuscript:

Varis O, Sirvio H, Kettunen J. 1989. Multivariate analysis of lake phytoplankton and environmental factors. Arch Hydrobiol. 117:163-175.

Salonen V-P, Varjo E. 2000. Vihdin Enäjärven kunnostuksen vaikutus pohjasedimentin ominaisuuksiin [The effects of restoration actions at the Lake Enäjärvi in Vihti, Finland on bottom sediment characteristics]. Geologi. 52:159-163. Finnish.

"L201: Part of the Reco signal is due to hydrological input of CO2, and does not equate with ecosystem respiration."

We will remove the section of NEE partitioning (Line 201-214).

L 236: A nine year old paper is not a 'recent study'. There are numerous other studies that show a disagreement between floating chamber and other methods, for instance Vachon et al. (2010). Conversely, there are numerous studies that report gas transfer velocities in lakes that diverge from the parameterization of Cole and Caraco (1998) such as Jonsson et al. (2008) and MacIntyre et al. (2010). This is particularly the case in small water bodies where turbulence is largely unrelated to wind (Holgerson et al. 2016)."

See the corresponding responses above. While we acknowledge that both wind shear and convection have significant contributions to turbulence in the surface mixed layer above small water bodies, but we think the current method is sufficient to capture the basic patterns in the diffusive fluxes.

"L240: The Fveg term also includes the CH4 ebullition component, however the fveg term for CH4 only corresponds to the vegetation, so when ebullition occurs (most of the time probably) the Fveg term is over-estimated."

We acknowledge that Fveg term can be over-estimated as we did not have independent measures for ebullition. We have discussed about it in the text as one of the potential uncertainties in our

[Figure]

Figure 1: The daily average of (a) rainfall, (b) total phosphorus concentration and (c) $NO_3$-N concentration measured at the outlet monitoring station in year 2013. The lake was covered by ice from January to March and it was free of ice after the end of March.

study (Line 478-485). Furthermore, in a recent study where ebullition was measured with chambers in a restored wetland, ebullition from the open water was shown to have only minor significance accounting for 4.1% of ecosystem $CH_4$ flux (McNicol et al., 2017). We think that our ignorance of ebullition would not change much of the general conclusion of our study.

"L 262: This GWP value is much higher than the one proposed by the IPCC that is unanimously used. For consistency with the rest of the literature it could have been wiser to use the IPCC values."

We used sustained global warming potential with a 45 as the $CO_2$ equivalents of $CH_4$ fluxes (Neubauer & Megonigal, 2015) because greenhouse gas emissions are not single pulses, so it is reasonable to "use a sustained emission method" as mentioned by Reviewer #1. But for a easier comparison with other studies, we now also calculate $CH_4$ fluxes as $CO_2$ equivalents using a global warming potential (GWP) of 34 following the 5th Assessment Report of IPCC (Myhre et al., 2013). The GWP of $CH_4$ fluxes from ecosystem, water and vegetation are 0.177, 0.077 and 0.195 kg $CO_2$-eq m$^{-2}$, and they will be added to the result section.

"L 302: ppm unit in aquatic GHG literature relates to a partial pressure of CO2 and not the concentration of CO2 as stated."

ppm unit will be converted to µmol/L using Henry's law.

**References**

Erkkila, K.-M., Ojala, A., Bastviken, D., Biermann, T., Heiskanen, J. J., Lindroth, A., ... Mammarella, I. (2018). Methane and carbon dioxide fluxes over a lake: comparison between eddy covariance, floating chambers and boundary layer method. *BIOGEOSCIENCES*, *15*(2), 429-445. doi: 10.5194/bg-15-429-2018

Heiskanen, J. J., Mammarella, I., Haapanala, S., Pumpanen, J., Vesala, T., Macintyre, S., & Ojala, A. (2014). Effects of cooling and internal wave motions on gas transfer coefficients in a boreal lake. *TELLUS SERIES B-CHEMICAL AND PHYSICAL METEOROLOGY*, *66*. doi: 10.3402/tellusb.v66.22827

Holgerson, M. A., Farr, E. R., & Raymond, P. A. (2017). Gas transfer velocities in small forested ponds. *JOURNAL OF GEOPHYSICAL RESEARCH-BIOGEOSCIENCES*, *122*(5), 1011-1021. doi: 10.1002/2016JG003734

Lu, S. Y., Wu, F. C., Lu, Y., Xiang, C. S., Zhang, P. Y., & Jin, C. X. (2009, MAR 4). Phosphorus removal from agricultural runoff by constructed wetland. *ECOLOGICAL ENGINEERING*, *35*(3), 402-409. doi: 10.1016/j.ecoleng.2008.10.002

Mammarella, I., Nordbo, A., Rannik, U., Haapanala, S., Levula, J., Laakso, H., ... Vesala, T. (2015). Carbon dioxide and energy fluxes over a small boreal lake in Southern Finland. *JOURNAL OF GEOPHYSICAL RESEARCH-BIOGEOSCIENCES*, *120*(7), 1296-1314. doi: 10.1002/2014JG002873

McNicol, G., Sturtevant, C. S., Knox, S. H., Dronova, I., Baldocchi, D. D., & Silver, W. L. (2017). Effects of seasonality, transport pathway, and spatial structure on greenhouse gas fluxes in a restored wetland. *GLOBAL CHANGE BIOLOGY*, *23*(7), 2768-2782. doi: 10.1111/gcb.13580

Moffat, A. M., Papale, D., Reichstein, M., Hollinger, D. Y., Richardson, A. D., Barr, A. G., ... Stauch, V. J. (2007). Comprehensive comparison of gap-filling techniques for eddy covariance net carbon fluxes. *AGRICULTURAL AND FOREST METEOROLOGY*, *147*(3-4), 209-232. doi: 10.1016/j.agrformet.2007.08.011

Myhre, G., Shindell, D., Breon, F., Collins, W., Fuglestvedt, J., Huang, J., ... Zhang, H. (2013). Anthropogenic and natural radiative forcing [Book Section]. In T. Stocker et al. (Eds.), *Climate change 2013: The physical science basis. contribution of working group i to the fifth assessment report of the intergovernmental panel on climate change* (p. 659-740). Cambridge, United Kingdom and New York, NY, USA: Cambridge University Press.

Neubauer, S. C., & Megonigal, J. P. (2015). Moving Beyond Global Warming Potentials to Quantify the Climatic Role of Ecosystems. *ECOSYSTEMS*, *18*(6), 1000-1013. doi: 10.1007/s10021-015-9879-4

Papale, D., & Valentini, A. (2003). A new assessment of European forests carbon exchanges by eddy fluxes and artificial neural network spatialization. *GLOBAL CHANGE BIOLOGY*, *9*(4), 525-535. doi: 10.1046/j.1365-2486.2003.00609.x

Richardson, A. D., Mahecha, M. D., Falge, E., Kattge, J., Moffat, A. M., Papale, D., ... Hollinger, D. Y. (2008). Statistical properties of random CO2 flux measurement uncertainty inferred from model residuals. *AGRICULTURAL AND FOREST METEOROLOGY*, *148*(1), 38-50. doi: 10.1016/j.agrformet.2007.09.001

Valkama, P., Makinen, E., Ojala, A., Vahtera, H., Lahti, K., Rantakokko, K., ... Wahlroos, O. (2017). Seasonal variation in nutrient removal efficiency of a boreal wetland detected

by high-frequency on-line monitoring. *ECOLOGICAL ENGINEERING*, *98*, 307-317. doi: 10.1016/j.ecoleng.2016.10.071

Vohla, C., Alas, R., Nurk, K., Baatz, S., & Mander, U. (2007). Dynamics of phosphorus, nitrogen and carbon removal in a horizontal subsurface flow constructed wetland. *SCIENCE OF THE TOTAL ENVIRONMENT*, *380*(1-3, SI), 66-74. doi: 10.1016/j.scitotenv.2006.09.012

---

## Author Response (AR1)

**UNIVERSITY OF HELSINKI**

**Dr. Xuefei Li**

[Figure]

Dr. Xuefei Li
*Institute for Atmospheric and*
*Earth System Research/Physics*
*Faculty of Science*
*University of Helsinki*
*P.O. Box 68, Helsinki*
*00014 Finland*
*Email: xuefei.z.li@helsinki.fi*
*Phone: +358 400 285 630*

March 10, 2020

Steven Bouillon, Dr
Associate Editor
Biogeosiences

Dear Dr. Bouillon,

With the constructive comments from both reviewers and the editor, we have now substantially revised the manuscript and are including 3 additional Figures (site description map, concentration of nutrients in the water and wavelet analysis of environmental variables), and included global warming potential in the analysis. We have discussed and quantified the uncertainties of using Cole-Caraco model in estimating air-water fluxes. We have also applied alternative gap-filling method without using dissolved gas concentration (which is now explained in the new Supplement Material).

Please find a point-by-point response to the reviewers' comments at the bottom of this letter noted in bold with the reviewer's original comments in blue. Special attention was paid to the three specific points underlined by the editor (data gap-filling, relative contribution of water-air gas exchange to total fluxes and the parameterization of diffusive fluxes across the water-air interface) in the response to review #2 (marked in red).

Thank you for considering our manuscript for your journal.

Sincerely,

Dr. Xuefei Li

Referee # 1:

"While there is a growing number of CO2 and CH4 studies from natural ecosystems, relatively few studies come from urban wetlands. Hence, this paper caught my attention as being a potentially important, new and novel contribution.
What does the term urban wetlands mean and why may greenhouse gas exchange to and from it differ from other wetlands? To my mind, I would expect urban wetlands to be recycling water from urban uses and be subject to runoff from urban landscapes, which may have elevated levels of N applications, herbicides, oil runoff from roads etc. So. these factors may affect the redox ladder and alter methane fluxes compared to those from more remote wetlands. Let's see what the authors find." I suspect the definition of an urban wetland is overly broad and more specification may be needed. In this case the authors are studying a constructed, stormwater wetland. I suspect there are many other types of urban wetlands, just look at the urban LTER in Baltimore, MD and Phoenix, AZ as a comparison. So building a database on how they may differ or be similar should be a long term goal, initated by a project like this. It would be nice to frame this urban wetland in Finland in context to those in wetter/drier and warmer worlds."

**We thank the reviewer for the effort spent on our manuscript and the appreciation of the importance of our study. We agree with the reviewer that the definition of an urban wetland is very broad. We rephrase it in the text as follows: "In this paper we present measurements carried out at a created urban wetland in Southern Finland in the boreal climate." (Line 22-23)**

"A limitation of this study is the time scale..
'The measurements were commenced the fourth year after construction and lasted for one full year and two subsequent growing seasons'. This study is missing many of the important pulses after construction to truly under- stand the dynamics of this system. This aspect is one of the greatest weaknesses of this work. But given so little data on this topic, I decided it is not a fatal flaw, in this instance. But I would not view future studies of this type that miss the dynamic of the restoration pulse viable."

**We are aware of the time scale of this study limited our capability to draw conclusions about the climate impact of the management (rewetting) when constructing an urban wetland. However, our study focused on the climate impact of the urban wetland after its establishment.**

"The authors report:
The annual NEE of the studied wetland was 8.0 g C-CO2 m-2 yr-1 with the 95% confidence interval between -18.9 and 34.9 g C-CO2m-2 yr-1 and FCH4 was 3.9 g C-CH4 m-2 yr-1 with the 95% confidence interval between 3.75 and 4.07 g CH4 m-2 yr-1.
I must admit I am surprised how tiny the fluxes are, given it is a wetland, even if in Finland. I would expect a stronger sink, but granted this would be conditional of what is in the flux footprint. So careful correspondence between fluxes and footprints are key to interpret these data.
As I read on I take home the key point that it is a weak sink for only 2 months and a slow C source rest of the year. Guess in hindsight it all makes sense. As I read the introduction, I am finding necessary conditional information. For example, open water is just not always open water. With N inputs there can be other life forms. Here the authors note
'At open-water surfaces, the net production of CO2 is a result of photosynthesis by algae, cyanobacteria as well as submerged aquatic plants, respiration of organic carbon and oxidation of CH4 produced in the water.'
This conditions meets with some of our experiences where we see azola and other aquatic plants in the open water sections. It has changed my perspective and open to this observation. The authors will need to be careful as they evaluate their 'open' water data and inform the reader if it is or not truly open water."

**In fact, we did not observe lots of metaphytic or filamentous algae during the study period. There was not large number of free floating small plants neither. There were some submerged aquatic plants which did not affect the openness of the water.**

"Glad to see citation to the work of the Estonian team of Mander et al, as they are among the few teams looking at this problem. I would also double check literature by Bill Mitsch. Their wetlands in Ohio may qualify as an urban wetland as it was close to the University in Columbus OH. Recent reports of methane fluxes come from Gil Bohrer's group, Morin et al and others.
Glad to see the authors are clued in about the key role of flux footprints. As we bend the rules of eddy covariance and ask contemporary questions and problems, we will need footprint models to partition the heterogeneity of the landscape."

**The two papers from Morin et al. were cited in the manuscript (now in Line 77 and 457).**

"Materials
The wetland is over 500 ha. This is a good size field for this work. Standard and well vetted eddy covariance is used by experts in the field who know how to carefully interpret the data. Closed path $CO_2$, Licor, and TDL is used to measure methane fluctuations. Given the cold, wet environment I think closed path is best for this work. The authors have looked at cospectra to ensure filtering is limited or appropriated corrected for. Good micromet protocol.
Standard neural networks are used to gap fill. The methods are described in great detail and proper attention to nodes, validation data, etc are made.
Overall I am confident about these measurements as this team has a long history of well vetted studies. The paper needs an assessment, map of the heterogeneous fetch and the flux footprint. I did not see this in the material. It is in the supplement, but it may be better placed in the paper. Starting to lose track of what is a paper vs supplement."

**The map of study site overlapped with climatological footprints is now shown in Figure 1 of the recised manuscript.**

"This paper is novel with water ch4 sensors to apply the diffusion model. First time I have seen these sensors. Bravo/brava/bravum.
The authors try to partition fluxes by the veg water fraction. I realize this is a legitmate quest and one with good intentions. We have tried this approach in the past and failed. We used multiple towers to close the system of equations with water/veg fractions. But my student, Jacyln Hatala Matthes found that the fractal dimension of the patches was key. So be careful in your partitioning.
Matthes, Jaclyn Hatala, et al. "Parsing the variability in CH4 flux at a spatially het- erogeneous wetland: Integrating multiple eddy covariance towers with high resolution flux footprint analysis." Journal of Geophysical Research: Biogeosciences,119.7 (2014): 1322-1339.
Use of the Kljun model is good. It has evolved as one of the better and most widely used.
With this the authors calculate the veg water fractions. But I must confess I don't have confidence in these numbers, especially from one tower. The reason we tried to use two towers was to get different fractions of water and vegetation with two equations and two unknowns.
I'd like to have the authors discuss the uncertainty more and critique the pros and cons of their method.
The reporting of flux reports is straight forward and standard. I have no critique or suggestions for this part."

The discussion about the uncertainty of our method is now added in the manuscript (L252-261): "The uncertainty of the vegetation and water fraction come from two sources. Firstly, the delineation of the distinct surface types was conducted based on a land surface map of the growing season in 2013, which neglected the change in the spatial extent of the vegetation throughout time. Secondly, although the footprint model used here is proved to be robust and general, there are uncertainties in the model prediction. To be more confident in the footprint estimation, it would be good to compare our results with large eddy simulation, however it is out of the scope of the current study. With only one EC tower we could not cross check the results as done in another study (Matthes, Sturtevant, Verfaillie, Knox, & Baldocchi, 2014).

However, we chose to follow a simple approach dividing the landscape into vegetation and open water because we did not observe significant vegetation expansion during the growing season and the area of open water is relatively constant. Furthermore, the clear effect of the footprint-weighted fraction of open water on the synchronization between EC $CH_4$ measurements and diffusive $CH_4$ flux from water was nicely demonstrated in our analysis (Fig. S4 in the new Supplement Material), so that we think the simple method used is sufficient to capture the major pattern in vegetation and water fraction in our study."

"What interests me is information on controls and processes. Here the paper has an advantage with measurements of the fluxes from the water section. But we have to be careful here. If the water is open then simple models will work. But with urban systems the N inputs can green up the water and the presence of green material will cause the diffusion models to be invalid. I need to hear more about this. So first confirm if the open water is open or is it clogged."

The open water is open (see the corresponding responses above). Submerged aquatic plants should not affect the validation of the diffusion model. Furthermore, the estimated diffusive fluxes of $CH_4$ and carbon dioxide $CO_2$ were well situated in the range of the diffusive gas fluxes over open lakes from other studies (Erkkila et al., 2018; Mammarella et al., 2015), which supported our assumption that the water was not covered by floating plants.

"The controls need a bit more information on N load of the water. What is the nitrate or phosphorus levels. If there is runoff P and NO3- may affect the CH4 fluxes."

The nutrient levels in the water are now added in Fig. 2 of the revised manuscript (Fig. 1 in this reply). Description of the measurements and nutrient levels is now added in the revised manuscript (L136-138 and L323-328).

"The control and process section is very simple and using correlations. I does not go into great enough detail and I am not sure if it makes a dent in our ignorance. I like methods using information theory at different time scales, I continue to worry about the roles of photosynthetic inputs to prime archaea. We learn that at different time scales temperature control may be dominant and photosynthesis may at others. Water table is important, but if it does not vary much it will not be a notable factor, yet we know mechanistically it is and if water table dropped below ground level one would see the effect."

We conducted wavelet coherence analysis to reveal the processes and environmental controls of the gases at different time scales. The magnitude of the wavelet coherence and the phase differences between ecosystem $CO_2$ and $CH_4$ fluxes and environmental variables are shown in Figs. 2 and 3 (figure numbers refer to the figures attached in this document, if not indicated specifically). Here we show the results of net ecosystem productivity (NEP; NEP= -NEE) instead of NEE for a better interpretation of the phase arrow (higher positive value in NEP means higher $CO_2$ uptake).

We found strong positive correlations between NEP and temperature, radiation at 1-day scale due to the diel temperature and radiation cycles. On average, $T_{air}$ and $T_{water}$ are leading NEP by ∼3h and ∼8h, respectively, while radiation is almost in-phase with NEP. The variation of TP is leading the change in NEP at 1-day scale (more TP leads to more $CO_2$ uptake) where the time lag varies between 1 to 5 hours (Fig.4 (d)).

$CH_4$ flux has correlation with temperature at 1-day scale where $T_{air}$ and $T_{water}$ are leading $CH_4$ flux by ∼1h and ∼6h, respectively (Fig. 5). $CH_4$ flux has also correlation with temperatures at 16-32-day scale (Fig. 3). Radiation is in-phase with $CH_4$ flux at 1-day scale (Fig. 4(c)). TP has positive correlation with $CH_4$ flux (more TP leads to more $CH_4$ emission) at 1-day scale and TP is leading $CH_4$ flux by ∼2h. Surprisingly, water level did not show any consistent correlation with $CH_4$ flux at any time scale which may be due to the small variation in water level during the growing season.

After all, it is worth noting that the correlations between the fluxes and environmental variables revealed by wavelet coherence analysis can be overstated, as much of the flux data has been gap-filled using these variables. Therefore, in the revised manuscript we only add figures which show the results between fluxes ($CO_2$ and $CH_4$) and those independent environmental variables, $NO_3$-N and TP (See Figure 7 in the revised manuscript).

"Glad to see the authors using sustained warming potential method of Neubauer and Megonigal. I just reviewed another wetland restoration paper and they Did NOT use this method and it was a criticism of mine Methane emissions are not a single pulse, like used with the old method. It is key to use a sustained emission method."

To be consistent with other references using IPCC value as reviewer # 2 suggested, we add also the results using the conventional global warming potential in Table 2 of the revised manuscript.

Discussion
"The authors do a nice job putting this work in context and reviewing the literature. I don't want to micromanage as there are many ways to go. I do like the discussion on O2 consumption. This is a nice angle and looks at mechanisms. I do like seeing a bit of advice on how best to design these systems. What are the pros and cons of different water/veg fractions and what can one do to minimize methane emissions or what are the effects of nutrient inputs on the greening of open water spaces."

We now add one more paragraph on the advices of designing urban wetland ecosystem in the revised manuscript (L511-521).

"In closing this paper has some novel aspects and I think it will merit publication. I do think it has some lingering issues that need to be resolved. Most seriously fraction of the water and vegetation and the modeling of fluxes from the water portion if the water is not pure. The other limitation is the time scale. It misses critical dynamic of the pulse and recovery after the wetland has been developed. This is a hole that cannot be filled."

We thank the reviewer for his constructive suggestions. We are aware of the limitation in our study and they are clearly acknowledged in the revised manuscript. Future studies are ought to be planned in a manner which can "fill the hole".

Referee # 2:

Major comments

"Li et al. report a data-set of CO2 and CH4 fluxes measured by eddy-covariance (EC) in an artificial wetland in Southern Finland. The topic of the study is to quantify air-water and air-vegetation CO2 and CH4 fluxes in wetlands which is very interesting as well as extremely challenging, and rarely investigated. However, the analysis relies heavily on data gap filling, and data are reconstructed up to $> 70\%$ for the first year and up to $> 50\%$ for the second year. I'm aware that there is commonly a very substantial data rejection for EC measurements, and that data filling is a common and accepted practice in studies of terrestrial ecosystem fluxes. However, in terrestrial ecosystem flux studies, data filling relies on relations that make sense such as primary production vs PAR and respiration vs temperature that are based on robust biological principles. Here, the authors used correlations with the dissolved CO2 concentration to data fill the EC CO2 fluxes, which does not necessary make sense specially for the air-vegetation fluxes (because some of the CO2 signal must come from hydrological input and is independent from wetland metabolism)."

We thank the reviewer for the time and effort used to our manuscript.
The wetland ecosystem in our study is comprised of both open water and vegetation surface type, both of which contribute simultaneously to the EC measurement. As the dictinct processes involved in each surface type, the relationships between environmental variables and EC fluxes are complicated. **To be confident in our results, we gap-filled the EC data using both an artificial neural network (ANN) model and parameterization based on biological principles (see below).** ANN is essentially a empirical non-linear regression model (Papale & Valentini, 2003), which is a data-based model rather than process-based models such as Michaelis-Menten light response function for photosynthesis. ANN is known for its capability of modelling complex relationships (Moffat et al., 2007; Richardson et al., 2008). The input parameters of the model are chosen to maximize the model accuracy in keeping with the principle of parsimony. The dissolved $CO_2$ and $CH_4$ concentrations are chosen in the model as they greatly increased the model precision (see Figure S2 in the new Supplement Material). This is also reasonable because a fraction of the flux measured by EC tower comes from the diffusive fluxes from the open water which is linked to gas concentration in the water.

"Furthermore, the authors use the CO2 concentration to compute the air- water CO2 fluxes that are then used in a more detailed analysis in conjunction with the EC CO2 fluxes to discuss the relative contribution of air-water and air-vegetation fluxes. So, the same variable (CO2 dissolved concentration) is used to compute two variables (air-water CO2 and EC CO2 fluxes) that are subsequently treated as independent, when they are obviously not. This, in my opinion, strongly weakens the analysis and conclusions of this study."

**To avoid the problem of computing two variables based on the same variable, we also gapfilled the EC data based on biological principles without using dissolved gas concentration (which is now shown in the new Supplement Material). We gapfilled the missing NEE using the following parameterization (Aurela et al., 2009):**

$$NEE = \frac{PI \times \alpha \times PPFD \times GP_{max}}{\alpha \times PPFD + GP_{max}} + R_0 \exp[E(\frac{1}{T_0} - \frac{1}{T_{air}+T_1})]$$

**where NEE is the net ecosystem $CO_2$ exchange, $GP_{max}$ is the gross photosynthesis rate in optimal light conditions, PI is an empirically determined effective phytomass index**

(Aurela, Tuovinen et al. 2001), $\alpha$ is the initial slope of NEE versus PPFD, $R_0$ is the rate of ecosystem respiration at $10°C$, E is a physiological parameter (in degree Kelvin), $T_{air}$ is the air temperature, T0 = 56.02 K, T1= 227.13 K (Lloyd & Taylor, 1994). PI was calculated by substracting the nighttime (PPFD < 20 $\mu$mol $^{-2}$ s$^{-1}$ ) respiration flux from the daytime (PPFD > 500 $\mu$mol $^{-2}$ s$^{-1}$ ) flux. The PI was calculated on a six-day basis. During summer time (day of year 90 - 283), E was determined by fitting the respiration to the nighttime data through the year, which was 342.24 K. $\alpha$, $GP_{max}$ and $R_0$ wre fitted in b-weekly periods: an $R_0$ value was first determined by fitting the respiration to the nighttime data for each of these periods, then the values of $\alpha$ and $GP_{max}$ were obtained by fitting the NEE equations to the daytime and nighttime data. During winter when no uptake of $CO_2$ was observed, the gaps were filled by a moving average with a 30-day window. At the beginning and end of the winter periods, the window was 8 days.

For gap-filling $CH_4$ flux in year 2013, we fitted the observed data points on the air and water temperature using exponential functions. To maximize the goodness of fit, the fitting was conducted separately for data points before and after day of the year 160 (detailed in the new Supplement Material).

Following this gap-filling method, the annual cumulative flux of NEE was 8.9 g C m$^{-2}$ and that of $CH_4$ was 3.8 g C g m$^{-2}$. They are not significantly different from the annual cumulative fluxes when ANN was used as gap-filling method in the manuscript, which were 8.0 g C m$^{-2}$ with the 95% confidence interval between -18.9 and 34.9 g C m$^{-2}$ for $CO_2$ and 3.9 g C m$^{-2}$ with the 95% confidence interval between 3.8 and 4.1 g C m$^{-2}$ for $CH_4$. Since the partitioning was only based on annual cumulative value, the contribution of different land cover is not changing within confidence interval.

"My other concern is that the air-water CO2 fluxes were computed from a gas transfer velocity parameterization, when it could have been relatively easy and inexpensive to measure it directly with floating domes. While it is not necessarily very constructive to point out what should have been measured, I have also some strong concerns on the choice of the parameterization. The gas transfer parameterization of Cole and Caraco (1998) was developed for large lakes, and is most probably inadequate for very small water bodies (such as the one in the present case) that usually have much lower gas transfer velocity values (Holgerson et al. 2017). The gas transfer velocity in small water bodies are even less constrained than in larger water bodies, and are bound to lead to a large source of uncertainty for computation of the fluxes that will propagate into the additional analysis based on these fluxes. Turbulence (hence gas transfer velocity) in small water bodies is mainly related to convection and less to wind speed (Holgerson et al. 2016), so wind speed based parameterizations are inadequate for small water bodies."

We are fully aware of the limitation of using Cole-Caraco parameterization to estimate air-water fluxes from small lake (discussed in Line 252- 261).

To quantify the potential uncertainty, we calculated the gas transfer velocity normalized to $CO_2$ at $20°C$ ($k_{600}$) using another model which takes heat flux into account (Heiskanen et al., 2014). We compared the $k_{600}$ derived from Cole-Caraco model ($k_{600CC}$) and from Heiskanen model ($k_{600He}$), where $k_{600CC}$ is 62% smaller than $k_{600He}$. However, applying Heiskanen model brought in large uncertainty as it requires net shortwave and longwave radiation data which we do not have measurements from the water body. The current calculation using Heiskanen model was based on the radiation data from a meteorological station located in the vegetation surface type, which does not fully reflect the conditions from water surface.

The water body in our study is located in an open area where the contribution of wind shear to the turbulence in the surface mixed layer is relatively high. During the study period in 2013, the average wind speed was 1.57 m s$^{-1}$ with a maximum of 7.1 m s$^{-1}$, much higher than the wind speed measured at ponds surrounded by the forest where the average values ranged from 0.28 to 0.35 m s$^{-1}$ with a maximum of 4.3 m s$^{-1}$ (Holgerson, Farr, & Raymond, 2017). Additionally, the $k_{600CC}$ estimated in our study was on average 0.66 m/day, well situated within the range of the $k_{600}$ directly measured by floating chmaber or gas tracer for small lakes and ponds (Holgerson et al., 2017). The estimated air-water fluxes of $CH_4$ and $CO_2$ based on the current model were also well within the range of the diffusive gas fluxes over small lakes from other studies (Erkkila et al., 2018; Mammarella et al., 2015). Finally, the parameterization of Cole and Caraco has been similarly applied to connected small open-water pools in a restored wetland which found reasonable agreement between the model estimation and the measurements (McNicol et al., 2017).

Considering all the above-mentioned reasons, we decide to continue using Cole-Caraco model to estimate diffusive fluxes from the water, bearing in mind that the calculated fluxes could be underestimated.

Minor comments

"L 51: What "UN report" ? Please provide a reference."

The reference has been added: United Nations, Department of Economic and Social Affairs, "Global Sustainable Development Report 2016", New York, July, 2016.

"L58: The Kyoto protocol is obsolete, we've moved on to the Paris Agreement."

We changed "Kyoto protocol" to "Paris Agreement".

"L62-66: Are these hypothetical or based on prior studies?"

1) is based on the knowledge of vegetation dynamics. We now spell it out in the text (L64-66): "When an urban wetland is newly created by rewetting the landscape, it takes time for the vegetation to establish itself in the new environment. The low coverage of vegetation at the initial phase of wetland establishment can lead to low $CO_2$ sequestration on a ecosystem scale." For 2), the high nutrient level in the receiving water into the urban wetland was observed by multiple studies. We added references to back up this statement (Lu et al., 2009; Vohla, Alas, Nurk, Baatz, & Mander, 2007; Valkama et al., 2017). And for 3), we agree with the reviewer that natural wetlands can also exhibit large spatial heterogeneity in vegetation and hydrology, thus we now removed this sentence.

"L 66: Does this mean you assume "spatial heterogeneity" of artificial wetlands to be stronger than natural ones ? Why ? Natural wetlands also have "different processes of production and transportation of GHGs"

We removed this sentence as mentioned above.

"L68: dissolved CO2 concentrations are usually orders of magnitude larger than CH4 concentrations, so CH4 oxidation plays a negligible role in the balance of production and uptake of CO2."

We removed the "oxidation of $CH_4$ produced in the water".

"L83: 'the situation are' "

We changed it to "the situations are ".

"L107: Might be useful to provide nutrient and chlorophyll levels to characterize the eutrophication of the lake."

The level of total phosphorus and $NO_3$-N are provided now in Fig. 2g and 2h of the revised manuscript. Chlorophyll level was not measured, unfortunately.

"L108: Please provide a reference."

The following references are now added to the manuscript:

Varis O, Sirvio H, Kettunen J. 1989. Multivariate analysis of lake phytoplankton and environmental factors. Arch Hydrobiol. 117:163-175.

Salonen V-P, Varjo E. 2000. Vihdin Enäjärven kunnostuksen vaikutus pohjasedimentin ominaisuuksiin [The effects of restoration actions at the Lake Enäjärvi in Vihti, Finland on bottom sediment characteristics]. Geologi. 52:159-163. Finnish.

"L201: Part of the Reco signal is due to hydrological input of CO2, and does not equate with ecosystem respiration."

We now removed the section of NEE partitioning.

L 236: A nine year old paper is not a 'recent study'. There are numerous other studies that show a disagreement between floating chamber and other methods, for instance Vachon et al. (2010). Conversely, there are numerous studies that report gas transfer velocities in lakes that diverge from the parameterization of Cole and Caraco (1998) such as Jonsson et al. (2008) and MacIntyre et al. (2010). This is particularly the case in small water bodies where turbulence is largely unrelated to wind (Holgerson et al. 2016)."

See the corresponding responses above. While we acknowledge that both wind shear and convection have significant contributions to turbulence in the surface mixed layer above small water bodies, but we think the current method is sufficient to capture the basic patterns in the diffusive fluxes.

"L240: The Fveg term also includes the CH4 ebullition component, however the fveg term for CH4 only corresponds to the vegetation, so when ebullition occurs (most of the time probably) the Fveg term is over-estimated."

We acknowledge that Fveg term can be over-estimated as we did not have independent measures for ebullition. We have discussed about it in the text as one of the potential uncertainties in our study (Line 492-495). However, due to the minor significance of ebullition found in other restored wetland, we think our ignorance of ebullition would not change much of the general conclusion of our study.

"L 262: This GWP value is much higher than the one proposed by the IPCC that is unanimously used. For consistency with the rest of the literature it could have been wiser to use the IPCC values."

We used sustained global warming potential with a 45 as the $CO_2$ equivalents of $CH_4$ fluxes (Neubauer & Megonigal, 2015). Because greenhouse gas emissions are not single pulses, it is reasonable to use a sustained emission method. But for an easier comparison with other studies, we now also calculate $CH_4$ fluxes as $CO_2$ equivalents using a global warming potential (GWP) of 34 following the 5th Assessment Report of IPCC (Myhre et al., 2013). The GWP of $CH_4$ fluxes from ecosystem, water and vegetation are 0.177, 0.077 and 0.195 kg $CO_2$-eq $m^{-2}$, and they are now added to the Table 2 of the revised manuscript.

"L 302: ppm unit in aquatic GHG literature relates to a partial pressure of CO2 and not the concentration of CO2 as stated."

ppm unit has been converted to µmol/L using Henry's law. The result is presented in Fig. 2(e) and L 313-317 of the revised manuscript.

[Figure]

Figure 1: The daily average of (a) rainfall, (b) total phosphorus concentration and (c) NO₃-N concentration measured at the outlet monitoring station in year 2013. The lake was covered by ice from January to March and it was free of ice after the end of March.

[Figure]

Figure 2: Wavelet coherence analysis and the phase difference between net ecosystem production (NEP; NEP=-NEE) and environmental controls from January to December 2013. The color represents the power of the coherence from 0 to 1. The phase difference is indicated by black arrows which only show up where the coherence is greater than or equal to 0.5. $\rightarrow$ indicates in-phase (two time series in synchrony) and arrows in other direction indicate out of phase (representing lags between time series), i.e. $\leftarrow$ indicates anti-phase, $\downarrow$ indicates the $1^{st}$ series (NEP) leads by quarter-cycle and $\uparrow$ indicates $2^{nd}$ series (environmental controls) leads by quarter-cycle. White dash contour lines indicate the cone of influence. PCO2, PCH4, NO3-N and TP indicate the concentrations of $CO_2$, $CH_4$, $NO_3$-N and total phosphorus in the water.

[Figure]

Figure 3: Wavelet coherence analysis and the phase difference between ecosystem $CH_4$ flux (FCH4) and environmental controls from January to December 2013. The color represents the power of the coherence from 0 to 1. The phase difference is indicated by black arrows which only show up where the coherence is greater than or equal to 0.5. $\rightarrow$ indicates in-phase (two time series in synchrony) and arrows in other direction indicate out of phase (representing lags between time series), i.e. $\leftarrow$ indicates anti-phase, $\downarrow$ indicates the $1^{st}$ series (FCH4) leads by quarter-cycle and $\uparrow$ indicates $2^{nd}$ series (environmental controls) leads by quarter-cycle. White dash contour lines indicate the cone of influence. PCO2, PCH4, NO3-N and TP indicate the concentrations of $CO_2$, $CH_4$, $NO_3$-N and total phosphorus in the water.

[Figure]

Figure 4: Time lag between NEP and (a) air temperature($T_{air}$), (b) water temperature ($T_{water}$), (c) radiation and (d) total phosphorus (TP) at 1-day time scale. Positive time lags indicate the environmental variables are leading NEP and vice versa.

[Figure]

Figure 5: Time lag between $CH_4$ flux ($F_{CH4}$) and (a) air temperature ($T_{air}$), (b) water temperature ($T_{water}$) and (c) radiation and (d) total phosphorus (TP) at 1-day time scale. Positive time lag indicate the environmental variables are leading ($F_{CH4}$) and vice versa.

[revised manuscript text omitted]

Reports on boreal wetlands, such as peatlands, have shown that large carbon storage remains in the soil due to anaerobic conditions limiting microbial decomposition, and thus offering a global cooling effect (Frolking et al., 2006). However, in newly constructed urban wetlands on mineral soil the gas exchange may be very different from natural wetlands: 1) T~t~he cooling effect of a wetland may be reduced or it becomes a source of carbon due to the early successional stage of the wetland. When an urban wetland is newly created by rewetting the landscape, it takes time for the vegetation to establish itself in the new environment. The low coverage of vegetation at the initial phase of wetland establishment can lead to low $CO_2$ sequestration on an ecosystem scale.~,~ 2) W~w~etlands 
[revised manuscript text omitted]

**2.2 Water and micrometeorological measurements**

Water monitoring stations were set up at the inlet (60.3283° N, 24.3356° E) and at the outlet (60.3281° N, 24.3377° E) of the wetland. During the 2012-2013 and 2013-2014 monitoring periods, water temperature as well as water turbidity, oxygen concentration, conductivity and pH were measured at the inlet and outlet monitoring station with the YSI-6000 series multiparameter sonde (YSI Inc., Yellow Springs, OH, USA). Measurements were conducted continuously with 10-minute interval. Water level at the outlet was measured continuously with a pressure gauge (STS sensor, Sensor Technik Sirnach AG, Switzerland). At the outlet monitoring station, the concentration of dissolved carbon dioxide ($[CO_2]$) and dissolved methane ($[CH_4]$) were measured with Contros HydroC™ $CO_2$ and HydroC™ $CH_4$ sensors (CONTROS Systems & Solutions GmbH, Germany). In 2014, the same sensors were also installed at the inlet monitoring station to measure $[CO_2]$ and $[CH_4]$. Dissolved $CO_2$ and $CH_4$ molecules diffuse from water column into the detection chamber through a thin-film composite membrane where the concentration of $CO_2$ and $CH_4$ is determined by means of IR absorption spectrometry and Tunable Diode Laser Absorption Spectroscopy, respectively. $NO_3$-N was measured with Scan sensors (Scan gmbh, Austria) and total phosphorus (TP) was calculated based on turbidity data which was measured at 10-min intervals (Valkama et al., 2017).

[revised manuscript text omitted]

The uncertainty of the vegetation and water fraction come from two sources. Firstly, the delineation of the distinct surface types was conducted based on a land surface map of the growing season in 2013, which neglected the change in the spatial extent of the vegetation throughout time. Secondly, -although the footprint model used here although Kljun model (Kljun,

Calanca, Rotach, & Schmid, 2015) is proved to be robust and general, there are uncertainties in the model predictions. To be more confident in the footprint estimation, it would be good to compare our results with footprint estimates based on large eddy simulations, however it is out of the scope of the current study. With only one EC tower we could not cross check the results as done in another study (Matthes et al., , Sturtevant, Verfaillie, Knox, & Baldocchi, 2014). However, we chose to follow a simple approach dividing the landscape into vegetation and open water because we did not observe significant vegetation expansion during the growing season and the area of open water is relatively constant. Furthermore, the clear effect of the footprint-weighted fraction of open water on the synchronization between EC CH$_4$ measurements
and diffusive CH$_4$  flux from water  was nicely demonstrated
in our analysis (Fig. S4).

[revised manuscript text omitted]

The median concentration of total phosphorus measured at the outflow monitoring station was 56
$\mu g\ L^{-1}$ and the median $NO_3$-N concentration was 0.69 mg $L^{-1}$ in year 2013 (Fig. 2g, 2  h). In the annual perspective,
TP and  $NO_3$-N concentration consisted of several runoff peaks occurring after rain or snow melting events. This
wetland serves as a nutrient removal measure as it improved water quality by retaining P and N from runoff before the
release to the receiving lake, where the annual TP reduction was 13% and $NO_3$-N  reduction was 14% from the
original concentration in year 2014 (Valkama et al., 2017).

[revised manuscript text omitted]
 vegetation area in an urban wetland in Finland. Thus, if considering only the climate impact, it would be advisable to have lower water/vegetation fraction which means limiting open-water surfaces and setting a design preference for areas of emergent vegetation in the establishment of urban wetlands. Secondly, our Our results also showed that total phosphorus enhanced both $CO_2$ CO2 uptake and $CH_4$ CH4 emission which have contradictory climate impacts to the ecosystem (Fig. 7b, 7d). Although it is out of the scope of our study, it would be very interesting to understand the mechanisms, to quantify the magnitude and the duration of these enhancements induced by nutrient input. Previous studies have found that nutrien nutrient t inputs can influence the identity of the key primary producer (submerged plants versus phytoplankton) in the water, which is crucial in shaping the $CH_4$ CH4 emission from shallow water (West et al., , Creamer, & Jones, 2016; ; Davidson et al., 2018). Submerged plants may decrease $CH_4$ CH4 production in the lake by producing alleochemicals, transporting oxygen to the sediment and providing good habitat for $CH_4$ CH4 oxidizing bacteria (Heilman & Carlton, 2001), while phytoplankton was shown to significantly increase $CH_4$ CH4 ebullition by changing the quality of the dissolved organic carbon which promotes methanogenesis (West et al., 2016) or/and by altering the sediment texture and redox conditions favoring the release of bubbles. As a result, we suggest to control the nutrient input to the water of the newly established wetland to limit the abundance of phytoplankton as well as to support the existence of submerged plants.

**5 Conclusions**

Urban wetlands have received global attention as a nature-based urban runoff management solution for sustainable cities, as they provide cost efficient flood control and water quality mitigation as well as many ecological and cultural services. In the meantime, the climate impact of urban wetlands should also be considered. Wetting a landscape may enhance the $CO_2$ sequestration in the ecosystem, whereas $CH_4$ can be emitted due to the anaerobic conditions in the soil after wetting. Furthermore, heterogeneity induced in newly created urban wetland may contribute differently to the overall climate impact.

In the present study, for the first time a full annual carbon balance of an urban stormwater wetland in the boreal region was evaluated and the radiative forcing from heterogeneous landscapes were presented. We found that, during the monitored period at the study wetland, both the open water area and the vegetated area within the created wetland were carbon sources, and thus the urban wetland had a net climate warming effect, the monitored fourth year after the wetland establishment. The radiative forcing effect of the open-water area exceeded the vegetated area, which indicated that limiting open-water surfaces and setting a design preference for areas of emergent vegetation in the establishment of urban wetlands can be a beneficial practice when considering only the climate impact of a created urban wetland. In the meanwhile, we also emphasize that the value of urban wetlands should not be determined solely by GHG radiative forcing.

The values of urban wetlands in other areas e.g. flood control, pollutant removal, biodiversity, recreation and education are as well of paramount importance to human society.

**Data availability**
Eddy covariance, gas concentration and meteorological data are available from the DRYAD database at https://datadryad.org/stash/share/WrtTNnpIt6FgLoMSZ_Wlr0lK22IcxqjGZAStuuKdHLs

**Author contribution**
IM, OW, HV, AO and TV designed the field study. SH, IM and JP carried out eddy covariance measurements, automatic gas concentration measurements in the open water and manual field measurements. XL and IM participated in eddy covariance data processing and analysis. XL analysed the results and prepared the manuscript with contributions from all co-authors.

**Competing interests**
The authors declare that they have no conflict of interest.

**Acknowledgments**

We thank Mikko Yli-Rosti and Kiril Aspila for assistance for the maintenance of the field measurements. This research was supported by the EU Life+11 ENV/FI/911 Urban Oases project grant, Academy of Finland, Academy Professor projects (312571 and 282842), ICOS-Finland (by Academy of Finland 281255 and University of Helsinki), the Maa- ja vesitekniikan tuki ry, the Ministry of the Environment of Finland and the Municipality of Vihti. In memoriam: The greenhouse gas exchange measurements at the Gateway Wetland were made possible due to the creative and caring support by late Professor Eero Nikinmaa to the Urban Oases project.

**Tables**

Table 1. Pearson correlation coefficient ($r$) between the daily averages of environmental variables and fluxes in year

2013 and 2014. NEE – net ecosystem exchange; $T_{air}$ – air temperature; $T_{water}$ – water temperature; PPFD –

photosynthetic photon flux density; WL – water level; $[CO_2]$ and $[CH_4]$ – $CO_2$ and $CH_4$ concentration measured in the outlet; * indicates only peak growing season (June, July and August) are included in the analysis.

| Flux | Year | $T_{air}$ | $T_{water}$ | PPFD | WL | $[CO_2]$ | $[CH_4]$ |
|------|------|-----------|-------------|------|-----|----------|----------|
| $CO_2$ | 2013 | -0.45 | -0.61 | -0.62 | 0.46 | -0.34 | 0.18 |
|      | 2014 | 0.43 | 0.54 | -0.12 | 0.12 | -0.12 | -0.05 |
| $CH_4$ | 2013 | 0.61 | 0.65 | 0.56 | -0.3 | 0.17 | -0.09 |
|      | 2014* | 0.37 | 0.26 | 0.27 | -0.24 | 0.28 | 0.25 |

Table 2. Annual $CO_2$ and $CH_4$ exchange from different surface zones,  their sustained global warming potential (SGWP) and global warming potential (GWP). Ecosystem, water and vegetation represent flux,  SGWP and GWP measured or calculated from the ecosystem by EC tower, from open water and from vegetated area. The numbers in the square bracket represent the 95% confidence interval of the average. No error bounds are reported for flux,  SGWP and GWP from water as they are modelled using gas concentration in the water and meteorological measurements.

| | | Ecosystem | Water | Vegetation |
|---|---|---|---|---|
| Flux (g C m$^{-2}$) | $CO_2$ | 8 [-18.9, 34.9] | 297.5 | -39.5 [-70.8, -8.1] |
| | $CH_4$ | 3.9 [3.8, 4.1] | 1.7 | 4.3 [4.1, 4.5] |
| SGWP (kg $CO_2$-eq m$^{-2}$) | $CO_2$ | 0.029 [-0.069, 0.128] | 1.090 | -0.145 [-0.260, -0.030] |
| | $CH_4$ | 0.234 [0.225, 0.244] | 0.104 | 0.256 [0.246, 0.268] |
| GWP (kg $CO_2$-eq m$^{-2}$) | $CH_4$ | 0.177 [0.170, 0.185] | 0.077 | 0.195 [0.187, 0.204] |

**Figures**

[Figure]

[Figure]

Figure 1:- The landscape classification of Nummela wetland. Wetland subareas specified according to mean water level are shown with different colors. The arrows indicate the direction of water flow. The black dots indicate the inflow and outflow measuring station and the location of eddy covariance tower.

[Figure]

[Figure]

Figure 1̶2: The daily-average of (a) photosynthetic photon flux density (PPFD), (b) air and water temperature (T_air and

T_water), (c) water level, (d) rainfall, (e) CO₂ C̶O̶₂̶ concentration ([CO₂]), a̶n̶d̶ (f) CH₄ concentration ([CH₄]), (g)

concentration of total phosphorus (TP) and (h) concentration of NO₃-N of f̶r̶o̶m̶ ̶i̶n̶l̶e̶t̶ ̶a̶n̶d̶ ̶o̶u̶t̶l̶e̶t̶ ̶o̶f̶ Nummela wetland from January 2013 to August 2014. T̶h̶e̶ ̶g̶r̶e̶y̶ ̶z̶o̶n̶e̶ ̶i̶n̶d̶i̶c̶a̶t̶e̶s̶ ̶t̶h̶e̶ ̶i̶c̶e̶-̶c̶o̶v̶e̶r̶e̶d̶ ̶p̶e̶r̶i̶o̶d̶.̶

[Figure]

Figure 32: Daily average of net ecosystem exchange of $CO_2$ (NEE, µmol m$^{-2}$ s$^{-1}$) in year 2013. Filled dots indicate measurement (wen available half-hourly measurement data ⩾ 10) and circles indicate gap-filled data (when available half-hourly measurement data < 10). The insert shows cumulative NEE (g C m$^{-2}$) in the ecosystem and the red line indicates the zero reference line. Error bounds (marked in grey) on cumulative NEE reflect the 95 % confidence interval for the gap-filling procedure.

[Figure]

Figure 34: Daily average of ecosystem CH₄ flux measured by EC tower and cumulative CH₄ emission in year 2013. Filled
dots indicate measurement (when available half-hourly measurement data ≥ 10) and circles indicate gap-filled data (when
available half-hourly measurement data < 10). The insert shows cumulative CH₄ emission with the error bounds in grey
reflecting the 95 % confidence interval for the gap-filling procedure.

[Figure]

Figure 54: Monthly-average of (a) diffusive CO₂ and (b) CH₄ flux from the open-water in year 2013. Error bar
indicates the standard error of the mean. From January to March there was ice-covered period.

[Figure]

Figure 6̲5̶: Mean diel pattern of the half-hourly net $CO_2$ and $CH_4$ fluxes in summer ((a) and (c)) and in winter ((b) and (d)). The dashed lines represent the standard deviation. Red lines indicate measurement from EC tower and the blue lines show the fluxes modelled for vegetated area.

[Figure]

Figure 7: Wavelet coherence analysis and the phase difference between ecosystem fluxes, the net ecosystem exchange
(NEE) and the CH$_4$ flux (FCH4), and nutrient concentration in the water, NO$_3$-N and total phosphorus from January to
December 2013. The color represents the power of the coherence from 0 to 1. The phase difference is indicated by black
arrows which only show up where the coherence is greater than or equal to 0.5. → indicates in-phase (two time series in
synchrony) and arrows in other direction indicate out of phase (representing lags between time series), i.e. ← indicates
anti-phase, ↓ indicates the 1$^{st}$ series (fluxes) leads by quarter-cycle and ↑ indicates 2$^{nd}$ series (NO$_3$-N and total
phosphorus) leads by quarter-cycle. White dash contour lines indicate the cone of influence.

---

## Author Response (AR2)

**Dr. Xuefei Li**

Dr. Xuefei Li Institute for Atmospheric and Earth System Research/Physics Faculty of Science University of Helsinki P.O. Box 68, Helsinki 00014 Finland Email: xuefei.z.li@helsinki.fi Phone: +358 400 285 630

May 22, 2020

Steven Bouillon, Dr Associate Editor Biogeosiences

Dear Dr. Bouillon,

We would like to thank you for your positive decision for our manuscript.

We have modified the manuscript according to your suggestions. Please find a point-by-point response at the bottom of this letter noted in bold with the editor's original comments in blue.

Thank you for considering our manuscript for your journal.

Sincerely,

Dr. Xuefei Li

Comments to the Author:

"Dear dr Li,

Thank you for your detailed reply to both referee reports, and for your revised version. Review report #1 was favorable to start with, and I feel their suggestions and points of concern were well addressed. For the 2nd review report, most of the concerns have been incorporated to the extent possible, e.g. the issue of gap filling is addressed in the revised version with the additional information provided in the supplement; this is transparent and the reader can make their own appreciation of the merits and limitations. I do feel that some more notes on the gas transfer velocity parameterization. I understand your rationale to stick with the Cole Caraco model, but feel that a summary of the information you provide in your author replies would be worth including in the manuscript itself; this will strengthen the rationale to use their model and at the same time provide some more background to the reader what the limitations are and why these were not straightforward to address with your dataset"

We now explained in more details on the rationale of using Cole & Caraco model in the manuscript (Line 225-239)

"Two small additional notes from my side: -the total phosphorus data: you mention these were "calculated based on turbidity data" and refer to Valkama et al. (2017). I would suggest to replace 'calculated' by 'estimated"'

We changed 'calculated' by 'estimated' (Line 140).

"I assume the data you use are in fact the same as those of the latter publication, and the same holds for the nitrate data? If that is the case: avoid confusion and make it clear that these nitrate and TP data were published earlier and were taken from Valkama et al. (2017). "

While the method of estimating phosphorus using turbidity data has been explained in Valkama et al.(2017), the data was not taken from that paper. But these data was published previously in Wahlroos et al. (2015). It has been now clarified in the manuscript (Lines 138-140).

"Figure 7 caption: subscript in 'FCH4""

'FCH4' has been changed to  $F_{\rm CH4}$  in Figure 7 and its captions.

| 1           | Carbon dioxide and methane fluxes from different surface types in a created                                                                                                                                                                | Formatted: Font: 14 pt                                |
|-------------|--------------------------------------------------------------------------------------------------------------------------------------------------------------------------------------------------------------------------------------------|-------------------------------------------------------|
| 2           | urban wetland                                                                                                                                                                                                                              |                                                       |
| 3
| Xuefei Li 1 , Outi Wahlroos 2 , Sami Haapanala 3 , Jukka Pumpanen 4 , Harri Vasander 5 , Anne Ojala 1, 5, 6 Timo Vesala 1, 5 and Ivan Mammarella 1 | Formatted: Font: 10 pt                                |
| 7           | 1 Institute for Atmospheric and Earth System Research (INAR)/Physics, Faculty of Science, University of Helsinki, P.O.                                                                                                          |                                                       |
| 8           | Box 68, 00014_ <del>University of</del> Helsinki, Finland                                                                                                                                                                                  |                                                       |
| 9           | 2 Palustrine Design Oy, Finland 2 Palustrine Design Oy, Poste Restante, Inkoo, Finland                                                                                                                               |
| 10          | University of Turku, Turku, Finland                                                                                                                                                                                                        | Formatted: Swedish (Sweden)                           |
| 11          | 3 Suvilumi Oy, Ohrahuhdantie 2 B, 00680 Helsinki, Finland                                                                                                                                                                       |                                                       |
| 12          | 4 Department of Environmental and Biological Sciences, University of Eastern Finland, P.O. Box 1627, 70211 Kuopio,                                                                                                              |                                                       |
| 13          | Finland                                                                                                                                                                                                                                    |                                                       |
| 14          | 5 Institute for Atmospheric and Earth System Research (INAR)/Forest Sciences, Faculty of Agriculture and Forestry,                                                                                                              |                                                       |
| 15          | University of Helsinki, P.O. Box 27, 00014 University of Helsinki, Finland                                                                                                                                                                 |                                                       |
| 16          | 6 Ecosystems and Environment Research Programme, Faculty of Biological and Environmental Sciences, University of                                                                                                                |                                                       |
| 17          | Helsinki, P.O. Box 65, 00014 University of Helsinki, Finland                                                                                                                                                                               |                                                       |
| 18          | Correspondence to: Dr. Xuefei Li (xuefei.z.li@helsinki.fi)                                                                                                                                                                                 |                                                       |
| 19          |                                                                                                                                                                                                                                            | Formatted: Font: (Default) Times New Roman, 10 pt     |
|             |                                                                                                                                                                                                                                            |                                                       |

Abstract. Many wetlands have been drained due to urbanization, agriculture, forestry or other purposes, which has resulted in losing their ecosystem services. To protect receiving waters and to achieve services such as flood control and stormwater quality mitigation, new wetlands are created in urbanized areas. However, our knowledge of greenhouse gas exchange in newly created wetlands in urban areas is currently limited. In this paper we present measurements carried out at a created urban wetland in Southern Finland in the boreal climate.

We conducted measurements of ecosystem CO2 flux (NEE) and CH4 flux (FCH4) at the constructed created stormwater 26 wetland Gateway in Nummela, Vihti, Southern Finland using eddy covariance (EC) technique. The measurements were 27 commenced the fourth year after construction and lasted for one full year and two subsequent growing seasons. Besides 28 ecosystem scale fluxes measured by the EC tower, the diffusive CO2 and CH4 fluxes from the open-water areaareas 29 (Fw\_CO2 and Fw\_CH4 respectively) were modelled based on measurements of CO2 and CH4 concentration in the water. 30 Fluxes from the vegetated areas were estimated by applying a simple mixing model using the above-mentioned fluxes 31 and the footprint-weighted fractional area. The half-hourly footprint-weighted contribution of diffusive fluxes from open 32 water ranged from 0 to 25.5 % in year 2013.

The annual NEE of the studied wetland was 8.0 g C-CO2 m-2 yr-1 with the 95 % confidence interval between -18.9 and 34 34.9 g C-CO2 m-2 yr-1 and FCH4 was 3.9 g C-CH4 m-2 yr-1 with the 95 % confidence interval between 3.75 and 4.07 g C-35 CH4 m-2 yr-1. The ecosystem sequestered CO2 during summer months (June-August), while the rest of the year it was a CO2 source. CH4 displayed strong seasonal dynamics, higher in summer and lower in winter, with a sporadic emission 36 37 episode in the end of May 2013. Both CH4 and CO2 fluxes, especially those obtained from vegetated areas, exhibited 38 strong diurnal cycles during summer with synchronized peaks around noon. The annual Fw\_CO2 was 297.5 g C-CO2 m-2 39 yr-1 and Fw\_CH4 was 1.73 g C-CH4 m-2 yr-1. The peak diffusive CH4 flux was 137.6 nmol C-CH4 m-2 s-1, which was 40 synchronized with the FCH4.

Overall, during the monitored time period, the established stormwater wetland had a climate warming effect with 0.263 42 kg CO2-eq m-2 yr-1 of which 89 % was contributed by CH4. The radiative forcing of the open-water areas exceeded that of 43 thee vegetation areas (1.194 kg CO2-eq m-2 yr-1 and 0.111 kg CO2-eq m-2 yr-1, respectively), which implies that, when 44 considering solely the climate impact of a created wetland over a 100-year horizon, it would be more beneficial to design 45 and establish wetlands with large patches of emergent vegetation, and to limit the areas of open-water to the minimum 46 necessitated by other desired ecosystem services.

**47 1 Introduction**

Wetlands provide many beneficial ecosystem services such as flood control and water quality mitigation, natural habitat 49 for flora and fauna and recreational opportunities (Mitsch and Gosselink, 2015). Many wetlands have been drained 50 globally for agriculture, forestry and other purposes including urbanization at the cost of losing wetland ecosystem 51 services (Vasander et al., 2003). Migration from rural area to cities will increase in even greater numbers in the near 52 future, and the United Nations report (United Nations, 2016)UN 2016 report has predicted that 75 % of the world 53 population will be living in cities by 2030. There is an urgent need for more sustainable urbanism and one effective 54 measure is to create functional and connected wetland networks in cities [Lucas et al., 2015;]\_Mungasavalli and 55 Viraraghavan, 2006),

Wetlands can take up carbon dioxide (CO2) through emergent and submerged vegetation but they are also important sources of methane (CH4), a greenhouse gas more potent than CO2 when considered over a 100-year horizon (Stocker et

| Formatt            | ed: Font: (Default)                                               | Times New                           | Roman, 10 pt                                 |  |
|--------------------|-------------------------------------------------------------------|-------------------------------------|----------------------------------------------|--|
|                    |                                                                   |                                     |                                              |  |
| Formatt            | e d: Font: (Default)                                       | Times New                           | Roman, 10 pt                                 |  |
|                    |                                                                   |                                     |                                              |  |
|                    |                                                                   |                                     |                                              |  |
|                    |                                                                   |                                     |                                              |  |
| Formatt            | ed: Font: (Default)                                               | Times New                           | Roman, 10 pt                                 |  |
| Formatt            | ed: Font: (Default)
ed: Font: (Default)                        | Times New
Times New              | Roman, 10 pt
Roman, 10 pt                 |  |
| Formatt
Formatt | r ed: Font: (Default)
r ed: Font: (Default)      | Times New
Times New              | Roman, 10 pt
Roman, 10 pt                 |  |
| Formatt
Formatt | ed: Font: (Default)
ed: Font: (Default)
ed: Font: (Default) | Times New
Times New
Times New | Roman, 10 pt
Roman, 10 pt
Roman, 10 pt |  |

| 58                                                                                                                         | al., 2014). The exchange of greenhouse gases (GHG) such as $\mathrm{CO}_2$ and $\mathrm{CH}_4$ between atmosphere and ecosystem have                                                                                                                                                                                                                                                                                                                                                                                                                                                                                                                                                                                                                                                                                                                                                                                                                                                                                                                                                                                                                                                                                                                                                                                                                                                                                                                                                                                                                                                                                                                                                                                                                                                                                                                                                                                                                                                                                                                                                                                                                                                                                                                                                                             |                                                                                                                                                                                                                                                                                                                                                                                                                                                                         |
|----------------------------------------------------------------------------------------------------------------------------|------------------------------------------------------------------------------------------------------------------------------------------------------------------------------------------------------------------------------------------------------------------------------------------------------------------------------------------------------------------------------------------------------------------------------------------------------------------------------------------------------------------------------------------------------------------------------------------------------------------------------------------------------------------------------------------------------------------------------------------------------------------------------------------------------------------------------------------------------------------------------------------------------------------------------------------------------------------------------------------------------------------------------------------------------------------------------------------------------------------------------------------------------------------------------------------------------------------------------------------------------------------------------------------------------------------------------------------------------------------------------------------------------------------------------------------------------------------------------------------------------------------------------------------------------------------------------------------------------------------------------------------------------------------------------------------------------------------------------------------------------------------------------------------------------------------------------------------------------------------------------------------------------------------------------------------------------------------------------------------------------------------------------------------------------------------------------------------------------------------------------------------------------------------------------------------------------------------------------------------------------------------------------------------------------------------|-------------------------------------------------------------------------------------------------------------------------------------------------------------------------------------------------------------------------------------------------------------------------------------------------------------------------------------------------------------------------------------------------------------------------------------------------------------------------|
| 59                                                                                                                         | direct influence on the atmospheric concentration of these gases, thus besides the ecosystem services that wetland $\underline{s}$                                                                                                                                                                                                                                                                                                                                                                                                                                                                                                                                                                                                                                                                                                                                                                                                                                                                                                                                                                                                                                                                                                                                                                                                                                                                                                                                                                                                                                                                                                                                                                                                                                                                                                                                                                                                                                                                                                                                                                                                                                                                                                                                                                               |                                                                                                                                                                                                                                                                                                                                                                                                                                                                         |
| 60                                                                                                                         | provide, the GHG budget of constructed wetlands should be accounted for according to international agreements such as                                                                                                                                                                                                                                                                                                                                                                                                                                                                                                                                                                                                                                                                                                                                                                                                                                                                                                                                                                                                                                                                                                                                                                                                                                                                                                                                                                                                                                                                                                                                                                                                                                                                                                                                                                                                                                                                                                                                                                                                                                                                                                                                                                                            |                                                                                                                                                                                                                                                                                                                                                                                                                                                                         |
| 61                                                                                                                         | the Kyoto-Paris Agreementprotocol                                                                                                                                                                                                                                                                                                                                                                                                                                                                                                                                                                                                                                                                                                                                                                                                                                                                                                                                                                                                                                                                                                                                                                                                                                                                                                                                                                                                                                                                                                                                                                                                                                                                                                                                                                                                                                                                                                                                                                                                                                                                                                                                                                                                                                                                                | Formatted: Font: (Default) Times New Roman, 10 pt                                                                                                                                                                                                                                                                                                                                                                                                                       |
| 62                                                                                                                         | Reports on boreal wetlands, such as peatlands, have shown that large carbon storage remains in the soil due to anaerobic                                                                                                                                                                                                                                                                                                                                                                                                                                                                                                                                                                                                                                                                                                                                                                                                                                                                                                                                                                                                                                                                                                                                                                                                                                                                                                                                                                                                                                                                                                                                                                                                                                                                                                                                                                                                                                                                                                                                                                                                                                                                                                                                                                                         |                                                                                                                                                                                                                                                                                                                                                                                                                                                                         |
| 63                                                                                                                         | conditions limiting microbial decomposition, and thus offering a global cooling effect (Frolking et al., 2006). However,                                                                                                                                                                                                                                                                                                                                                                                                                                                                                                                                                                                                                                                                                                                                                                                                                                                                                                                                                                                                                                                                                                                                                                                                                                                                                                                                                                                                                                                                                                                                                                                                                                                                                                                                                                                                                                                                                                                                                                                                                                                                                                                                                                                         | Formatted: Font: (Default) Times New Roman 10 nt                                                                                                                                                                                                                                                                                                                                                                                                                        |
| 64                                                                                                                         | in newly constructed urban wetlands on mineral soil the gas exchange may be very different from natural wetlands: 1)                                                                                                                                                                                                                                                                                                                                                                                                                                                                                                                                                                                                                                                                                                                                                                                                                                                                                                                                                                                                                                                                                                                                                                                                                                                                                                                                                                                                                                                                                                                                                                                                                                                                                                                                                                                                                                                                                                                                                                                                                                                                                                                                                                                             |                                                                                                                                                                                                                                                                                                                                                                                                                                                                         |
| 65                                                                                                                         | Tthe cooling effect of a wetland may be reduced or it becomes a source of carbon due to the early successional stage of                                                                                                                                                                                                                                                                                                                                                                                                                                                                                                                                                                                                                                                                                                                                                                                                                                                                                                                                                                                                                                                                                                                                                                                                                                                                                                                                                                                                                                                                                                                                                                                                                                                                                                                                                                                                                                                                                                                                                                                                                                                                                                                                                                                          |                                                                                                                                                                                                                                                                                                                                                                                                                                                                         |
| 66                                                                                                                         | the wetland. When an urban wetland is newly created by rewetting the a landscape, it takes time for the vegetation to                                                                                                                                                                                                                                                                                                                                                                                                                                                                                                                                                                                                                                                                                                                                                                                                                                                                                                                                                                                                                                                                                                                                                                                                                                                                                                                                                                                                                                                                                                                                                                                                                                                                                                                                                                                                                                                                                                                                                                                                                                                                                                                                                                                            |                                                                                                                                                                                                                                                                                                                                                                                                                                                                         |
| 67                                                                                                                         | establish itself in the new environment. The low coverage of vegetation at the initial phase of wetland establishment can                                                                                                                                                                                                                                                                                                                                                                                                                                                                                                                                                                                                                                                                                                                                                                                                                                                                                                                                                                                                                                                                                                                                                                                                                                                                                                                                                                                                                                                                                                                                                                                                                                                                                                                                                                                                                                                                                                                                                                                                                                                                                                                                                                                        |                                                                                                                                                                                                                                                                                                                                                                                                                                                                         |
| 68                                                                                                                         | lead to low CO2 sequestration on an ecosystem scale., 2) Wwetlands in close proximity to urban centers receive a                                                                                                                                                                                                                                                                                                                                                                                                                                                                                                                                                                                                                                                                                                                                                                                                                                                                                                                                                                                                                                                                                                                                                                                                                                                                                                                                                                                                                                                                                                                                                                                                                                                                                                                                                                                                                                                                                                                                                                                                                                                                                                                                                                                                 | Formatted: Subscript                                                                                                                                                                                                                                                                                                                                                                                                                                                    |
| 69                                                                                                                         | significant amount of nutrients and dissolved organic carbon from runoff (Lu et al., 2009; Vohla et al., 2007; Valkama et                                                                                                                                                                                                                                                                                                                                                                                                                                                                                                                                                                                                                                                                                                                                                                                                                                                                                                                                                                                                                                                                                                                                                                                                                                                                                                                                                                                                                                                                                                                                                                                                                                                                                                                                                                                                                                                                                                                                                                                                                                                                                                                                                                                        |                                                                                                                                                                                                                                                                                                                                                                                                                                                                         |
| 70                                                                                                                         | al., 2017) and 3) urban wetlands exhibit high spatial heterogeneity and hydrology where different processes of the                                                                                                                                                                                                                                                                                                                                                                                                                                                                                                                                                                                                                                                                                                                                                                                                                                                                                                                                                                                                                                                                                                                                                                                                                                                                                                                                                                                                                                                                                                                                                                                                                                                                                                                                                                                                                                                                                                                                                                                                                                                                                                                                                                                               |                                                                                                                                                                                                                                                                                                                                                                                                                                                                         |
| 71                                                                                                                         | production and transportation of GHG are involved. At the areas with emergent vegetation, CO 2 is absorbed by                                                                                                                                                                                                                                                                                                                                                                                                                                                                                                                                                                                                                                                                                                                                                                                                                                                                                                                                                                                                                                                                                                                                                                                                                                                                                                                                                                                                                                                                                                                                                                                                                                                                                                                                                                                                                                                                                                                                                                                                                                                                                                                                                                                         |                                                                                                                                                                                                                                                                                                                                                                                                                                                                         |
| 72                                                                                                                         | photosynthetic activity during daytime and growing season and is released through respirational processes. At open-water                                                                                                                                                                                                                                                                                                                                                                                                                                                                                                                                                                                                                                                                                                                                                                                                                                                                                                                                                                                                                                                                                                                                                                                                                                                                                                                                                                                                                                                                                                                                                                                                                                                                                                                                                                                                                                                                                                                                                                                                                                                                                                                                                                                         |                                                                                                                                                                                                                                                                                                                                                                                                                                                                         |
| 73                                                                                                                         | surfaces, the net production of CO 2 is a result of photosynthesis by algae, cyanobacteria as well as submerged aquatic                                                                                                                                                                                                                                                                                                                                                                                                                                                                                                                                                                                                                                                                                                                                                                                                                                                                                                                                                                                                                                                                                                                                                                                                                                                                                                                                                                                                                                                                                                                                                                                                                                                                                                                                                                                                                                                                                                                                                                                                                                                                                                                                                                               |                                                                                                                                                                                                                                                                                                                                                                                                                                                                         |
| 74                                                                                                                         | plants, respiration of organic carbon-and oxidation of CH4 produced in the water. When the CO2 concentration in the                                                                                                                                                                                                                                                                                                                                                                                                                                                                                                                                                                                                                                                                                                                                                                                                                                                                                                                                                                                                                                                                                                                                                                                                                                                                                                                                                                                                                                                                                                                                                                                                                                                                                                                                                                                                                                                                                                                                                                                                                                                                                                                                                                                              |                                                                                                                                                                                                                                                                                                                                                                                                                                                                         |
| 75                                                                                                                         | water exceeds atmospheric equilibrium, the surface becomes a source of CO2. CH4 can be produced through anaerobic                                                                                                                                                                                                                                                                                                                                                                                                                                                                                                                                                                                                                                                                                                                                                                                                                                                                                                                                                                                                                                                                                                                                                                                                                                                                                                                                                                                                                                                                                                                                                                                                                                                                                                                                                                                                                                                                                                                                                                                                                                                                                                                                                                                                |                                                                                                                                                                                                                                                                                                                                                                                                                                                                         |
| 76                                                                                                                         | metabolism in wetland soil and can be transported to the atmosphere by plant-mediated pathway through aerenchyma,                                                                                                                                                                                                                                                                                                                                                                                                                                                                                                                                                                                                                                                                                                                                                                                                                                                                                                                                                                                                                                                                                                                                                                                                                                                                                                                                                                                                                                                                                                                                                                                                                                                                                                                                                                                                                                                                                                                                                                                                                                                                                                                                                                                                |                                                                                                                                                                                                                                                                                                                                                                                                                                                                         |
| 77                                                                                                                         | sediment ebullition and diffusive fluxes at the water-atmosphere interface. In open water, the transport is dominated by                                                                                                                                                                                                                                                                                                                                                                                                                                                                                                                                                                                                                                                                                                                                                                                                                                                                                                                                                                                                                                                                                                                                                                                                                                                                                                                                                                                                                                                                                                                                                                                                                                                                                                                                                                                                                                                                                                                                                                                                                                                                                                                                                                                         |                                                                                                                                                                                                                                                                                                                                                                                                                                                                         |
|                                                                                                                            |                                                                                                                                                                                                                                                                                                                                                                                                                                                                                                                                                                                                                                                                                                                                                                                                                                                                                                                                                                                                                                                                                                                                                                                                                                                                                                                                                                                                                                                                                                                                                                                                                                                                                                                                                                                                                                                                                                                                                                                                                                                                                                                                                                                                                                                                                                                  |                                                                                                                                                                                                                                                                                                                                                                                                                                                                         |
| 78                                                                                                                         | diffusion whereas in the vegetated areas the plant-mediated transport is most prominent.                                                                                                                                                                                                                                                                                                                                                                                                                                                                                                                                                                                                                                                                                                                                                                                                                                                                                                                                                                                                                                                                                                                                                                                                                                                                                                                                                                                                                                                                                                                                                                                                                                                                                                                                                                                                                                                                                                                                                                                                                                                                                                                                                                                                                         | Formatted: Font: (Default) Times New Roman, 10 pt                                                                                                                                                                                                                                                                                                                                                                                                                       |
| 78
                                                                                                         | diffusion whereas in the vegetated areas the plant-mediated transport is most prominent.                                                                                                                                                                                                                                                                                                                                                                                                                                                                                                                                                                                                                                                                                                                                                                                                                                                                                                                                                                                                                                                                                                                                                                                                                                                                                                                                                                                                                                                                                                                                                                                                                                                                                                                                                                                                                                                                                                                                                                                                                                                                                                                                                                                                                  | Formatted: Font: (Default) Times New Roman, 10 pt                                                                                                                                                                                                                                                                                                                                                                                                                       |
| 78
                                                                                                   | diffusion whereas in the vegetated areas the plant-mediated transport is most prominent.
Urban wetlands have received extensive attention globally and their societal and economical importance have been
evaluated (Salminen et al., 2013), whereas their climate impact is still largely overlooked except for only a few studies                                                                                                                                                                                                                                                                                                                                                                                                                                                                                                                                                                                                                                                                                                                                                                                                                                                                                                                                                                                                                                                                                                                                                                                                                                                                                                                                                                                                                                                                                                                                                                                                                                                                                                                                                                                                                                                                                                                                                                 | Formatted: Font: (Default) Times New Roman, 10 pt                                                                                                                                                                                                                                                                                                                                                                                                                       |
| 78
                                                                                             | diffusion whereas in the vegetated areas the plant-mediated transport is most prominent.
Urban wetlands have received extensive attention globally and their societal and economical importance have been
evaluated (Salminen et al., 2013), whereas their climate impact is still largely overlooked except for only a few studies
(e.g. Morin et al., 2014a; Morin et al., 2014b). The thus farently review of GHG emission in constructed wetlands for                                                                                                                                                                                                                                                                                                                                                                                                                                                                                                                                                                                                                                                                                                                                                                                                                                                                                                                                                                                                                                                                                                                                                                                                                                                                                                                                                                                                                                                                                                                                                                                                                                                                                                                                                                                                                                        | Formatted: Font: (Default) Times New Roman, 10 pt                                                                                                                                                                                                                                                                                                                                                                                                                       |
| 78
                                                                                       | diffusion whereas in the vegetated areas the plant-mediated transport is most prominent.
Urban wetlands have received extensive attention globally and their societal and economical importance have been
evaluated (Salminen et al., 2013), whereas their climate impact is still largely overlooked except for only a few studies
(e.g. Morin et al., 2014a; Morin et al., 2014b). The thus farenty review of GHG emission in constructed wetlands for
wastewater treatment reported that the average CO 2 emission was 92.3 mg CO 2 -C m -2 h -1 and that the CH 4 emission ranged                                                                                                                                                                                                                                                                                                                                                                                                                                                                                                                                                                                                                                                                                                                                                                                                                                                                                                                                                                                                                                                                                                                                                                                                                                                                                                                                                                                                                                                                                                                                                                                                                                           | Formatted: Font: (Default) Times New Roman, 10 pt Formatted: Font: (Default) Times New Roman, 10 pt                                                                                                                                                                                                                                                                                                                                                                     |
| 78
                                                                                 | diffusion whereas in the vegetated areas the plant-mediated transport is most prominent.
Urban wetlands have received extensive attention globally and their societal and economical importance have been
evaluated (Salminen et al., 2013), whereas their climate impact is still largely overlooked except for only a few studies
(e.g. Morin et al., 2014a; Morin et al., 2014b). The thus farenty review of GHG emission in constructed wetlands for
wastewater treatment reported that the average CO 2 emission was 92.3 mg CO 2 -C m -2 h -1 and that the CH 4 emission ranged
from 1.6 to 27 mg CH 4 -C m -2 h -1 from free water surface (Mander et al., 2014). All of the studies were based on static                                                                                                                                                                                                                                                                                                                                                                                                                                                                                                                                                                                                                                                                                                                                                                                                                                                                                                                                                                                                                                                                                                                                                                                                                                                                                                                                                                                                                                                                            | Formatted: Font: (Default) Times New Roman, 10 pt Formatted: Font: (Default) Times New Roman, 10 pt Formatted: Font: (Default) Times New Roman, 10 pt                                                                                                                                                                                                                                                                                                                   |
| 78
                                                                           | diffusion whereas in the vegetated areas the plant-mediated transport is most prominent.
Urban wetlands have received extensive attention globally and their societal and economical importance have been
evaluated (Salminen et al., 2013), whereas their climate impact is still largely overlooked except for only a few studies
(e.g. Morin et al., 2014a; Morin et al., 2014b). The thus faronly review of GHG emission in constructed wetlands for
wastewater treatment reported that the average CO 2 emission was 92.3 mg CO 2 -C m -2 h -1 and that the CH 4 emission ranged
from 1.6 to 27 mg CH 4 -C m -2 h -1 from free water surface (Mander et al., 2014). All of the studies were based on static
chamber measurements during a short period so that the annual carbon balance of the ecosystem could not be assessed.                                                                                                                                                                                                                                                                                                                                                                                                                                                                                                                                                                                                                                                                                                                                                                                                                                                                                                                                                                                                                                                                                                                                                                                                                                                                                                                                                    | Formatted: Font: (Default) Times New Roman, 10 pt Formatted: Font: (Default) Times New Roman, 10 pt Formatted: Font: (Default) Times New Roman, 10 pt                                                                                                                                                                                                                                                                                                                   |
| 78
                                                                     | diffusion whereas in the vegetated areas the plant-mediated transport is most prominent.
Urban wetlands have received extensive attention globally and their societal and economical importance have been
evaluated (Salminen et al., 2013), whereas their climate impact is still largely overlooked except for only a few studies
(e.g. Morin et al., 2014a; Morin et al., 2014b) . The thus farenty review of GHG emission in constructed wetlands for
wastewater treatment reported that the average CO 2 emission was 92.3 mg CO 2 -C m -2 h -1 and that the CH4 emission ranged
from 1.6 to 27 mg CH4-C m -2 h -1 from free water surface (Mander et al., 2014) . All of the studies were based on static
chamber measurements during a short period so that the annual carbon balance of the ecosystem could not be assessed.
In contrast to static chamber measurements, the eddy covariance (EC) method provides continuous measurements of                                                                                                                                                                                                                                                                                                                                                                                                                                                                                                                                                                                                                                                                                                                                                                                                                                                                                                                                                                                                                                                                                                                                                                                                                                          | Formatted: Font: (Default) Times New Roman, 10 pt Formatted: Font: (Default) Times New Roman, 10 pt Formatted: Font: (Default) Times New Roman, 10 pt                                                                                                                                                                                                                                                                                                                   |
| 78
                                                               | diffusion whereas in the vegetated areas the plant-mediated transport is most prominent.
Urban wetlands have received extensive attention globally and their societal and economical importance have been
evaluated (Salminen et al., 2013), whereas their climate impact is still largely overlooked except for only a few studies
(e.g. Morin et al., 2014a; Morin et al., 2014b). The thus farenty review of GHG emission in constructed wetlands for
wastewater treatment reported that the average CO 2 emission was 92.3 mg CO 2 -C m -2 h -1 and that the CH 4 emission ranged
from 1.6 to 27 mg CH 4 -C m -2 h -1 from free water surface (Mander et al., 2014). All of the studies were based on static
chamber measurements during a short period so that the annual carbon balance of the ecosystem could not be assessed.
In contrast to static chamber measurements, the eddy covariance (EC) method provides continuous measurements of
GHG exchange at ecosystem scale, presenting the net result of fluxes as exchange in different source areas contributing                                                                                                                                                                                                                                                                                                                                                                                                                                                                                                                                                                                                                                                                                                                                                                                                                                                                                                                                                                                                                                                                                               | Formatted: Font: (Default) Times New Roman, 10 pt Formatted: Font: (Default) Times New Roman, 10 pt Formatted: Font: (Default) Times New Roman, 10 pt                                                                                                                                                                                                                                                                                                                   |
| 78
                                                         | diffusion whereas in the vegetated areas the plant-mediated transport is most prominent.
Urban wetlands have received extensive attention globally and their societal and economical importance have been
evaluated (Salminen et al., 2013), whereas their climate impact is still largely overlooked except for only a few studies
(e.g. Morin et al., 2014a; Morin et al., 2014b). The thus farenty review of GHG emission in constructed wetlands for
wastewater treatment reported that the average CO 2 emission was 92.3 mg CO 2 -C m -2 h -1 and that the CH 4 emission ranged
from 1.6 to 27 mg CH 4 -C m -2 h -1 from free water surface (Mander et al., 2014). All of the studies were based on static
chamber measurements during a short period so that the annual carbon balance of the ecosystem could not be assessed.
In contrast to static chamber measurements, the eddy covariance (EC) method provides continuous measurements of
GHG exchange at ecosystem scale, presenting the net result of fluxes as exchange in different source areas contributing
simultaneously within the footprint extent (Baldocchi, 2003). It is worth noticing that one of the assumptions of the EC                                                                                                                                                                                                                                                                                                                                                                                                                                                                                                                                                                                                                                                                                                                                                                                                                                                                                                                                                                   | Formatted: Font: (Default) Times New Roman, 10 pt                                                                                                                                                                                                                                                                 |
| 78
                                                   | diffusion whereas in the vegetated areas the plant-mediated transport is most prominent.
Urban wetlands have received extensive attention globally and their societal and economical importance have been
evaluated (Salminen et al., 2013), whereas their climate impact is still largely overlooked except for only a few studies
(e.g. Morin et al., 2014a; Morin et al., 2014b). The thus farently review of GHG emission in constructed wetlands for
wastewater treatment reported that the average CO 2 emission was 92.3 mg CO 2 -C m -2 h -1 and that the CH 4 emission ranged
from 1.6 to 27 mg CH 4 -C m -2 h -1 from free water surface (Mander et al., 2014). All of the studies were based on static
chamber measurements during a short period so that the annual carbon balance of the ecosystem could not be assessed.
In contrast to static chamber measurements, the eddy covariance (EC) method provides continuous measurements of
GHG exchange at ecosystem scale, presenting the net result of fluxes as exchange in different source areas contributing
simultaneously within the footprint extent (Baldocchi, 2003). It is worth noticing that one of the assumptions of the EC
method is surface homogeneity, yet in many study sites the <del>situations arelandscape setting is</del> far from ideal. The change                                                                                                                                                                                                                                                                                                                                                                                                                                                                                                                                                                                                                                                                                                                                                                                                                           | Formatted: Font: (Default) Times New Roman, 10 pt                                                                                                                                                                                                                                                                 |
| 78
                                             | diffusion whereas in the vegetated areas the plant-mediated transport is most prominent.
Urban wetlands have received extensive attention globally and their societal and economical importance have been
evaluated (Salminen et al., 2013), whereas their climate impact is still largely overlooked except for only a few studies
[e.g. Morin et al., 2014a; Morin et al., 2014b). The thus farenty review of GHG emission in constructed wetlands for
wastewater treatment reported that the average CO 2 emission was 92.3 mg CO 2 -C m -2 h -1 and that the CH4 emission ranged
from 1.6 to 27 mg CH4-C m -2 h -1 from free water surface [Mander et al., 2014). All of the studies were based on static
chamber measurements during a short period so that the annual carbon balance of the ecosystem could not be assessed.
In contrast to static chamber measurements, the eddy covariance (EC) method provides continuous measurements of
GHG exchange at ecosystem scale, presenting the net result of fluxes as exchange in different source areas contributing
simultaneously within the footprint extent [Baldocchi, 2003) . It is worth noticing that one of the assumptions of the EC
method is surface homogeneity, yet in many study sites the <del>situations arelandscape setting is</del> far from ideal. The change
of source area due to changes in wind provides difficulties in estimating GHG emissions in spatially heterogeneous sites                                                                                                                                                                                                                                                                                                                                                                                                                                                                                                                                                                                                                                                                                                   | Formatted: Font: (Default) Times New Roman, 10 pt                                                                                                                                                                                                                                                                 |
| 78
                                       | diffusion whereas in the vegetated areas the plant-mediated transport is most prominent.
Urban wetlands have received extensive attention globally and their societal and economical importance have been
evaluated (Salminen et al., 2013), whereas their climate impact is still largely overlooked except for only a few studies
(e.g. Morin et al., 2014a; Morin et al., 2014b). The thus farenty review of GHG emission in constructed wetlands for
wastewater treatment reported that the average CO 2 emission was 92.3 mg CO 2 -C m -2 h -1 and that the CH 4 emission ranged
from 1.6 to 27 mg CH 4 -C m -2 h -1 from free water surface (Mander et al., 2014). All of the studies were based on static
chamber measurements during a short period so that the annual carbon balance of the ecosystem could not be assessed.
In contrast to static chamber measurements, the eddy covariance (EC) method provides continuous measurements of
GHG exchange at ecosystem scale, presenting the net result of fluxes as exchange in different source areas contributing
simultaneously within the footprint extent (Baldocchi, 2003). It is worth noticing that one of the assumptions of the EC
method is surface homogeneity, yet in many study sites the situations arelandscape setting is far from ideal. The change
of source area due to changes in wind provides difficulties in estimating GHG emissions in spatially heterogeneous sites
especially in short-term flux measurements (Baldocchi et al., 2012). Therefore, for heterogeneous sites such as urban                                                                                                                                                                                                                                                                                                                                                                                                                                                                                                                                                                           | Formatted: Font: (Default) Times New Roman, 10 pt                                                                                                                                                                                                               |
| 78
                                 | diffusion whereas in the vegetated areas the plant-mediated transport is most prominent.
Urban wetlands have received extensive attention globally and their societal and economical importance have been
evaluated (Salminen et al., 2013), whereas their climate impact is still largely overlooked except for only a few studies
(e.g. Morin et al., 2014a; Morin et al., 2014b). The thus faronly review of GHG emission in constructed wetlands for
wastewater treatment reported that the average CO 2 emission was 92.3 mg CO 2 -C m -2 h -1 and that the CH 4 emission ranged
from 1.6 to 27 mg CH 4 -C m -2 h -1 from free water surface (Mander et al., 2014). All of the studies were based on static
chamber measurements during a short period so that the annual carbon balance of the ecosystem could not be assessed.
In contrast to static chamber measurements, the eddy covariance (EC) method provides continuous measurements of
GHG exchange at ecosystem scale, presenting the net result of fluxes as exchange in different source areas contributing
simultaneously within the footprint extent (Baldocchi, 2003). It is worth noticing that one of the assumptions of the EC
method is surface homogeneity, yet in many study sites the situations arelandscape setting is far from ideal. The change
of source area due to changes in wind provides difficulties in estimating GHG emissions in spatially heterogeneous sites
especially in short-term flux measurements (Baldocchi et al., 2012). Therefore, for heterogeneous sites such as urban
wetlands, accurate footprint modelling and surface area map at high spatial resolution are important in identifying the                                                                                                                                                                                                                                                                                                                                                                                                                                                | Formatted: Font: (Default) Times New Roman, 10 pt                                                                                                                     |
| 78
                           | diffusion whereas in the vegetated areas the plant-mediated transport is most prominent.
Urban wetlands have received extensive attention globally and their societal and economical importance have been
evaluated (Salminen et al., 2013), whereas their climate impact is still largely overlooked except for only a few studies
(e.g. Morin et al., 2014a; Morin et al., 2014b). The thus farenty review of GHG emission in constructed wetlands for
wastewater treatment reported that the average CO 2 emission was 92.3 mg CO 2 -C m -2 h -1 and that the CH 4 emission ranged
from 1.6 to 27 mg CH 4 -C m -2 h -1 from free water surface (Mander et al., 2014). All of the studies were based on static
chamber measurements during a short period so that the annual carbon balance of the ecosystem could not be assessed.
In contrast to static chamber measurements, the eddy covariance (EC) method provides continuous measurements of
GHG exchange at ecosystem scale, presenting the net result of fluxes as exchange in different source areas contributing
simultaneously within the footprint extent (Baldocchi, 2003). It is worth noticing that one of the assumptions of the EC
method is surface homogeneity, yet in many study sites the <del>situationg arelandscape setting is</del> far from ideal. The change
of source area due to changes in wind provides difficulties in estimating GHG emissions in spatially heterogeneous sites
especially in short-term flux measurements (Baldocchi et al., 2012). Therefore, for heterogeneous sites such as urban
wetlands, accurate footprint modelling and surface area map at high spatial resolution are important in identifying the
source area, and a land-surface specific analysis is vital to reveal the diel pattern, sink/source strength of the wetland,                                                                                                                                                                                                                                                                                                             | Formatted: Font: (Default) Times New Roman, 10 pt         Formatted: Font: (Default) Times New Roman, 10 pt |
| 78
                     | diffusion whereas in the vegetated areas the plant-mediated transport is most prominent.
Urban wetlands have received extensive attention globally and their societal and economical importance have been
evaluated (Salminen et al., 2013), whereas their climate impact is still largely overlooked except for only a few studies
(e.g. Morin et al., 2014a; Morin et al., 2014b). The thus farently review of GHG emission in constructed wetlands for
wastewater treatment reported that the average CO 2 emission was 92.3 mg CO 2 -C m -2 h -1 and that the CH 4 emission ranged
from 1.6 to 27 mg CH 4 -C m -2 h -1 from free water surface (Mander et al., 2014). All of the studies were based on static
chamber measurements during a short period so that the annual carbon balance of the ecosystem could not be assessed.
In contrast to static chamber measurements, the eddy covariance (EC) method provides continuous measurements of
GHG exchange at ecosystem scale, presenting the net result of fluxes as exchange in different source areas contributing
simultaneously within the footprint extent (Baldocchi, 2003). It is worth noticing that one of the assumptions of the EC
method is surface homogeneity, yet in many study sites the <del>situations arelandscape setting is</del> far from ideal. The change
of source area due to changes in wind provides difficulties in estimating GHG emissions in spatially heterogeneous sites
especially in short-term flux measurements (Baldocchi et al., 2012). Therefore, for heterogeneous sites such as urban
wetlands, accurate footprint modelling and surface area map at high spatial resolution are important in identifying the
source area, and a land-surface specific analysis is vital to reveal the diel pattern, sink/source strength of the wetland.
The objective of this study is to investigate how CO 2 and CH 4 surface-atmosphere exchange vary with seasonality and                                                                                                                                                             | Formatted: Font: (Default) Times New Roman, 10 pt         Formatted: Font: (Default) Times New Roman, 10 pt |
| 78
         | diffusion whereas in the vegetated areas the plant-mediated transport is most prominent.
Urban wetlands have received extensive attention globally and their societal and economical importance have been
evaluated (Salminen et al., 2013), whereas their climate impact is still largely overlooked except for only a few studies
(e.g. Morin et al., 2014a; Morin et al., 2014b). The thus farenty review of GHG emission in constructed wetlands for
wastewater treatment reported that the average CO 2 emission was 92.3 mg CO 2 -C m -2 h -1 and that the CH 4 emission ranged
from 1.6 to 27 mg CH 4 -C m -2 h -1 from free water surface (Mander et al., 2014). All of the studies were based on static
chamber measurements during a short period so that the annual carbon balance of the ecosystem could not be assessed.
In contrast to static chamber measurements, the eddy covariance (EC) method provides continuous measurements of
GHG exchange at ecosystem scale, presenting the net result of fluxes as exchange in different source areas contributing
simultaneously within the footprint extent (Baldocchi, 2003). It is worth noticing that one of the assumptions of the EC
method is surface homogeneity, yet in many study sites the situations arelandscape setting is far from ideal. The change
of source area due to changes in wind provides difficulties in estimating GHG emissions in spatially heterogeneous sites
especially in short-term flux measurements (Baldocchi et al., 2012). Therefore, for heterogeneous sites such as urban
wetlands, accurate footprint modelling and surface area map at high spatial resolution are important in identifying the
source area, and a land-surface specific analysis is vital to reveal the diel pattern, sink/source strength of the wetland.
The objective of this study is to investigate how CO 2 and CH 4 surface-atmosphere exchange vary with seasonality and
spatial beterogeneity and what the annual radiative forcing of these gases are in a constructed created urban wetland near                                    | Formatted: Font: (Default) Times New Roman, 10 pt         Formatted: Font: (Default) Times New Roman, 10 pt |
| 78
         | diffusion whereas in the vegetated areas the plant-mediated transport is most prominent.
Urban wetlands have received extensive attention globally and their societal and economical importance have been
evaluated (Salminen et al., 2013), whereas their climate impact is still largely overlooked except for only a few studies
(e.g. Morin et al., 2014a; Morin et al., 2014b). The thus faronly review of GHG emission in constructed wetlands for
wastewater treatment reported that the average CO 2 emission was 92.3 mg CO 2 -C m -2 h -1 and that the CH 4 emission ranged
from 1.6 to 27 mg CH 4 -C m -2 h -1 from free water surface (Mander et al., 2014). All of the studies were based on static
chamber measurements during a short period so that the annual carbon balance of the ecosystem could not be assessed.
In contrast to static chamber measurements, the eddy covariance (EC) method provides continuous measurements of
GHG exchange at ecosystem scale, presenting the net result of fluxes as exchange in different source areas contributing
simultaneously within the footprint extent (Baldocchi, 2003). It is worth noticing that one of the assumptions of the EC
method is surface homogeneity, yet in many study sites the situations arelandscape setting is far from ideal. The change
of source area due to changes in wind provides difficulties in estimating GHG emissions in spatially heterogeneous sites
especially in short-term flux measurements (Baldocchi et al., 2012). Therefore, for heterogeneous sites such as urban
wetlands, accurate footprint modelling and surface area map at high spatial resolution are important in identifying the
source area, and a land-surface specific analysis is vital to reveal the diel pattern, sink/source strength of the wetland.
The objective of this study is to investigate how CO 2 and CH 4 surface-atmosphere exchange vary with seasonality and
spatial heterogeneity and what the annual radiative forcing of these gases are in a constructed created urban wetland near
attewn the Nummela subur | Formatted: Font: (Default) Times New Roman, 10 pt                                                                                                                                                             |
| 78
| diffusion whereas in the vegetated areas the plant-mediated transport is most prominent.
Urban wetlands have received extensive attention globally and their societal and economical importance have been
evaluated (Salminen et al., 2013), whereas their climate impact is still largely overlooked except for only a few studies
(e.g. Morin et al., 2014a; Morin et al., 2014b). The thus faronly review of GHG emission in constructed wetlands for
wastewater treatment reported that the average CO 2 emission was 92.3 mg CO 2 -C m -2 h -1 and that the CH 4 emission ranged
from 1.6 to 27 mg CH 4 -C m -2 h -1 from free water surface (Mander et al., 2014). All of the studies were based on static
chamber measurements during a short period so that the annual carbon balance of the ecosystem could not be assessed.
In contrast to static chamber measurements, the eddy covariance (EC) method provides continuous measurements of
GHG exchange at ecosystem scale, presenting the net result of fluxes as exchange in different source areas contributing
simultaneously within the footprint extent (Baldocchi, 2003). It is worth noticing that one of the assumptions of the EC
method is surface homogeneity, yet in many study sites the situationg arelandscape setting is far from ideal. The change
of source area due to changes in wind provides difficulties in estimating GHG emissions in spatially heterogeneous sites
especially in short-term flux measurements (Baldocchi et al., 2012). Therefore, for heterogeneous sites such as urban
wetlands, accurate footprint modelling and surface area map at high spatial resolution are important in identifying the
source area, and a land-surface specific analysis is vital to reveal the diel pattern, sink/source strength of the wetland.
The objective of this study is to investigate how CO 2 and CH 4 surface-atmosphere exchange vary with seasonality and
spatial heterogeneity and what the annual radiative forcing of these gases are in a constructed created urban wetland near
attown the Nummela suburb . Mun               | Formatted: Font: (Default) Times New Roman, 10 pt         Formatted: Font: (Default) Times New Roman, 10 pt |

biodiversity. Besides taking advantage of ecosystem-scale EC measurements, we also parse the variability of gas exchange
induced by surface heterogeneity (the open water and the vegetated areas) using diffusional flux modeling and footprint
modelling overlapped on a high-resolution surface map. To illustrate how the urban wetland functions as a source or a
sink of GHG equivalents, we calculate separately the sustained global warming potential (SGWP) of CO2 and CH4 over
a hundred-year horizon in each surface type.

**102 2 Materials and Methods**

**103 2.1 Site description**

Our study site is a created stormwater wetland Gateway, located by an eutrophicated Lake Enäjärvi in the District of
 Nummela, Municipality of Vihti, Southern Finland (60.3272°N, 24.3369°E). Southern Finland experiences a climate with
 a 30-year mean air temperature of 4.6 °C and an annual precipitation rate of 627 mm in the period of year 1981-2010
 (Pirinen et al., 2012),

The wetland was constructed in 2010 at the mouth of a 550 hectare largely urbanized (35 % impervious) watershed of 109 Stream Kilsoi. It was excavated over six weeks in early winter 2010 on an abandoned agricultural field growing meadow 110 vegetation. All of the old drainage ditches were blocked as amphibian habitats, which also ensured only one inlet route 111 receiving water from Stream Kilsoi and one outlet route discharging water to the nearby Lake Enäjärvi. Lake Enäjärvi is 112 an eutrophicated lake. The internal phosphorus load from human activities and the run-off from its catchments have 113 resulted in regular cyanobacterial blooms and fish kills in the lake (Varis et al., 1989; Salonen et al., 2000). 114 The wetland park has a total area of 7 hectares within which - during mean water flow conditions - a 0.5 hectare 115 inundated wetland is located. This stormwater treatment wetland consists of an inlet stilling pond, a meandering shallow 116 water area with three habitat islands, and an outlet pond. The average water depth in the ponds is 1.5 m; within 117 emergent vegetation patches water depth ranges between 0.3 and 0.5 m. There are also submerged macrophytes in the 118 open water as the water is shallow, thus in the paper we refer the "vegetated area" to the area with emergent vegetation 119 and "open water" to the area covered by water in the absence of emergent macrophytes. The outlet bottom dam sets a 120 low water level (WL) to 50.04 m above the Baltic Sea level (N60+ coordinate system). Herbaceous vegetation has been 121 allowed to fully self-establish after the construction of the wetland. Annual monitoring of vegetation carried out in 122 summers 2010, 2011 and 2012 indicated rapid self-establishment of vegetation which was rich in taxa and dominated 123 by native species (Wahlroos et al., 2015). At frequently-inundated area (elevation levels of 50-50.35 m), vegetation was 124 arranged in dense patches with different dominating wetland plant species: Typha latifolia L., Iris pseudacorus L., 125 Carex spp. or Juncus effuses L. At the major less-frequently inundated area (elevation levels of 50.35-50.45m), the wet 126 meadow species Filipendula ulmaria L. (Maxim.), Lysimachia vulgaris L., and Lythrum salicaria L. with the three 127 species co-existing at 1:1:1 ratio formed the plant community. Drier areas (elevation levels of 50.45-50.60 m) were 128 mostly colonized by dry meadow species such as Poa spp. and Calamagrostis spp., including patches dominated by 129 Cirsium species (Fig. S1). Note that the area with water level lower than 49.5 m is defined as the open water area while 130 the rest is defined as the vegetated area in this study. 131

2.2 Water and micrometeorological measurements

| Formatted: Font: 10 pt, (Asian) Chinese (PRC)                                     |
|-----------------------------------------------------------------------------------|
| Formatted: Font: 10 pt, (Asian) Chinese (PRC)                                     |
| Formatted: Font: 10 pt, (Asian) Chinese (PRC)                                     |
| Formatted: Font: 10 pt, (Asian) Chinese (PRC)                                     |
| Formatted: Font: 10 pt, (Asian) Chinese (PRC), (Other)
English (United States) |
| Formatted: Font: (Default) Times New Roman, 10 pt                                 |

Water monitoring stations were set up at the inlet (60.3283° N, 24.3356° E) and at the outlet (60.3281° N, 24.3377° E) of 134 the wetland. During the 2012-2013 and 2013-2014 monitoring periods, water temperature as well as water turbidity, 135 oxygen concentration, conductivity and pH were measured at the inlet and outlet monitoring station with the YSI-60600 136 series multiparameter sonde (YSI Inc., Yellow Springs, OH, USA). Measurements were conducted continuously with a 137 10-minute interval. Water level at the outlet was measured continuously with a pressure gauge (STS sensor, Sensor 138 Technik Sirnach AG, Switzerland). At the outlet monitoring station, the concentration of dissolved carbon dioxide ([CO2]) 139 and dissolved methane ([CH4]) were measured with Contros HydroC™ CO2 and HydroC™ CH4 sensors (CONTROS 140 Systems & Solutions GmbH, Germany). In 2014, the same sensors were also installed at the inlet monitoring station to 141 measure [CO2] and [CH4]. Dissolved CO2 and CH4 molecules diffuse from water column into the detection chamber 142 through a thin-film composite membrane where the concentration of CO2 and CH4 is determined by means of IR 143 absorption spectrometry and Tunable Diode Laser Absorption Spectroscopy, respectively. NO3-N and total phosphorus 144 (TP) data data have been previously published in Wahlroos et al. (2015). Briefly, NQs-N was measured with Scan sensors 145 (Scan gmbh, Austria) and TP was ealculated estimated based on turbidity data-which was me measasured at 10-min 146 intervals. (Valkama et al., 2017).

[revised manuscript text omitted]

| 211      | In order to be confident in our gap-filling results, we also applied alternative gap-filling methods to EC fluxes using                                                                                                                                                                                                                                                                                                                                                                                                                                                                                                                                                                                                                                                                                                                                                                                                                                                                                                                                                                                                                                                                                                                                                                                                                                                                                                                                                                                                                                                                                                                                                                                                                                                                                                                                                                                                                                                                                                                                                                                                                                                                                                                                                                                                                                                                                                                                                                                                                                                                                                                                                                                                                                                                                                                                                                                                                                                                                                                                                                                                                                                                                                                                                                                                                                                                                                                                                                                                                          |                                                    |
|----------|--------------------------------------------------------------------------------------------------------------------------------------------------------------------------------------------------------------------------------------------------------------------------------------------------------------------------------------------------------------------------------------------------------------------------------------------------------------------------------------------------------------------------------------------------------------------------------------------------------------------------------------------------------------------------------------------------------------------------------------------------------------------------------------------------------------------------------------------------------------------------------------------------------------------------------------------------------------------------------------------------------------------------------------------------------------------------------------------------------------------------------------------------------------------------------------------------------------------------------------------------------------------------------------------------------------------------------------------------------------------------------------------------------------------------------------------------------------------------------------------------------------------------------------------------------------------------------------------------------------------------------------------------------------------------------------------------------------------------------------------------------------------------------------------------------------------------------------------------------------------------------------------------------------------------------------------------------------------------------------------------------------------------------------------------------------------------------------------------------------------------------------------------------------------------------------------------------------------------------------------------------------------------------------------------------------------------------------------------------------------------------------------------------------------------------------------------------------------------------------------------------------------------------------------------------------------------------------------------------------------------------------------------------------------------------------------------------------------------------------------------------------------------------------------------------------------------------------------------------------------------------------------------------------------------------------------------------------------------------------------------------------------------------------------------------------------------------------------------------------------------------------------------------------------------------------------------------------------------------------------------------------------------------------------------------------------------------------------------------------------------------------------------------------------------------------------------------------------------------------------------------------------------------------------------|----------------------------------------------------|
| 212      | parameterization based on biological principles (see Supplement Material). The results on annual cumulative fluxes were                                                                                                                                                                                                                                                                                                                                                                                                                                                                                                                                                                                                                                                                                                                                                                                                                                                                                                                                                                                                                                                                                                                                                                                                                                                                                                                                                                                                                                                                                                                                                                                                                                                                                                                                                                                                                                                                                                                                                                                                                                                                                                                                                                                                                                                                                                                                                                                                                                                                                                                                                                                                                                                                                                                                                                                                                                                                                                                                                                                                                                                                                                                                                                                                                                                                                                                                                                                                                          |                                                    |
| 213      | not significantly different from the ones gap-filled using ANN, thus we only report the results from ANN in the following                                                                                                                                                                                                                                                                                                                                                                                                                                                                                                                                                                                                                                                                                                                                                                                                                                                                                                                                                                                                                                                                                                                                                                                                                                                                                                                                                                                                                                                                                                                                                                                                                                                                                                                                                                                                                                                                                                                                                                                                                                                                                                                                                                                                                                                                                                                                                                                                                                                                                                                                                                                                                                                                                                                                                                                                                                                                                                                                                                                                                                                                                                                                                                                                                                                                                                                                                                                                                        |                                                    |
| 214      | text.                                                                                                                                                                                                                                                                                                                                                                                                                                                                                                                                                                                                                                                                                                                                                                                                                                                                                                                                                                                                                                                                                                                                                                                                                                                                                                                                                                                                                                                                                                                                                                                                                                                                                                                                                                                                                                                                                                                                                                                                                                                                                                                                                                                                                                                                                                                                                                                                                                                                                                                                                                                                                                                                                                                                                                                                                                                                                                                                                                                                                                                                                                                                                                                                                                                                                                                                                                                                                                                                                                                                            |                                                    |
| 215      | The gap-filled net ecosystem exchange (NEE) can be further partitioned into two components gross ecosystem production                                                                                                                                                                                                                                                                                                                                                                                                                                                                                                                                                                                                                                                                                                                                                                                                                                                                                                                                                                                                                                                                                                                                                                                                                                                                                                                                                                                                                                                                                                                                                                                                                                                                                                                                                                                                                                                                                                                                                                                                                                                                                                                                                                                                                                                                                                                                                                                                                                                                                                                                                                                                                                                                                                                                                                                                                                                                                                                                                                                                                                                                                                                                                                                                                                                                                                                                                                                                                            |                                                    |
| 216      | (GEP) and ecosystem respiration ( $R_{eco}$ ) according to the following equation:                                                                                                                                                                                                                                                                                                                                                                                                                                                                                                                                                                                                                                                                                                                                                                                                                                                                                                                                                                                                                                                                                                                                                                                                                                                                                                                                                                                                                                                                                                                                                                                                                                                                                                                                                                                                                                                                                                                                                                                                                                                                                                                                                                                                                                                                                                                                                                                                                                                                                                                                                                                                                                                                                                                                                                                                                                                                                                                                                                                                                                                                                                                                                                                                                                                                                                                                                                                                                                                               |                                                    |
| 217      | $NEE = GEP + R_{eco,} \tag{1}$                                                                                                                                                                                                                                                                                                                                                                                                                                                                                                                                                                                                                                                                                                                                                                                                                                                                                                                                                                                                                                                                                                                                                                                                                                                                                                                                                                                                                                                                                                                                                                                                                                                                                                                                                                                                                                                                                                                                                                                                                                                                                                                                                                                                                                                                                                                                                                                                                                                                                                                                                                                                                                                                                                                                                                                                                                                                                                                                                                                                                                                                                                                                                                                                                                                                                                                                                                                                                                                                                                                   |                                                    |
| 218      | where positive Record represents a net carbon flux from the ecosystem to the atmosphere and negative GEP represents a net                                                                                                                                                                                                                                                                                                                                                                                                                                                                                                                                                                                                                                                                                                                                                                                                                                                                                                                                                                                                                                                                                                                                                                                                                                                                                                                                                                                                                                                                                                                                                                                                                                                                                                                                                                                                                                                                                                                                                                                                                                                                                                                                                                                                                                                                                                                                                                                                                                                                                                                                                                                                                                                                                                                                                                                                                                                                                                                                                                                                                                                                                                                                                                                                                                                                                                                                                                                                                        |                                                    |
| 219      | carbon input from the atmosphere to the ecosystem. Thus the negative NEE indicates that the ecosystem is a carbon sink                                                                                                                                                                                                                                                                                                                                                                                                                                                                                                                                                                                                                                                                                                                                                                                                                                                                                                                                                                                                                                                                                                                                                                                                                                                                                                                                                                                                                                                                                                                                                                                                                                                                                                                                                                                                                                                                                                                                                                                                                                                                                                                                                                                                                                                                                                                                                                                                                                                                                                                                                                                                                                                                                                                                                                                                                                                                                                                                                                                                                                                                                                                                                                                                                                                                                                                                                                                                                           |                                                    |
| 220      | and the positive NEE means the ecosystem is a carbon source. Reco was estimated using a model describing the temperature                                                                                                                                                                                                                                                                                                                                                                                                                                                                                                                                                                                                                                                                                                                                                                                                                                                                                                                                                                                                                                                                                                                                                                                                                                                                                                                                                                                                                                                                                                                                                                                                                                                                                                                                                                                                                                                                                                                                                                                                                                                                                                                                                                                                                                                                                                                                                                                                                                                                                                                                                                                                                                                                                                                                                                                                                                                                                                                                                                                                                                                                                                                                                                                                                                                                                                                                                                                                                         |                                                    |
| 221      | dependence of R eco                                                                                                                                                                                                                                                                                                                                                                                                                                                                                                                                                                                                                                                                                                                                                                                                                                                                                                                                                                                                                                                                                                                                                                                                                                                                                                                                                                                                                                                                                                                                                                                                                                                                                                                                                                                                                                                                                                                                                                                                                                                                                                                                                                                                                                                                                                                                                                                                                                                                                                                                                                                                                                                                                                                                                                                                                                                                                                                                                                                                                                                                                                                                                                                                                                                                                                                                                                                                                                                                                                                   |                                                    |
| 222      | $R_{\text{reces}} = R_{\text{ref}} \left[ \frac{\left[ E\left(\frac{1}{2\pi}, \frac{1}{2\pi} \frac{1}{2\pi}$ | Formatted                                          |
| 223      | where $E = 346.37$ K is an activation energy related physiological parameter. $T_{abc}$ is the air temperature. $T_d = 56.02$ K and                                                                                                                                                                                                                                                                                                                                                                                                                                                                                                                                                                                                                                                                                                                                                                                                                                                                                                                                                                                                                                                                                                                                                                                                                                                                                                                                                                                                                                                                                                                                                                                                                                                                                                                                                                                                                                                                                                                                                                                                                                                                                                                                                                                                                                                                                                                                                                                                                                                                                                                                                                                                                                                                                                                                                                                                                                                                                                                                                                                                                                                                                                                                                                                                                                                                                                                                                                                                              |                                                    |
| 224      | $T_{i}=227.13$ K (Lloyd and Taylor, 1994; Aurela et al., 2009). $R_{i}$ is the rate of ecosystem respiration at 10 °C. We first fitted                                                                                                                                                                                                                                                                                                                                                                                                                                                                                                                                                                                                                                                                                                                                                                                                                                                                                                                                                                                                                                                                                                                                                                                                                                                                                                                                                                                                                                                                                                                                                                                                                                                                                                                                                                                                                                                                                                                                                                                                                                                                                                                                                                                                                                                                                                                                                                                                                                                                                                                                                                                                                                                                                                                                                                                                                                                                                                                                                                                                                                                                                                                                                                                                                                                                                                                                                                                                           | Formatted: Font: (Default) Times New Roman 10 nt   |
| 225      | the model with nighttime NEE (which represents the nighttime ecosystem respiration since photosynthesis is assumed to